# Targeting SWI/SNF ATPases in enhancer-addicted prostate cancer

Lanbo Xiao[1,2,14], Abhijit Parolia[1,2,3,14], Yuanyuan Qiao[1,2,4,14], Pushpinder Bawa[1,2], Sanjana Eyunni[1,2,3], Rahul Mannan[1,2], Sandra E. Carson[1,2], Yu Chang[1,2], Xiaoju Wang[1,2,4], Yuping Zhang[1,2], Josh N. Vo[1,2,5], Steven Kregel[1,2], Stephanie A. Simko[1,2], Andrew D. Delekta[1,2], Mustapha Jaber[1], Heng Zheng[1,2], Ingrid J. Apel[1,2], Lisa McMurry[1,2], Fengyun Su[1,2], Rui Wang[1,2], Sylvia Zelenka-Wang[1,2], Sanjita Sasmal[6], Leena Khare[6], Subhendu Mukherjee[6], Chandrasekhar Abbineni[6], Kiran Aithal[6], Mital S. Bhakta[7], Jay Ghurye[7], Xuhong Cao[1,2,8], Nora M. Navone[9], Alexey I. Nesvizhskii[1,2,4,5], Rohit Mehra[1,2,4], Ulka Vaishampayan[10], Marco Blanchette[7], Yuzhuo Wang[11,12], Susanta Samajdar[6], Murali Ramachandra[6] & Arul M. Chinnaiyan[1,2,4,5,8,13 ✉]

The switch/sucrose non-fermentable (SWI/SNF) complex has a crucial role in chromatin remodelling[1] and is altered in over 20% of cancers[2,3]. Here we developed a proteolysis-targeting chimera (PROTAC) degrader of the SWI/SNF ATPase subunits, SMARCA2 and SMARCA4, called AU-15330. Androgen receptor (AR)⁺ forkhead box A1 (FOXA1)⁺ prostate cancer cells are exquisitely sensitive to dual SMARCA2 and SMARCA4 degradation relative to normal and other cancer cell lines. SWI/SNF ATPase degradation rapidly compacts *cis*-regulatory elements bound by transcription factors that drive prostate cancer cell proliferation, namely AR, FOXA1, ERG and MYC, which dislodges them from chromatin, disables their core enhancer circuitry, and abolishes the downstream oncogenic gene programs. SWI/SNF ATPase degradation also disrupts super-enhancer and promoter looping interactions that wire supra-physiologic expression of the *AR*, *FOXA1* and *MYC* oncogenes themselves. AU-15330 induces potent inhibition of tumour growth in xenograft models of prostate cancer and synergizes with the AR antagonist enzalutamide, even inducing disease remission in castration-resistant prostate cancer (CRPC) models without toxicity. Thus, impeding SWI/SNF-mediated enhancer accessibility represents a promising therapeutic approach for enhancer-addicted cancers.

In eukaryotic cells, DNA is wrapped around histone octamers (referred to as nucleosomes), which form a physical barrier to DNA-based processes[4]. Thus, gene expression is regulated by modifying physical accessibility of the DNA through nucleosomal remodelling and, when in an accessible state, through binding of transcription factors[5,6]. In this regulatory context, non-coding genomic elements called enhancers have emerged as central hubs serving as integrative platforms for transcription factor binding and activation of lineage-specific gene programs[7,8]. The enhancer elements can lie within untranslated or distal intergenic regions and make looping interactions with their target gene promoters to potentiate RNA polymerase II (PolII)-mediated transcription[9,10].

In cancer, genetic alterations invariably lead to an aberrant transcriptional state that is often wired through expansion and remodelling of the enhancer landscape[11,12]. This includes de novo commissioning of new enhancers (neo-enhancers) by reprogramming of pioneer factor cistromes[13], enhancer hijacking via structural rearrangements[14,15], and/or abnormal enhancer–promoter interactions via alterations in chromatin topology[16]—all to enable hyper-expression of driver oncogenes. Although there has been intense interest in therapeutically targeting aberrant enhancer function in cancer, the molecular machinery responsible for enhancer maintenance and/or activation remains poorly characterized.

Recent studies have uncovered alterations in genes encoding constituent subunits of the SWI/SNF complex in over 20% of human cancers[2]. SWI/SNF is a multi-subunit chromatin-remodelling complex that uses energy from ATP hydrolysis to reposition or eject nucleosomes at non-coding regulatory elements, thereby enabling free DNA access for the transcriptional machinery[1]. In SWI/SNF-mutant tumours, the residual complex is thought to enable oncogenic transcriptional programs and speculated to be a viable therapeutic target[17–19]. Although

[1]Michigan Center for Translational Pathology, University of Michigan, Ann Arbor, MI, USA. [2]Department of Pathology, University of Michigan, Ann Arbor, MI, USA. [3]Molecular and Cellular Pathology Program, University of Michigan, Ann Arbor, MI, USA. [4]Rogel Cancer Center, University of Michigan, Ann Arbor, MI, USA. [5]Department of Computational Medicine and Bioinformatics, University of Michigan, Ann Arbor, MI, USA. [6]Aurigene Discovery Technologies, Electronic City Phase II, Bangalore, India. [7]Dovetail Genomics, Scotts Valley, CA, USA. [8]Howard Hughes Medical Institute, University of Michigan, Ann Arbor, MI, USA. [9]Department of Genitourinary Medical Oncology and the David H. Koch Center for Applied Research of Genitourinary Cancers, The University of Texas MD Anderson Cancer Center, Houston, TX, USA. [10]Department of Internal Medicine/Oncology, University of Michigan, Ann Arbor, MI, USA. [11]Vancouver Prostate Centre, Vancouver, British Columbia, Canada. [12]Department of Urologic Sciences, University of British Columbia, Vancouver, British Columbia, V6T 1Z3, Canada. [13]Department of Urology, University of Michigan, Ann Arbor, MI, USA. [14]These authors contributed equally: Lanbo Xiao, Abhijit Parolia, Yuanyuan Qiao. ✉e-mail: arul@umich.edu

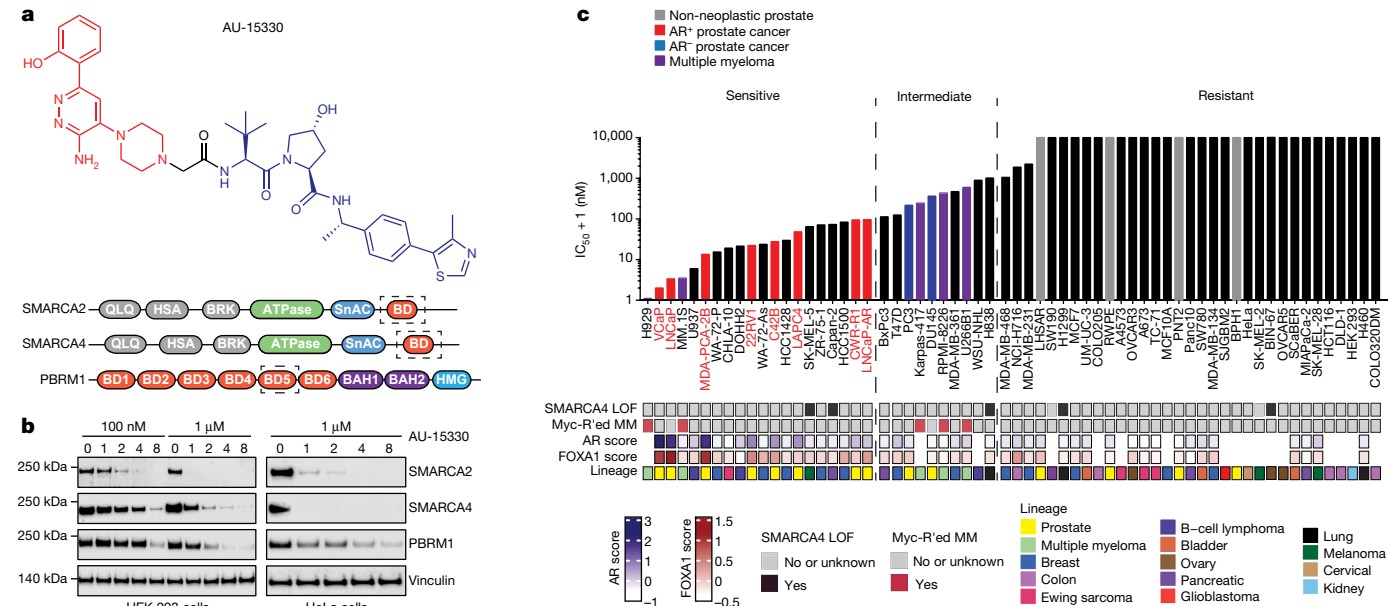

**Fig. 1 | AU-15330, a specific degrader of SWI/SNF ATPases, exhibits preferential cytotoxicity in enhancer-binding transcription factor-driven cancers. a**, Structure of AU-15330 and schematic of SMARCA2, SMARCA4 and PBRM1 domains. AU-15330-targeted bromodomains (BD) are shown. QLQ, conserved Gln, Leu, Gln motif containing domain; HSA, helicase/SANT-associated domain; BRK, Brahma and Kismet domain; SnAC, Snf2 ATP coupling domain; BAH1, bromo-adjacent homology domain 1; BAH2, bromo-adjacent homology domain 2. **b**, Immunoblots of SMARCA2, SMARCA4 and PBRM1 on treatment of HEK 293 and HeLa cells with AU-15330 at increasing concentrations or time durations. Vinculin is used as a loading control, and is probed on a representative immunoblot. This experiment was repeated independently twice. **c**, $IC_{50}$ of AU-15330 in a panel of human-derived cancer or normal cell lines after 5 days of treatment. Known SMARCA4 loss-of-function (LOF) alterations and multiple myeloma (MM) cell lines with MYC rearrangements (MYC-R'ed) are identified below the graph. AR and FOXA1 scores quantify their transcriptional activities using cognate multi-gene signatures.

inhibitors and degraders of ATPase and BRD7–BRD9 SWI/SNF subunits have been recently developed[20–22], to our knowledge, no studies have comprehensively assessed the therapeutic efficacy of SWI/SNF inactivation across a wide spectrum of cancers. To this end, we have developed and characterized a highly-selective PROTAC degrader of both SWI/SNF ATPase subunits—SMARCA2 (BRM) and SMARCA4 (BRG1)—that are required for the nucleosomal-remodelling functions of SWI/SNF complexes.

We found enhancer-binding transcription factor-addicted cancers (for example, AR−FOXA1-driven prostate cancer) to be exquisitely and preferentially sensitive to SWI/SNF ATPase degradation, which triggered an instantaneous, specific loss of physical accessibility and transcription factor binding at enhancer elements, thereby disrupting enhancer-wired oncogenic gene programs. To our knowledge, this study is the first preclinical proof of concept that targeted obstruction of chromatin accessibility at enhancer elements may be a potent therapeutic strategy in transcription factor-addicted tumours.

## Results

We developed the PROTAC degrader, AU-15330, comprising a bait moiety that binds the bromodomain in SMARCA2 and SMARCA4 and a ligand moiety for the von Hippel–Lindau (VHL) ubiquitin ligase (Fig. 1a, Extended Data Fig. 1a). AU-15330 also binds to the secondary SWI/SNF module component PBRM1, which relies on the ATPase module for assembly onto the core complex[23]. Although it binds to the same bromodomain in target proteins as the PROTAC degrader ACBI1[20], AU-15330 comprises a distinct linker structure that largely dictates a PROTAC's target selectivity and degradation kinetics[24]. Treatment of several cell lines with AU-15330 led to time and dose-dependent degradation of SMARCA2, SMARCA4 and PBRM1 (Fig. 1b). Mass spectrometry-based proteomics analysis confirmed SMARCA2, SMARCA4 and PBRM1 as the only significantly downregulated proteins (Extended Data Fig. 1b). Of note, we detected no change in the abundance of other bromodomain-containing proteins or non-targeted SWI/SNF subunits (Extended Data Fig. 1c, d). SWI/SNF complexes have been shown to assemble in a modular manner, with the ATPase module being the last to bind to the SMARCC1 (also known as BAF155)-containing core complex[23]. Accordingly, SMARCC1 nuclear immunoprecipitation followed by mass spectrometry showed no changes in the sequential assembly of the core and secondary modules but revealed detachment of ATPase module subunits upon AU-15330 treatment (Extended Data Fig. 1e).

Using a panel of normal and cancer cell lines from 14 distinct lineages, we found AR and FOXA1-driven prostate cancer cells to be preferentially sensitive to AU-15330 (half-maximal inhibitory concentrations ($IC_{50}$) <100 nM; Fig. 1c, Extended Data Fig. 1f, g, Supplementary Table 1). AR−FOXA1− prostate cancer cells showed moderate sensitivity ($IC_{50}$ between 100–400 nM), whereas normal and non-neoplastic prostate cells were resistant ($IC_{50}$ >1,000 nM) to AU-15330. We observed a similar cytotoxicity profile for ACBI1 and BRM014, an allosteric dual inhibitor of SMARCA2 and SMARCA4 ATPase activity[25] (Extended Data Fig. 1h, i). Notably, AR+FOXA1+ prostate cancer cells were more sensitive to these inhibitors than SMARCA4-null cancer cell lines. Several MYC-driven multiple myeloma cells and oestrogen receptor- and/or AR-positive breast cancer cells were also acutely sensitive to AU-15330 (Fig. 1c, Extended Data Fig. 1j, k).

In several prostate cancer cell lines, we detected substantial expression of both SWI/SNF ATPases, which were rapidly degraded in a dose-dependent manner by AU-15330 (Extended Data Fig. 2a, b). Concordantly, AU-15330 attenuated the growth of these cells and induced apoptotic cell death, while having no anti-proliferative effect on benign or non-neoplastic prostate cells (grey bars, Fig. 1c) at parallel doses (Extended Data Fig. 1f, 2c–e). Treatment with either the bromodomain ligand alone (AU-15139) or an inactive epimer of AU-15330 (AU-16235) had no effect on target protein levels or

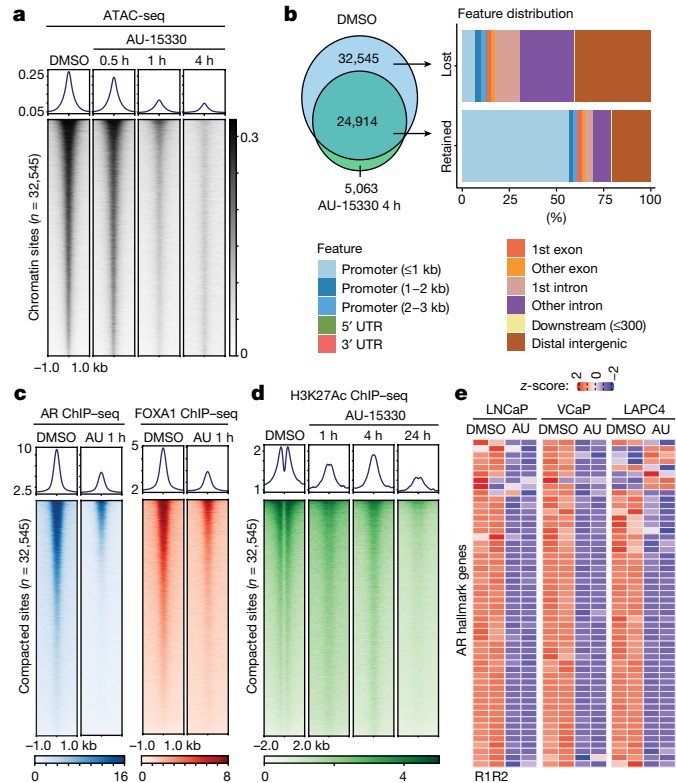

**Fig. 2 | SWI/SNF ATPase degradation disrupts physical chromatin accessibility at the core-enhancer circuitry to disable oncogenic transcriptional programs. a**, ATAC-seq read-density heat maps from VCaP cells treated with DMSO or AU-15330 for indicated durations (*n* = 2 biological replicates). **b**, Genome-wide changes in chromatin accessibility upon AU-15330 treatment for 4 h in VCaP cells along with genomic annotation of sites that lose physical accessibility (lost) or remain unaltered (retained). **c**, **d**, ChIP–seq read-density heat maps for AR and FOXA1 (**c**) and H3K27Ac (**d**) at the AU-15330 (AU)-compacted genomic sites after treatment with DMSO or AU-15330 (1 μM) for indicated times and stimulation with R1881 (1 nM, 3 h). **e**, RNA-seq heat maps for classical AR target genes in LNCaP, VCaP and LAPC4 prostate cancer cells with or without 24 h of AU-15330 treatment.

cancer cell survival and growth (Extended Data Figs. 1f, g, 2f, g). Next, competition of AU-15330 with a free VHL ligand (VL285), but not with thalidomide, reversed degradation of SWI/SNF targets (Extended Data Fig. 2g) and rescued the growth inhibitory effect in a dose-dependent manner (Extended Data Fig. 2h). Furthermore, pre-treatment of VCaP cells (an AR+FOXA1+ prostate cancer cell line model) with bortezomib (a proteasome inhibitor) or MLN4924 (a NEDD8-activating enzyme inhibitor) hindered target protein degradation, indicating that AU-15330 requires the proteasome machinery and ubiquitination cascade for its action (Extended Data Fig. 2g).

As SWI/SNF complexes actively remodel nucleosomal DNA packaging, we profiled the effect of AU-15330 on physical chromatin accessibility using the assay for transposase-accessible chromatin followed by sequencing (ATAC-seq). We detected a rapid and near-complete loss in chromatin accessibility at more than 30,000 sites in VCaP cells with as little as 1 h of AU-15330 treatment (Fig. 2a), which is within minutes of SMARCA2 and SMARCA4 degradation (Extended Data Fig. 3a); approximately 25,000 genomic sites showed little to no change in nucleosomal density (Extended Data Fig. 3b). Similar profound changes in chromatin accessibility were not observed upon treatment with a BRD4 degrader (ZBC-260; Extended Data Fig. 3a, b). In our genetic models using CRISPR–Cas9 and shRNA-mediated target inactivation, we detected a significant compaction of the chromatin only upon concurrent loss of both SWI/SNF ATPases (Extended Data Fig. 3c, d). More

than 90% of the AU-15330-compacted sites were within distal regulatory regions, which were enriched for enhancers, whereas the retained sites were predominantly within promoters (Fig. 2b). De novo motif and binding analysis for the regulation of transcription (BART) analyses of AU-15330-compacted sites identified DNA-binding elements for major oncogenic transcription factors in prostate cancer, including AR, FOXA1, HOXB13 and ERG (Extended Data Fig. 3e, f). As expected, retained promoter sites showed enrichment for PolII and E2F motifs (Extended Data Fig. 3g). Interrogation of chromatin changes in LNCaP cells upon AU-15330 treatment reproduced these findings (Extended Data Fig. 4a–c).

Concurrent with the loss of accessibility, chromatin immunoprecipitation followed by sequencing (ChIP–seq) revealed a decrease in chromatin binding of AR, FOXA1, and ERG in VCaP cells within 1 h of AU-15330 treatment (Fig. 2c, Extended Data Fig. 4d, e). We also detected disappearance of the characteristic 'valley' pattern in the H3K27Ac ChIP–seq signal, indicating the movement of flanking nucleosomes towards the centre of AU-15330-compacted enhancers (Fig. 2d). At early time points, we detected no loss in the abundance of the H3K27Ac mark; however, it was significantly depleted 24 h after AU-15330 treatment (Extended Data Fig. 4f). Similar results were observed upon AU-15330 treatment of LNCaP cells (Extended Data Fig. 4g, h). Loss of AR, FOXA1 and H3K27Ac ChIP signals was evident at enhancer sites of the classical AR target gene *KLK3* (Extended Data Fig. 4i). We found AR, FOXA1, ERG and SMARCC1 to co-occupy a large fraction of H3K27Ac-marked regulatory elements (Extended Data Fig. 5a–c). Furthermore, multiple core SWI/SNF components were present in the mass spectrometry-based datasets of AR, FOXA1, and ERG interactomes (Extended Data Fig. 5d), which we confirmed by reciprocal co-immunoprecipitation assays (Extended Data Fig. 5e). This positions SWI/SNF complexes as common chromatin cofactors of the oncogenic transcriptional machinery in prostate cancer cells. As an important control, we saw no changes in chromatin binding of CTCF in AU-15330-treated cells (Extended Data Fig. 6a–d).

Global transcriptomic profiling with RNA sequencing (RNA-seq) revealed significant downregulation of AR and FOXA1-regulated genes in multiple prostate cancer cells, as well as ERG-regulated transcripts in ERG fusion-positive VCaP cells. We also detected significant loss in the expression of MYC target genes with AU-15330 (Fig. 2e, Extended Data Fig. 6e, f). The global AU-15330 gene signature was highly concordant with transcriptional changes associated with ARID1A loss (Extended Data Fig. 6g). However, neither BRD7 nor BRD9 degradation alone attenuated the expression of classical AR, FOXA1 and ERG target genes or the *MYC* gene to an extent comparable to AU-15330, suggesting that canonical SWI/SNF (cBAF) complexes are the primary cofactors of oncogenic enhancer-binding transcription factors (Extended Data Fig. 6h–j). The expressions of *AR*, *MYC* and *FOXA1* genes themselves are frequently amplified in advanced prostate cancer by copy amplification and/or enhancer duplication[15,26,27]. We found that AU-15330 markedly decreased expression of *AR*, *FOXA1*, *MYC* and *TMPRSS2–ERG* transcripts to 40–60% of their baseline expression (Extended Data Fig. 7a), with parallel decreases at the protein level (Fig. 3a). More severe transcriptional attenuation of these oncogenes was noted upon BRD4 degradation by ZBC-260, with AU-15330 specifically abolishing expression of additional driver oncogenes (Extended Data Fig. 7b), again suggesting a distinct mechanism of action for AU-15330-mediated anti-tumour cytotoxicity. Similar results were observed in genetic-inactivation models (Extended Data Fig. 7c).

The hyper-expression of oncogenes like *AR*, *FOXA1* and *MYC* in cancer has been shown to be wired through looping interactions with multi-enhancer clusters[15,26,27], often referred to as super-enhancers. Several such regulatory clusters were identified in *cis*-proximity of the *AR*, *MYC* and *TMPRSS2–ERG* genes (Fig. 3b), and AU-15330 treatment led to immediate compaction of these sites and loss of H3K27Ac, AR and FOXA1 ChIP–seq signal at the super-enhancers (Fig. 3c, Extended Data Fig. 7d). To detect changes in the interaction of super-enhancers with

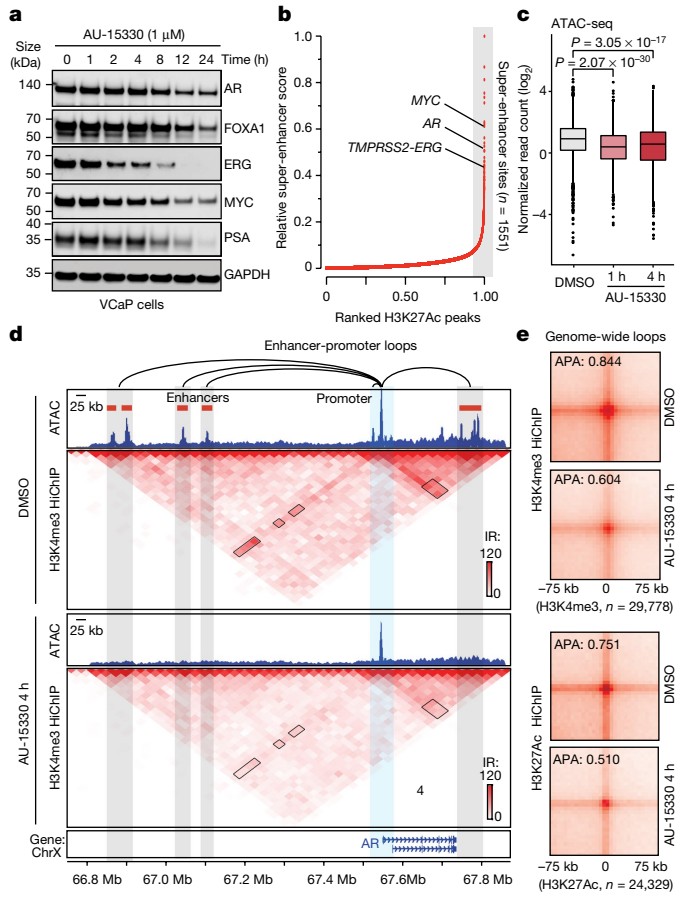

**Fig. 3 | SWI/SNF ATPase degradation disrupts enhancer–promoter loops to temper supra-physiologic expression of driver oncogenes. a**, Immunoblots of indicated proteins in VCaP cells treated with DMSO for 24 h or AU-15330 (1 µM) for increasing time durations. GAPDH is used as a loading control, and is probed on a representative immunoblot. This experiment was repeated independently twice. **b**, H3K27Ac ChIP–seq signal rank-ordered list of super-enhancers in VCaP cells with select *cis*-coded driver oncogenes denoted (HOMER). **c**, Normalized read density of ATAC-seq at super-enhancers (*n* = 32,545 sites) in VCaP cells treated with DMSO or AU-15330 (1 µM) for 1 or 4 h (two-sided *t*-test). In box plots, the centre line shows median, box edges mark quartiles 1–3, and whiskers span quartiles 1–3 ± 1.5 × interquartile range. **d**, H3K4me3 HiChIP–seq heat maps within the *AR* gene locus in VCaP cells with or without AU-15330 (1 µM) treatment for 4 h (bin size = 25 kb). ATAC-seq read-density tracks from the same treatment conditions are overlaid. Grey highlights mark enhancers; blue highlights the *AR* promoter. Loops indicate read-supported *cis* interactions within the locus. IR, interaction reads. **e**, APA plots for H3K4me3 and H3K27Ac HiChIP–seq data for all possible interactions between putative enhancers and gene promoters in VCaP cells with or without with AU-15330 treatment (1 µM, 4 h).

their target gene promoters, we performed H3K4me3 (active promoter mark) and H3K27Ac Hi-C coupled with ChIP–seq (HiChIP–seq) upon AU-15330 treatment. SWI/SNF inactivation markedly disrupted the three-dimensional looping interactions of *cis*-enhancers with the *AR* gene promoter (Fig. 3d, Extended Data Fig. 8a). Similar attenuation of enhancer–promoter interactions was detected by H3K27Ac HiChIP–seq at the *FOXA1* locus (Extended Data Fig. 8b), which is recurrently rearranged in advanced prostate cancer[15]. Aggregate peak analyses (APA) of enhancer–promoter interactions showed a marked attenuation of contact strength and/or frequency starting as early as 2 h after AU-15330 treatment, that is, within 1 h of SMARCA2 and SMARCA4 degradation (Fig. 3e, Extended Data Fig. 8c). At these early time points, we did not detect a significant decrease in H3K27Ac signal at the compacted

enhancer sites (Fig. 2d), strongly suggesting that physical chromatin accessibility and transcription factor binding serve as primary determinants of functional enhancer–promoter interactions. Of note, we found no change in the looping interactions between CTCF-bound elements (Extended Data Fig. 8d, e). Together, these data show that SWI/SNF ATPase inactivation specifically leads to genome-wide collapse of the AR, FOXA1, ERG and MYC-activated core enhancer circuitry in prostate cancer cells.

Next, we pharmacologically characterized AU-15330 in animal models of advanced prostate cancer. Notably, prolonged AU-15330 treatments showed no evident toxicity in immuno-competent mice (Extended Data Fig. 9, Supplementary Text, Supplementary Table 3). We first employed the VCaP castration-resistant prostate cancer (CRPC) model (VCaP-CRPC) to assess the efficacy of AU-15330. As expected, treatment of castrated male mice bearing the VCaP-CRPC xenografts with enzalutamide (an AR antagonist) showed moderate anti-tumour efficacy; however, treatment with AU-15330 led to potent inhibition of tumour growth, triggering disease regression in more than 20% of animals (Fig. 4a, b, Extended Data Fig. 10a, b). Treatment with the combinatorial regimen (AU-15330 plus enzalutamide) induced the most potent anti-tumour effect, with regression in all animals (Fig. 4a, b, Extended Data Fig. 10b). Tumours showed robust downregulation of SWI/SNF targets and AR, ERG, MYC and Ki67 after five days of AU-15330 treatment, both when administered alone or with enzalutamide (Fig. 4c, Extended Data Fig. 10c–e). No significant change in body weight was noted throughout any of these treatments, nor was there any histologic evidence of toxicity in essential organs at endpoint (Extended Data Fig. 10f–h). AU-15330 also strongly inhibited the growth of C4-2B cell line-derived CRPC xenografts in intact mice as a single agent and synergized with enzalutamide (Fig. 4d, Extended Data Fig. 11a–d). An in vitro evaluation of drug synergism between AU-15330 and enzalutamide confirmed synergism of the two drugs in multiple prostate cancer cell lines (Extended Data Fig. 11e–h), and pre-treatment with either drug significantly reduced the $IC_{50}$ value of the other (Extended Data Fig. 11i, j).

Treatment with AU-15330 was similarly effective in inhibiting the growth of enzalutamide-resistant cell lines, including derivatives of VCaP and LNCaP cells (Extended Data Fig. 11k–l). The combinatorial regimen also markedly inhibited tumour growth in MDA-PCa-146-12, a patient-derived xenograft (PDX) model that is inherently resistant to enzalutamide (Extended Data Fig. 12a–c). We further established a CRPC variant of the MDA-PCa-146-12 PDX by tumour implantation into castrated mice (Extended Data Fig. 12a). Even in this highly aggressive model, the combinatorial regimen induced significant tumour growth inhibition, causing regression in more than 30% of animals (Fig. 4e, Extended Data Fig. 12d). In all arms of these studies, we detected no changes in animal body weights (Extended Data Fig. 12e, f). There was also no sign of goblet cell depletion in the gastrointestinal tract (Extended Data Fig. 12g), no defect in germ cell maturation and no testicular atrophy (Extended Data Fig. 12h, i) in AU-15330-treated mice—all of which have been reported as toxicities of therapies targeted towards BET proteins[28].

## Discussion

We report AU-15330 as a novel, highly specific and VHL-dependent PRO-TAC degrader of SWI/SNF ATPase components (SMARCA2, SMARCA4 and PBRM1) that shows preferential cytotoxicity in enhancer-binding transcription factor-addicted cancers at low nanomolar concentrations. Our study identifies the SWI/SNF complex as a transcriptional dependency in AR/FOXA1-driven prostate cancer. Mechanistically, we show that complete inactivation of SWI/SNF ATPase induces a rapid, near-complete and targeted loss of chromatin accessibility at the core-enhancer circuitry of AR, FOXA1, MYC and ERG, thereby attenuating their cancer-promoting transcriptional programs and tempering

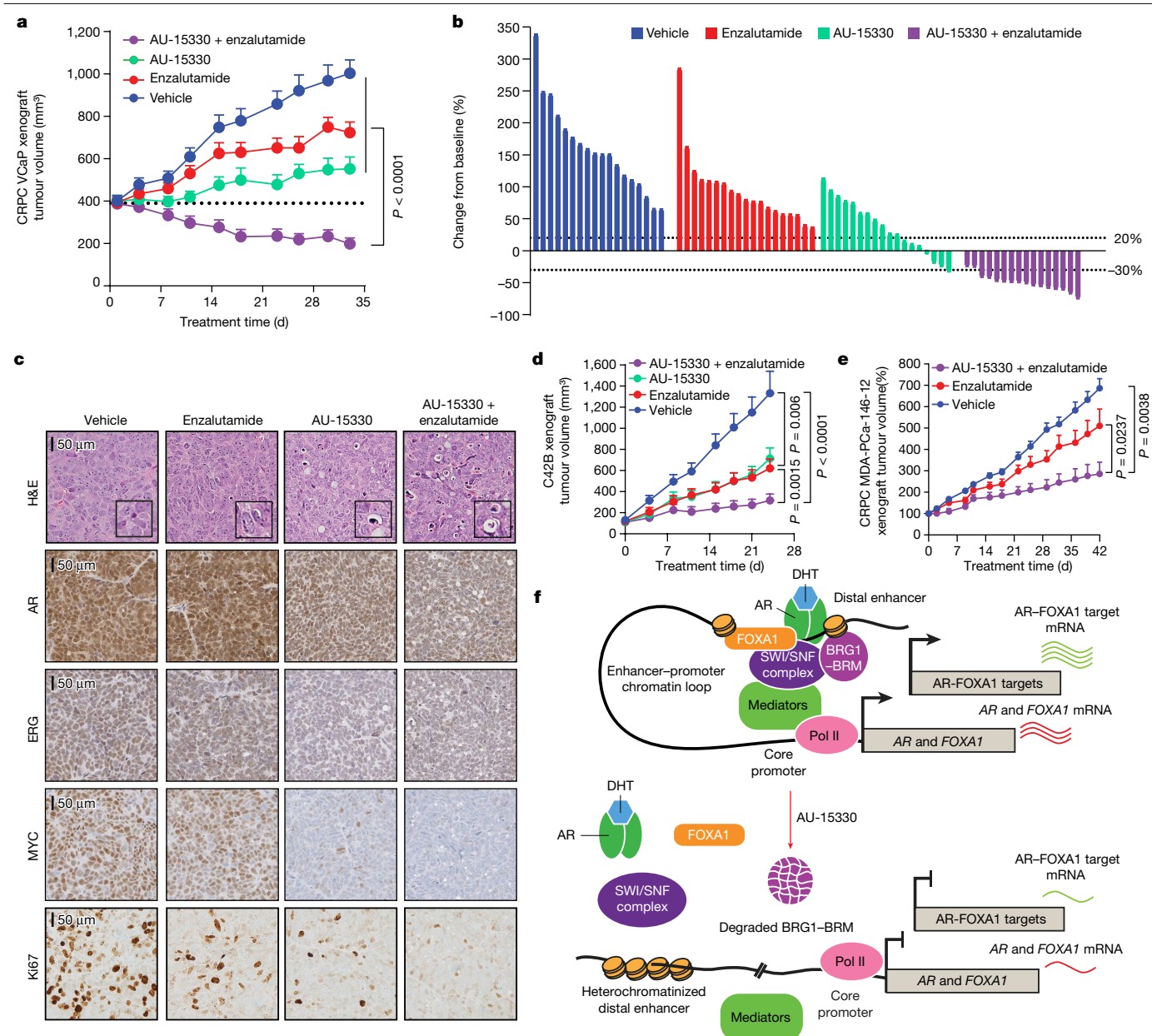

**Fig. 4 | AU-15330 inhibits tumour growth in preclinical models of CRPC and synergizes with enzalutamide. a,** Tumour volume (measured twice per week using callipers) in the VCaP-CRPC model with AU-15330 alone or in combination with enzalutamide (two-sided *t*-test). Data are mean ±s.e.m. (vehicle: *n* = 18; AU-15330: *n* = 20; enzalutamide: *n* = 18; AU-15330 + enzalutamide: *n* = 16). **b,** Waterfall plot depicting change in tumour volume after 33 days of treatment. Response evaluation criteria in solid tumours (RECIST) was used to stratify tumours: progressive disease (PD), at least a 20% increase in tumour size; stable disease (SD), increase of <20% to a decrease of <30%; partial response (PR), at least a 30% decrease. The vehicle and enzalutamide groups have 100% PD; the AU-15330 group has 61% PD, 33% SD and 6% PR; and the AU-15330 + enzalutamide group has 0% PD, 12% SD and 88% PR. **c,** Representative haematoxylin and eosin (H&E) staining and immunohistochemistry from the

VCaP-CRPC xenograft study (*n* = 2 tumours per condition). Insets in the H&E images show expanded views of apoptotic cells. **d,** Tumour volume measurements showing efficacy of AU-15330, enzalutamide or combined treatment in C4-2B-derived CRPC xenografts (*n* = 20 per condition; two-sided *t*-test). Data are mean ± s.e.m. **e,** Tumour volume measurements showing the effect of enzalutamide alone or in combination with AU-15330 in the castration-resistant MDA-PCa-146-12 PDX study (two-sided *t*-test). Data are mean ± s.e.m. **f,** Mechanism of action of AU-15330-triggered cytotoxicity in AR–FOXA1-signalling-driven prostate cancer. SWI/SNF ATPase degradation induces a rapid, targeted loss in chromatin accessibility at the core-enhancer circuitry of AR, FOXA1, ERG and MYC, thereby attenuating their cancer-promoting transcriptional programs and tempering the enhancer-wired supra-physiologic expression of driver oncogenes.

the enhancer-wired supra-physiologic expression of driver oncogenes (Fig. 4f). These findings are in line with those from recent studies that have used chemical and/or genetic approaches to show that continuous SWI/SNF-remodelling activity is needed to retain enhancers in an open, nucleosome-free conformation[29,30]. To our knowledge, this is the first study to demonstrate that physical chromatin accessibility

can be modulated at non-coding regulatory elements as a novel therapeutic strategy in cancer treatment. Thus, recently developed SWI/SNF ATPase inhibitors and degraders add to the growing arsenal of chromatin-targeted therapeutics for directly combating enhancer addiction in human cancers, warranting assessments of their their safety and efficacy in clinical trials.

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

## Methods

### Cell lines, antibodies, and compounds

Most cell lines were originally obtained from ATCC, DSMZ, ECACC, or internal stock. C4-2B cells were provided by E. Keller (University of Michigan). CWR-R1 cells and a series of enzalutamide-resistant prostate cancer cell lines (LNCaP_Parental, LNCaP_EnzR, CWR-R1_Parental, CWR-R1_EnzR, VCaP_Parental and VCaP_EnzR) were provided by D. Vander Griend (University of Illinois at Chicago)[31]. Bin-67 was generously provided by B. Vanderhyden (Ottawa Hospital Research Institute). All cells were genotyped to confirm their identity at the University of Michigan Sequencing Core and tested routinely for Mycoplasma contamination. LNCaP, 22RV-1, CWR-R1, PC-3, and DU145 were grown in Gibco RPMI-1640 + 10% FBS (ThermoFisher). VCaP was grown in Gibco DMEM + 10% FBS (ThermoFisher). BIN-67 cell lines were grown in custom media (20% fetal bovine serum (FBS), 40% Dulbecco's modified Eagle's medium, 40% Dulbecco's modified Eagle's medium/Ham's F12). Sources of all antibodies are described in Supplementary Table 2. AU-15330 was synthesized by Aurigene (see Supplementary Text), dBRD9 and VZ 185 were purchased from Tocris Bioscience, and enzalutamide was purchased from Selleck Chemicals.

### Computational modelling of AU-15330 - SMARCA2-BD binding

The binding model of AU-15330 in complex with SMARCA2-BD and VHL was generated using Aurigene's proprietary computing algorithm ALMOND (algorithm for modeling neosubstrate degraders). The algorithm is developed using the ICM-Pro integrated modelling platform (http://www.molsoft.com/icm_pro.html) and trained to predict models of ternary complexes of bi-functional molecules with very short or no linkers. The process employs protein-protein docking simulation, exhaustive conformational sampling, small molecule-protein docking, and site-directed scoring of predicted ternary complex models. The computed score estimates the force of induced interactions in the predicted target–E3 ligase complex and is used as a basis for prioritization of degrader binding models. The images were prepared using PyMOL (https://www.schrodinger.com/products/pymol).

### Cell viability assay

Cells were plated onto 96-well plates in their respective culture medium and incubated at 37 °C in an atmosphere of 5% $CO_2$. After overnight incubation, a serial dilution of compounds was prepared and added to the plate. The cells were further incubated for 5 days, and the CellTiter-Glo assay (Promega) was then performed according to the manufacturer's instruction to determine cell proliferation. The luminescence signal from each well was acquired using the Infinite M1000 Pro plate reader (Tecan), and the data were analyzed using GraphPad Prism software (GraphPad Software).

### Incucyte proliferation assays/Caspase-3/7 green apoptosis assay

A total of 4,000 cells per well were seeded in clear 96-well plates. After overnight incubation, compounds were added to the cells at logarithmic dose series. One day and 8 days after seeding, cellular ATP content was measured using CellTiterGlo (Promega). Measurements after 8 days were divided by the measurement after 1 day (that is, the T0 plate) to derive fold proliferation. For online analysis of cell growth, 4,000 cells per well were seeded in clear 96-well plates (Costar no. 3513). IncuCyte Caspase-3/7 Green Apoptosis Assay Reagent (1:1,000, Essen BioSciences no. 4440) was added, and cells were incubated at 37 °C and 5% $CO_2$ overnight. On the next day, compounds were added at the desired concentration using the HP digital dispenser D300, and plates were read in an Incucyte ZOOM. Every 2h, phase object confluence (percentage area) for proliferation and green object count for apoptosis were measured. Values for apoptosis were normalized for the total number of cells.

### Western blot and immunoprecipitation

Cell lysates were prepared in RIPA buffers (ThermoFisher Scientific) supplemented with cOmpleteTM protease inhibitor cocktail tablets (Sigma-Aldrich), and total protein was measured by Pierce BCA Protein Assay Kit (ThermoFisher Scientific). An equal amount of protein was resolved in NuPAGE 3 to 8%, Tris-Acetate Protein Gel (ThermoFisher Scientific) or NuPAGE 4 to 12%, Bis-Tris Protein Gel (ThermoFisher Scientific) and blotted with primary antibodies. Following incubation with HRP-conjugated secondary antibodies, membranes were imaged on an Odyssey CLx Imager (LiCOR Biosciences). Immunoprecipitations were performed in LNCaP and VCaP cells treated as described. 600 μg of nuclear extracts isolated using the NE-PER Nuclear and Cytoplasmic Extraction Reagents (ThermoFisher Scientific) were immunoprecipitated with SMARCC1, AR, FOXA1, or ERG antibodies according to the manufacturer's protocol. Eluted proteins were subjected to western blot or mass spectrometry analysis. For all immunoblots, uncropped and unprocessed images are provided in Supplementary Figure 1.

### RNA isolation and quantitative real-time PCR

Total RNA was isolated from cells using the Direct-zol kit (Zymo), and cDNA was synthesized from 1,000 ng total RNA using Maxima First Strand cDNA Synthesis Kit for PCR with reverse transcription (RT–PCR) (Thermo Fisher Scientific). Quantitative real-time PCR (qPCR) was performed in triplicate using standard SYBR green reagents and protocols on a QuantStudio 5 Real-Time PCR system (Applied Biosystems). The target mRNA expression was quantified using the ΔΔCt method and normalized to *ACTB* expression. All primers were designed using Primer 3 (http://frodo.wi.mit.edu/primer3/) and synthesized by Integrated DNA Technologies. Primer sequences are listed in Supplementary Table 2.

### CRISPR knock-out and inducible shRNA knockdown

Guide RNAs (sgRNAs) targeting the exons of human *SMARCA2/BRM* or *SMARCA4/BRG1* were designed using Benchling (https://www.benchling.com/). Non-targeting sgRNA, *SMARCA2/BRM* or *SMARCA4/BRG1*-targeting sgRNAs were cloned into lentiCRISPR v2 plasmid according to published literature[32]; lentiCRISPR v2 plasmid was a gift from F. Zhang (Addgene plasmid #52961). LNCaP cells were transiently transfected with lentiCRISPR v2 encoding non-targeting or pool of three independent *SMARCA2/BRM* or *SMARCA4/BRG1*-targeting sgRNAs. Twenty-four hours after transfection, cells were selected with 1 μg ml$^{-1}$ puromycin for three days. Western blot was performed to examine the knock-out efficiency. The sgRNA sequences are listed in Supplementary Table 2.

### ATAC-seq and analysis

ATAC-seq was performed as previously described[33]. In brief, 50,000 cancer cells treated with AU-15330 or ZBC-260[30] were washed in cold PBS and resuspended in cytoplasmic lysis buffer (CER-I from the NE-PER kit, Invitrogen, cat. no. 78833). This single-cell suspension was incubated on ice for 5–8 min (depending on the cell line) with gentle mixing by pipetting every 2 min. The lysate was centrifuged at 1,300*g* for 5 min at 4 °C. Nuclei were resuspended in 50 μl of 1× TD buffer, then incubated with 2–2.5 μl Tn5 enzyme for 30 min at 37 °C (Nextera DNA Library Preparation Kit; cat. no. FC-121-1031). Samples were immediately purified by Qiagen minElute column and PCR-amplified with the NEB-Next High-Fidelity 2X PCR Master Mix (NEB; cat. no. M0541L) following the original protocol[33]. qPCR was used to determine the optimal PCR cycles to prevent over-amplification. The amplified library was further purified by Qiagen minElute column and SPRI beads (Beckman Coulter; cat. no. A63881). ATAC-seq libraries were sequenced on the Illumina HiSeq 2500 (125-nucleotide read length, paired end).

Paired-end .fastq files were trimmed and uniquely aligned to the GRCh38/hg38 human genome assembly using Novoalign (Novocraft) (with the parameters -r None -k -q 13 -k -t 60 -o sam –a CTGTCTCTTATA-CACATCT), and converted to .bam files using SAMtools (version 1.3.1).

Reads mapped to mitochondrial or duplicated reads were removed by SAMtools and PICARD MarkDuplicates (version 2.9.0), respectively. Filtered .bam files from replicates were merged for downstream analysis. MACS2 (2.1.1.20160309) was used to call ATAC-seq peaks. The coverage tracks were generated using the program bam2wig (http://search.cpan.org/dist/Bio-ToolBox/) with the following parameters: –pe–rpm–span–bw. Bigwig files were then visualized using the IGV (Broad Institute) open-source genome browser, and the final figures were assembled using Adobe Illustrator.

### De novo and known motif enrichment analysis
All de novo and known motif enrichment analyses were performed using the HOMER (v.4.10) suite of algorithms[43]. Peaks were called by the findPeaks function (-style factor -o auto) at 0.1% false discovery rate; de novo motif discovery and enrichment analysis of known motifs were performed with findMotifsGenome.pl (–size given–mask). The top 10 motifs from the results are shown, and motifs were generally ascribed to the protein family instead of specific family members (unless known).

### RNA-seq and analysis
RNA-seq libraries were prepared using 200–1,000 ng of total RNA. PolyA+ RNA isolation, cDNA synthesis, end-repair, A-base addition, and ligation of the Illumina indexed adapters were performed according to the TruSeq RNA protocol (Illumina). Libraries were size selected for 250–300 bp cDNA fragments on a 3% Nusieve 3:1 (Lonza) gel, recovered using QIAEX II reagents (QIAGEN), and PCR amplified using Phusion DNA polymerase (New England Biolabs). Library quality was measured on an Agilent 2100 Bioanalyzer for product size and concentration. Paired-end libraries were sequenced with the Illumina HiSeq 2500, ($2 \times 100$ nucleotide read length) with sequence coverage to 15–20M paired reads.

Libraries passing quality control were trimmed of sequencing adaptors and aligned to the human reference genome, GRCh38. Samples were demultiplexed into paired-end reads using Illumina's bcl2fastq conversion software v2.20. The reference genome was indexed using bowtie2-build, and reads were aligned onto the GRCh38/hg38 human reference genome using TopHat2[34] with strand-specificity and allowing only for the best match for each read. The aligned file was used to calculate strand-specific read count for each gene using HTSeq-count (version 0.13.5)[35]. EdgeR (version 3.34.1)[36] was used to compute differential gene expression using raw read-counts as input. Heatmaps were generated using the ComplexHeatmap[37] package in R. For gene enrichment analysis (GSEA), we first defined ERG and FOXA1 gene signatures from VCaP or LNCaP cells treated with control siRNA or siRNA targeting *ERG*[38] or *FOXA1* (generated in this study) containing 250 significantly downregulated genes. For AR and MYC, the Hallmark gene signatures were used. These gene signatures were used to perform a fast pre-ranked GSEA using fgsea bioconductor package[39] in R. We used the function fgsea to estimate the net enrichment score and p-value of each pathway, and the plotEnrichment function was used to plot enrichment for the pathways of interest.

### ChIP–seq and data analysis
Chromatin immunoprecipitation experiments were carried out using the HighCell# ChIP-Protein G kit (Diagenode) as per the manufacturer's protocol. Chromatin from $5 \times 10^6$ cells was used for each ChIP reaction with 10 µg of the target protein antibody. In brief, cells were trypsinized and washed twice with 1× PBS, followed by cross-linking for 8 min in 1% formaldehyde solution. Crosslinking was terminated by the addition of 1/10 volume 1.25 M glycine for 5 min at room temperature followed by cell lysis and sonication (Bioruptor, Diagenode), resulting in an average chromatin fragment size of 200 bp. Fragmented chromatin was then used for immunoprecipitation using various antibodies, with overnight incubation at 4 °C. ChIP DNA was de-crosslinked and

purified using the iPure Kit V2 (Diagenode) using the standard protocol. Purified DNA was then prepared for sequencing as per the manufacturer's instructions (Illumina). ChIP samples (1–10 ng) were converted to blunt-ended fragments using T4 DNA polymerase, *Escherichia coli* DNA polymerase I large fragment (Klenow polymerase), and T4 polynucleotide kinase (New England BioLabs (NEB)). A single adenine base was added to fragment ends by Klenow fragment (3′ to 5′ exo minus; NEB), followed by ligation of Illumina adaptors (Quick ligase, NEB). The adaptor-ligated DNA fragments were enriched by PCR using the Illumina Barcode primers and Phusion DNA polymerase (NEB). PCR products were size-selected using 3% NuSieve agarose gels (Lonza) followed by gel extraction using QIAEX II reagents (Qiagen). Libraries were quantified and quality checked using the Bioanalyzer 2100 (Agilent) and sequenced on the Illumina HiSeq 2500 Sequencer (125-nucleotide read length).

Paired-end, 125 bp reads were trimmed and aligned to the human reference genome (GRC h38/hg38) with the Burrows-Wheeler Aligner (BWA; version 0.7.17-r1198-dirty)[40]. The SAM file obtained after alignment was converted into BAM format using SAMTools (version 1.9). MACS2 (version 2.1.1.20160309) callpeak was used for performing peak calling with the following option: 'macs2 callpeak–call-summits–verbose 3 -g hs -f BAM -n OUT–qvalue 0.05'. For H3K27ac data, the broad option was used. Using deepTools (version 3.3.1) bamCoverage, a coverage file (bigWig format) for each sample was created. The coverage was calculated as the number of reads per bin, where bins are short consecutive counting windows. While creating the coverage file, the data was normalized with respect to each library size. ChIP peak profile plots and read-density heat maps were generated using deepTools, and cistrome overlap analyses were carried out using the ChIPpeakAnno (version 3.0.0) or ChIPseeker (version 1.29.1) packages in R (version 3.6.0).

### HiChIP library preparation and data analysis
HiChIP assay was performed on 5x10^6 DMSO or AU-15330 treated VCaP cells. Frozen cells were resuspended in 1× PBS and crosslinked with 3 mM DSG and 1% formaldehyde. Washed cells were digested with 0.5 µl MNase in 100 µl of nuclease digest buffer with MgCl$_2$. Cells were lysed with 1× RIPA, and clarified lysate (approximately 1,400 ng) was used for ChIP. The antibody amount used per ChIP and vendor information are as follows: CTCF: 1.14 µg of Cell Signaling cat. no. 3418; H3K4me3: 3.4 µg of Cell Signaling cat. no. 9751; H3K27ac: 0.4 µg of Cell Signaling cat. no. 8173. The Protein A/G bead pulldown, proximity ligation, and libraries were prepared as described in the Dovetail protocol (Dovetail HiChIP MNase Kit). Libraries were sequenced on an Illumina HiSeq 4000.

Raw fastq files were aligned using BWA mem (version 0.7.17-r1198-dirty) with the –5SP options with an index containing only the main chromosome from the human genome release hg38 (available from the UCSC genome). The aligned paired reads were annotated with pairtools (version 0.3.0) parse (https://github.com/open2c/pairtools) with the following options–min-mapq 40–walks-policy 5unique–max-inter-align-gap 30 and the–chroms-path file corresponding to the size of the chromosome used for the alignment index. The paired reads were further processed to remove duplicated reads, sorted with unaligned reads removed with the pairtools sort and the pairtools dedup tools with the basic option to produce an alignment file in the bam format as well as the location of the valid pair. The valid pairs were finally converted to the .cool and .mcool format using the cooler cload and cooler zoomify tools (version 0.8.11)[41] and to the .hic format using the juicer tool (version 1.22.01)[42].

For the generation of the aggregate peak analyses (APA) plots, we used the HiCExplorer tools (version 3.7) and the hicAggregateContacts command with–range 50000:100000–numberOfBins 30. Plots for all chromosomes were individually computed and summated to generate the global APA plots. The ComplexHeatmaps package[37] in R was used for the generation of the final heatmap. For the Hi-ChIP contact heatmap, .hic files were uploaded to the WashU Epigenome Browser

(https://epigenomegateway.wustl.edu/), and screenshots from gene loci of interest were downloaded using the default viewing conditions.

## Super-enhancer analysis

Super-enhancer regions were identified with findPeaks function from HOMER (version v.4.10)[43] using options "-style super -o auto". In addition, the option "-superSlope −1000" was added to include all potential peaks, which were used to generate the super-enhancer plot (super-enhancer score versus ranked peaks). The slope value of greater than or equal to 1 was used to identify super-enhancer clusters. The input files to findPeaks were tag directories generated from alignment files in SAM format with makeTagDirectory function from HOMER.

## AU-15330 and enzalutamide formula for *in vivo* studies

AU-15330 was added in 40% of 2-hydroxypropyl-β-cyclodextrin (HPβCD) and sonicated until completely dissolved, and then the solution was further mixed with 5% dextrose in water (D5W) to reach a final concentration of 10% HPβCD. AU-15330 was freshly prepared right before administration to mice. AU-15330 was delivered to mice by intravenous injection either through the tail vein or retro-orbital injection unless otherwise indicated. Enzalutamide was added in 1% carboxymethyl cellulose (CMC) with 0.25% Tween-80 and sonicated until homogenized. Enzalutamide was delivered to mice by oral gavage.

## Human prostate tumour xenograft models

Six-week-old male CB17 severe combined immunodeficiency (SCID) mice were procured from the University of Michigan breeding colony. Subcutaneous tumours were established at both sides of the dorsal flank of mice. Tumours were measured at least biweekly using digital calipers following the formula $(\pi/6) (L \times W^2)$, where $L$ is length and $W$ is width of the tumour. At the end of the studies, mice were killed and tumours extracted and weighed. The University of Michigan Institutional Animal Care and Use Committee (IACUC) approved all in vivo studies.

For the VCaP non-castrated tumour model, $3 \times 10^6$ VCaP cells were injected subcutaneously into the dorsal flank on both sides of the mice in a serum-free medium with 50% Matrigel (BD Biosciences). Once tumours reached a palpable stage (~200 mm³), mice were randomized and treated with either 10, 30 mg kg⁻¹ AU-15330, or vehicle through intravenous injection 5 days per week for 3 weeks.

For the VCaP castration-resistant tumour model, $3 \times 10^6$ VCaP cells were injected subcutaneously into the dorsal flank on both sides of the mice in a serum-free medium with 50% Matrigel (BD Biosciences). Once tumours reached a palpable stage (~200 mm³), tumour-bearing mice were castrated. Once tumours grew back to the pre-castration size, mice were randomized and treated with either 60 mg kg⁻¹ AU-15330 or vehicle by intranvenous injection 3 days per week, and with or without 10 mg kg⁻¹ enzalutamide by oral gavage 5 days per week for 5 weeks.

For the C4-2B non-castrated tumour model, $1 \times 10^6$ cells were injected subcutaneously into the dorsal flank on both sides of the mice in a serum-free medium with 50% Matrigel (BD Biosciences). Once tumours reached a palpable stage (~100 mm³), mice were randomized and treated with either 60 mg kg⁻¹ AU-15330 or vehicle by intravenous injection 3 days per week, and with or without 30 mg kg⁻¹ enzalutamide by oral gavage 5 days per week for 4 weeks. Following the IACUC guidelines, in all treatment arms the maximal tumour size did not exceed the 2.0 cm limit in any dimension and animals with xenografts reaching that size were duly euthanized. The raw tumour volumes and/or weights from all animal efficacy studies are included in the Source Data files.

## Prostate patient-derived xenograft models

The University of Texas M. D. Anderson Cancer Center PDX series has been previously described[44]. PDXs were derived from men with CRPC undergoing cystoprostatectomy using described protocols. MDA-PCa-146-12 was derived from a CRPC patient diagnosed with Gleason 5+4=9 prostate adenocarcinoma. MDA-PCa-146-12 was derived from a specimen obtained from the left bladder wall and demonstrated conventional adenocarcinoma (AR⁺). PDXs were maintained in male SCID mice by surgically implanting 2 mm³ tumours coated with 100% Matrigel to both flanks of mice. Once tumours reached ~200 mm³ in size, mice were randomized and divided into different treatment groups receiving either 60 mg kg⁻¹ AU-15330 or vehicle by subcutaneous injection 3 days per week, and with or without 10 mg/kg enzalutamide by oral gavage 5 days per week for 3 weeks. For castration-resistant MDA-PCa-146-12, tumours were established on castrated male SCID mice. Once tumours reached ~100 mm³, mice were randomized and divided into different treatment groups receiving either 60 mg kg⁻¹ AU-15330 or vehicle by intravenous injection 3 days per week, and with or without 30 mg kg⁻¹ enzalutamide by oral gavage 5 days per week for 6 weeks. Following the IACUC guidelines, in all treatment arms the maximal tumour size did not exceed the 2.0 cm limit in any dimension and animals with xenografts reaching that size were duly euthanized. The raw tumour volumes and/or weights from all animal efficacy studies are included in the Source Data files.

## Histopathological analysis of organs harvested for drug toxicity

For the present study, organs (liver, spleen, kidney, colon, small intestine, prostate, and testis) were harvested and fixed in 10% neutral buffered formalin followed by embedding in paraffin to make tissue blocks. These blocks were sectioned at 4 µm and stained with Harris haematoxylin and alcoholic eosin-Y stain (both reagents from Leica Surgipath) and staining was performed on Leica autostainer-XL (automatic) platform. The stained sections were evaluated by two different pathologists using a brightfield microscope in a blinded fashion between the control and treatment groups for general tissue morphology and coherence of architecture. A detailed comprehensive analysis of the changes noted at the cellular and sub-cellular level were performed as described below for each specific tissue.

**Evaluation of liver.** Liver tissue sections were evaluated for normal architecture, and regional analysis for all three zones was performed for inflammation, necrosis, and fibrosis.

**Evaluation of spleen.** Splenic tissue sections were evaluated for the organization of hematogenous red and lymphoid white pulp regions including necrosis and fibrotic changes if any.

**Evaluation of kidney.** Kidney tissue sections were examined for changes noted if any in all the four renal functional components, namely glomeruli, interstitium, tubules, and vessels.

**Evaluation of colon.** Colonic tissue sections were examined for mucosal (epithelium and lamina propria), sub-mucosal, and seromuscular layer changes including crypt changes, goblet cells, inflammatory infiltrate granulation tissue, and mucosal ulceration. A detailed goblet cell evaluation was also performed utilizing Alcian blue staining wherein goblet cells and epithelial cells were counted in ten colonic crypt epithelia in each experimental animal of the various subgroups. Summation of all the goblet and epithelial cells was done, and a ratio of goblet cell to epithelial cell (GC ratio) was calculated per sample.

**Evaluation of small intestine.** Small intestine tissue sections were examined for mucosal changes such as villous blunting, villous: crypt ratio, and evaluated for inflammatory changes including intraepithelial lymphocytes, extent (mucosal, sub-mucosal, serosal), and type of inflammatory infiltrate including tissue modulatory effect.

**Evaluation of prostate.** Prostate tissue sections were evaluated to note for any epithelial abnormality and stromal changes identified in all four lobes (dorsal, anterior, lateral, and ventral). Additionally, any overt inflammatory infiltrate was also examined.

**Evaluation of testis.** Testicular tissues were examined for the architectural assessment of seminiferous tubules (orderly maturation of germinal epithelial cells devoid of maturation arrest and Sertoli cell prominence), Leydig cells, and interstitial reaction. For an in-depth comprehensive analysis to comment upon the spermatogenesis in a semi-quantitative method, a testicular biopsy score count (Johnsen score) in 100 orderly cross-sections of seminiferous tubules in each animal of all the subgroups at 20× magnification was performed. Each of the 100 seminiferous tubules assessed was given a score (score range: 0–10), and the average score was calculated (total sum of score/100).

## Alcian blue staining

Alcian blue staining was performed as per the manufacturer's protocol (Alcian Blue Stain Kit (pH 2.5) cat. no. ab150662). Following an overnight incubation of tissue sections at 58 °C, slides were deparaffinized in xylene followed by hydration in ethanol (100%, 70%) and water for 5 min each. Slides were then incubated in acetic acid solution for 3 min followed by a 30 min incubation at room temperature in Alcian blue stain (pH 2.5). Excess Alcian blue was removed by rinsing slides in acetic acid solution for 1 min, and three water washes for 2 min each. Nuclear Fast Red solution was used as a counterstain for 5 min. Slides were subsequently washed in running tap water, dehydrated in ethanol, xylene, and mounted using EcoMount (Thermo Fisher, cat. no. EM897L).

## Immunohistochemistry

Immunohistochemistry was performed on formalin-fixed paraffin-embedded 4μm sections of mouse or xenograft tissues. Slides with tissue sections were incubated at 58 °C overnight and the next day were deparaffinized in xylene, followed by serial hydration steps in ethanol (100%, 70%) and water for 5 min each. Endogenous tissue peroxidase activity was blocked by placing slides in 3% $H_2O_2$-methanol solution for 1 h at room temperature. Antigen retrieval was performed by microwaving slides in a solution of citrate buffer (pH 6) for 15 min, followed by blocking in 2.5% normal horse serum (Vector Laboratories, cat. no. S-2012-50) for 2 h. The slides were then incubated in the following primary antibodies overnight at 4 °C: BRG1 (Abcam cat. no. 108318, 1:100), AR (Millipore cat. no. 06-680, 1:2,000), BRM1 (Millipore Sigma cat. no. HPA029981, 1:100), FOXA1 (Thermo Fisher Scientific cat. no. PA5-27157, 1:1,000), ERG (Cell Signaling Technology cat. no. 97249S, 1:500). ImmPRESS-HRP conjugated anti-mouse–anti-rabbit cocktail from Vector Laboratories (cat. no. MP-7500-50) was used as secondary antibodies (room temperature, 1 h). Visualization of staining was done per the manufacturer's protocol (Vector Laboratories, cat. no. SK-4100). Following DAB staining, slides were dehydrated in ethanol, xylene (5 min each), and mounted using EcoMount (Thermo Fisher, cat. no. EM897L).

## TMT mass spectrometry

VCaP cells were seeded at $5 \times 10^6$ cells on a 100 mm plate 24 h before treatment. Cells were treated in triplicate by the addition of test compounds. After 4 h, the cells were harvested and processed by using EasyPep Mini MS Sample Prep Kit (Thermo Fisher, A40006). Samples were quantified using a micro BCA protein assay kit (Thermo Fisher Scientific) and cell lysates were proteolyzed and labelled with TMT 10-plex Isobaric Label Reagent (Thermo Fisher Scientific, 90110) essentially following the manufacturer's protocol. Briefly, upon reduction and alkylation of cysteines, the proteins were precipitated by adding 6 volumes of ice-cold acetone followed by overnight incubation at 20 °C. The precipitate was spun down, and the pellet was allowed to air dry. The pellet was resuspended in 0.1M TEAB and digested overnight with trypsin (1:50 enzyme:protein) at 37 °C with constant mixing using a thermomixer. The TMT 10-plex reagents were dissolved in 41 ml of anhydrous acetonitrile, and labelling was performed by transferring the entire digest to the TMT reagent vial and incubating it at room temperature for 1 h. The reaction was quenched by adding 8 ml of 5% hydroxylamine and a further 15 min incubation. Labelled samples were mixed together and dried using a vacufuge. An offline fractionation of the combined sample (200 mg) into 10 fractions was performed using high pH reversed-phase peptide fractionation kit according to the manufacturer's protocol (Pierce, 84868). Fractions were dried and reconstituted in 12 ml of 0.1% formic acid/2% acetonitrile in preparation for LC–MS/MS analysis.

To obtain superior accuracy in quantitation, we employed multinotch-MS3[45] which minimizes the reporter ion ratio distortion resulting from fragmentation of co-isolated peptides during MS analysis. Orbitrap Fusion (Thermo Fisher Scientific) and RSLC Ultimate 3000 nano-UPLC (Dionex) was used to acquire the data. The sample (2 ml) was resolved on a PepMap RSLC C18 column (75 mm i.d. × 50 cm; Thermo Scientific) at the flowrate of 300 nl min$^{-1}$ using 0.1% formic acid/acetonitrile gradient system (2–22% acetonitrile in 150 min; 22–32% acetonitrile in 40 min; 20 min wash at 90% followed by 50 min re-equilibration) and direct spray into the mass spectrometer using EasySpray source (Thermo Fisher Scientific). The mass spectrometer was set to collect one MS1 scan (Orbitrap; 60K resolution; AGC target $2 \times 10^5$; max IT 100 ms) followed by data-dependent, "Top Speed" (3 s) MS2 scans (collision-induced dissociation; ion trap; NCD 35; AGC $5 \times 10^3$; max IT 100 ms). For multinotch-MS3, top 10 precursors from each MS2 were fragmented by HCD followed by Orbitrap analysis (NCE 55; 60K resolution; AGC $5 \times 10^4$; max IT 120 ms, 100-500 $m/z$ scan range). Proteome Discoverer (v2.1; Thermo Fisher) was used for data analysis. MS2 spectra were searched against SwissProt human protein database (release 11 November 2015; 42,084 sequences) using the following search parameters: MS1 and MS2 tolerances were set to 10 ppm and 0.6 Da, respectively; carbamidomethylation of cysteines (57.02146 Da) and TMT labelling of lysine and N-termini of peptides (229.16293 Da) were considered static modifications; oxidation of methionine (15.9949 Da) and deamidation of asparagine and glutamine (0.98401 Da) were considered variable. Identified proteins and peptides were filtered to retain only those that passed FDR threshold. Quantitation was performed using high-quality MS3 spectra (Average signal-to-noise ratio of 20 and <30% isolation interference).

**Meta-analyses of protein interactomes.** Interactome proteomics data of AR and ERG was downloaded from published literature[38,46]. The FOXA1 nuclear co-immunoprecipitation/mass spectrometry experiment was performed in this study as described above. The protein interactomes of AR, ERG, and FOXA1 were ranked based on FDR at the top 10%, and the intersection was taken from these three independent studies.

## Assessment of drug synergism

To determine the presence of synergy between two drug treatments, cells were treated with increasing concentrations of either drug for 120 h, followed by the determination of viable cells using the CellTiter-Glo Luminescent Cell Viability Assay (Promega). The experiment was carried out in four biological replicates. The data were expressed as percentage inhibition relative to baseline, and the presence of synergy was determined by the Bliss method using the synergy finder R package[47].

## Reporting summary

Further information on research design is available in the Nature Research Reporting Summary linked to this paper.

## Data availability

All raw next-generation sequencing, ATAC, ChIP, RNA, and HiChIP–seq data generated in this study have been deposited in the Gene Expression Omnibus (GEO) repository at NCBI under accession code GSE171592. Source data are provided with this paper.

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

**Acknowledgements** We thank M. Cieslik, E. Young, Y. Cheng and C. Wang from the Michigan Center for Translational Pathology at the University of Michigan, V. Basrur and the Rogel Cancer Center Proteomics Shared Resource, S. Chelur and A. Kumar from Aurigene Discovery Technologies, and T. Dickinson and E. Schulak from Dovetail Genomics for insightful discussions, sharing experimental protocols, and providing technical assistance; S. Wang for providing the BRD4 degrader ZBC260; S. Ellison for editing and proofreading the manuscript; and J. Athanikar for helping with the journal submission. This work was supported by the following mechanisms: Prostate Cancer Foundation (PCF), Prostate Specialized Programs of Research Excellence Grant P50-CA186786, National Cancer Institute Outstanding Investigator Award R35-CA231996, the Early Detection Research Network U01-CA214170, National Cancer Institute P30-CA046592, and the 2020 Movember Distinguished Gentleman's Ride PCF Challenge Award. L.X. is supported by a Department of Defense Prostate Cancer Research Program Idea Development Award (W81XWH-21-1-0500). A.P. is supported by the NIH/NCI F99/K00 pre-doctoral to post-doctoral transition fellowship and the PCF Young Investigator Award. A.M.C. is a Howard Hughes Medical Institute Investigator, A. Alfred Taubman Scholar, and American Cancer Society Professor.

**Author contributions** L.X., A.P., Y.Q. and A.M.C. conceived and designed the studies; L.X. and A.P. performed all of the in vitro and functional genomics experiments with assistance from S.E., S.E.C., H.Z., X.W., S.K., I.J.A. and M.J.; Y.Q. performed all of the animal efficacy studies with assistance from S.A.S. and A.D.D.; A.P. and P.B. carried out all of the bioinformatics analyses with assistance from Y.Z. and J.N.V.; R. Mannan and R. Mehra carried out all of the histopathological evaluations of drug toxicity and quantified all of the histology-based data; S.E. and S.Z.-W. carried out all of the immunohistochemistry with L.M. helping with tissue processing and cross-sectioning. M.S.B., J.G. and M.B. helped with the HiChIP–seq experiment and data analyses. Y.C. helped with modelling drug–protein interaction. F.S. and R.W. generated next-generation sequencing libraries, and X.C. performed the sequencing. S. Sasmal. L.K., S.M., C.A., S. Samajdar, K.A. and M.R. were involved in the discovery, synthesis and initial profiling of the AU-15330 compound. N.M.N., U.V. and Y.W. provided various key preclinical and clinical resources. A.I.N. guided all of the proteomics analyses. L.X., A.P. and A.M.C. wrote the manuscript and organized the final figures.

**Competing interests** S. Sasmal., L.K., S.M., C.A., S. Samajdar, K.A. and M.R. are affiliated with Aurigene Discovery Technologies, which is a clinical-stage biotech company with working sites in Bangalore, India and Kuala Lumpur, Malaysia. J.G., M.S.B. and M.B. are affiliated with Dovetail Genomics, which is an early-stage Santa Cruz-based start-up company developing cutting-edge genomics technologies. A.M.C. is a co-founder and serves on the scientific advisory boards of LynxDx, Oncopia and Esanik. A.M.C. serves on the scientific advisory board of Tempus and Ascentage. The other authors declare no competing interests.

**Additional information**
**Correspondence and requests for materials** should be addressed to Arul M. Chinnaiyan.

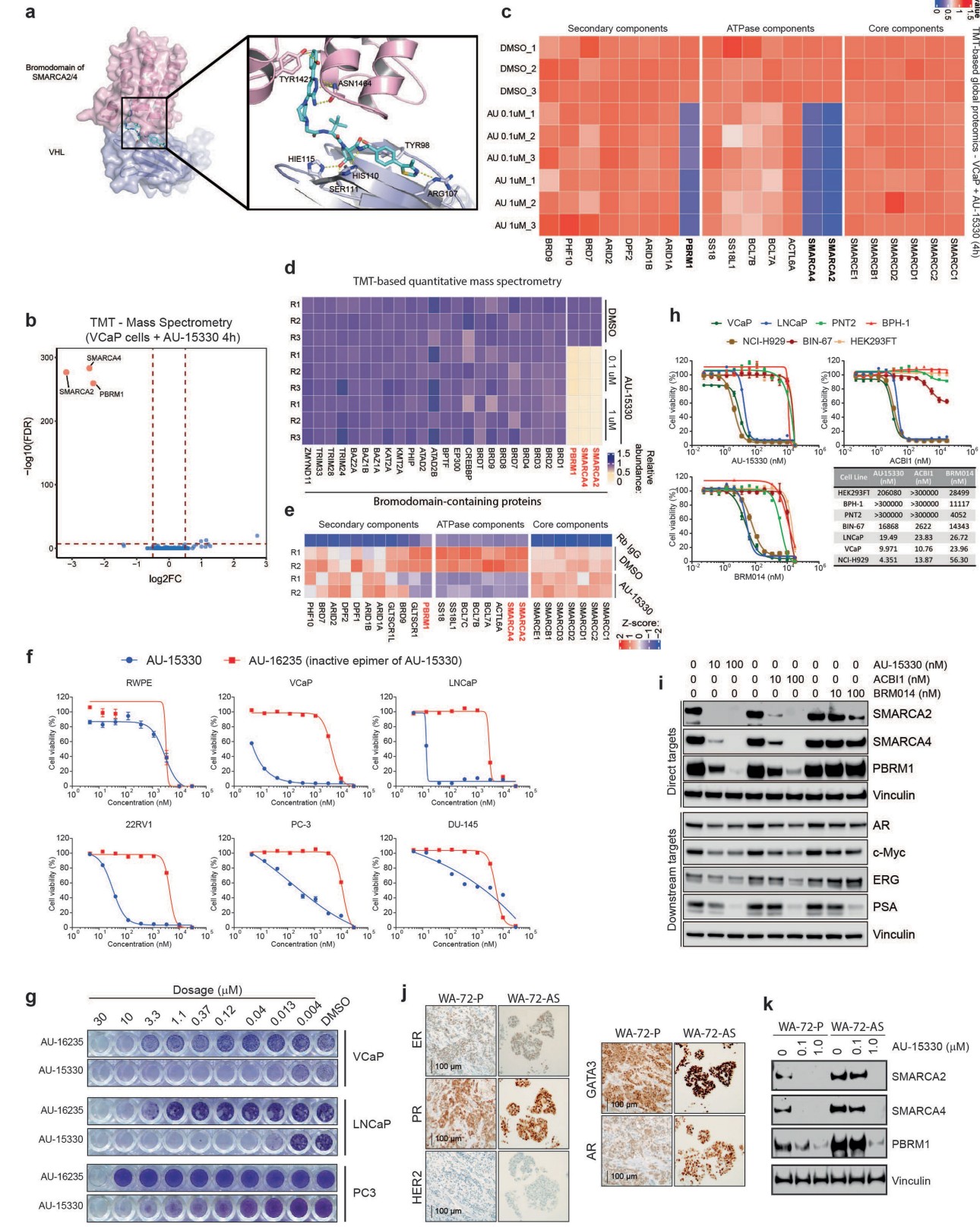

**Extended Data Fig. 1** | See next page for caption.

**Extended Data Fig. 1 | Conformational model of AU-15330 target interaction and activity profile in diverse cell lines.** (**a**) Docking model of AU-15330 (cyan sticks) with the SMARCA2 and VHL complex. AU-15330 is suggested to fit into the pocket of SMARCA2 and VHL and capture several key interactions. Key hydrogen bond interactions with protein residues (pink sticks in SMARCA2, white sticks in VHL) are shown by yellow dashes. (**b**) Effects of AU-15330 (1 μM, 4h) on the proteome of VCaP cells. Data plotted Log2 of the fold change (FC) versus DMSO (dimethyl sulfoxide) control against −Log10 of the *p-value* per protein (FDR, false discovery rate) from n = 3 independent experiments. All t-tests performed were two-tailed t-tests assuming equal variances. TMT, tandem mass tag. (**c**) Heatmap showing TMT-based MS abundance of detectable SWI/SNF components after 4h of treatment with AU-15330 at 1 μM. Data from three independent replicates are shown. (**d**) Heatmap of relative abundance of several bromodomain-containing proteins detected via Tandem Mass Tag (TMT)-based quantitative MS upon 4h AU-15330 treatment. DMSO, dimethyl sulfoxide (vehicle). (**e**) Heatmap of mammalian SWI/SNF (BAF) complex subunits split into three constituent modules detected in SMARCC1 (also known as BAF155) nuclear co-immunoprecipitation followed by MS. Direct AU-15330 targets are in bold. (**f**) Dose-response curves of cells treated with AU-15330 and AU-16235 (inactive epimer of AU-15330). Data are presented as mean +/− SD (n = 6) from one-of-three independent experiments. (**g**) Crystal violet staining showing the effect of AU-15330 on colony formation. This experiment was repeated independently twice. (**h**) Dose-response curves and $IC_{50}$ of cells treated with AU-15330, ACBI1, and BRM014. Data are presented as mean +/− SD (n = 6) from one-of-three independent experiments. (**i**) Immunoblots of noted proteins in VCaP cells treated with AU-15330, ACBI1, or BRM014 at increasing concentrations for 24h. Vinculin is the loading control probed on all immunoblots. This experiment was repeated independently twice. (**j**) Representative immunohistochemistry images showing expression of indicated proteins in patient-derived breast cancer cell lines. (**k**) Immunoblots of noted proteins in WA-72-P or WA-72-As breast cancer cells treated with DMSO or AU-15330 at noted concentrations for 24h, Vinculin is the loading control probed on a representative immunoblot. This experiment was repeated independently twice.

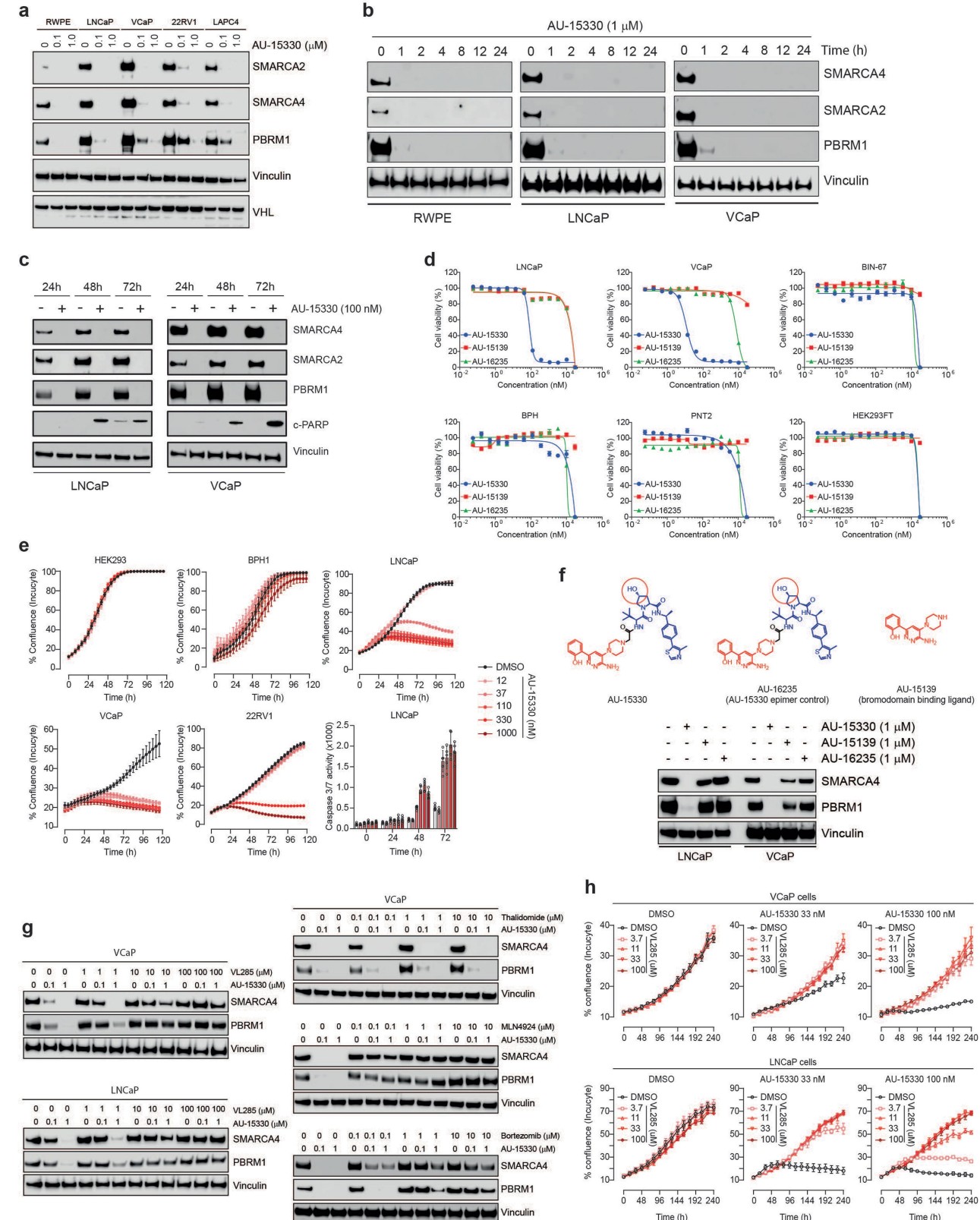

**Extended Data Fig. 2** | See next page for caption.

**Extended Data Fig. 2 | Verification of PROTAC design of AU-15330 and confirmation of on-target growth effects.** (**a**) Immunoblots for indicated proteins in normal (RWPE) or PCa cells (LNCaP, VCaP, 22RV1, and LAPC4) treated with AU-15330 at varied concentrations. Vinculin is the loading control probed on a representative immunoblot. This experiment was repeated independently twice. (**b**) Western blot analysis showing the time-dependent effect of AU-15330 on SMARCA2, SMARCA4, and PBRM1 in RWPE, LNCaP, and VCaP cells. Vinculin is the loading control probed on a representative immunoblot. This experiment was repeated independently twice. (**c**) Immunoblots in LNCaP and VCaP cells examining time-dependent cleavage of PARP upon AU-15330 treatment. Vinculin is the loading control probed on a representative immunoblot. This experiment was repeated independently twice. (**d**) Dose-response curves of VCaP, LNCaP, PNT2, PNT2, BPH1, Bin67, and HEK293 cells treated with AU-15330, AU-15139, or AU-16235. Data are presented as mean +/− SD (n = 6) from one-of-three independent experiments. (**e**) Growth curves of non-neoplastic or PCa cells upon treatment with increasing concentrations of AU-15330. Bottom, rightmost panel shows real-time assessment of apoptotic signals in LNCaP cells after treatment with DMSO or increasing AU-15330 concentrations. Data are presented as mean +/− SD (n = 5) from one-of-three independent experiments. (**f**) (top) Chemical structure of AU-15330, AU-16235 (an epimer control of AU-15330), and AU-15139 (parent bromodomain-binding ligand of AU-15330). (bottom) Immunoblots for SMARCA4 and PBRM1 in LNCaP and VCaP cells treated with AU-15330, AU-15139, or AU-16235 at indicated concentrations. Vinculin is the loading control probed on all immunoblots. This experiment was repeated independently twice. (**g**) Immunoblots of SMARCA4 and PBRM1 in VCaP and LNCaP cells pre-treated with VL285, MLN4924, bortezomib, or thalidomide for 1h, then treated with AU-15330 at noted concentrations for 4h. Vinculin is the loading control probed on all immunoblots. This experiment was repeated independently twice. (**h**) Real-time measure showing the rescue effect of VHL ligand on AU-15330-mediated growth inhibition in VCaP and LNCaP cells. Data are presented as mean +/− SD (n = 4) from one-of-three independent experiments.

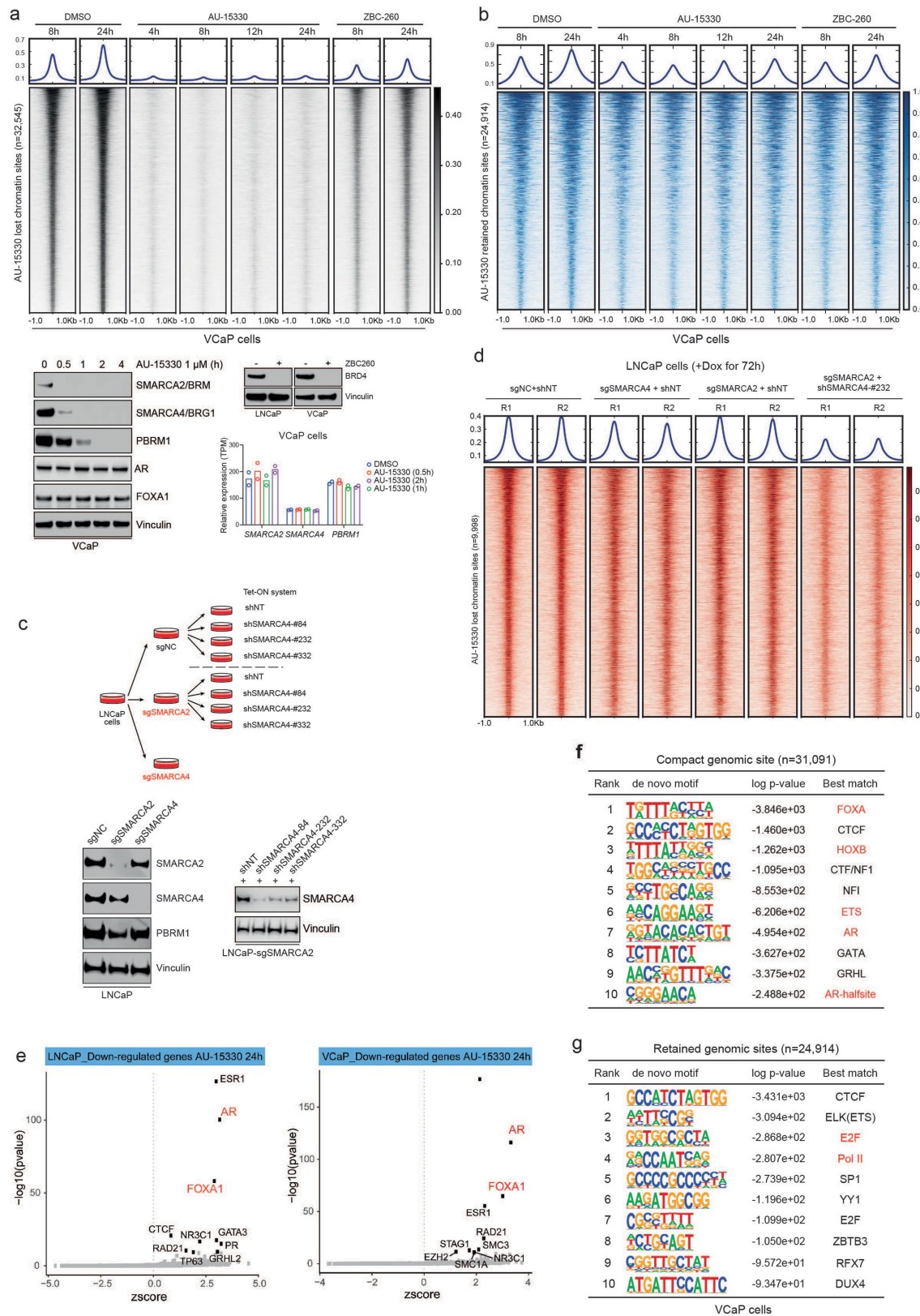

**Extended Data Fig. 3** | See next page for caption.

**Extended Data Fig. 3 | SWI/SNF ATPases, SMARCA2 and SMARCA4, mediate chromatin accessibility at numerous sites across the genome in PCa cells.** (**a**, **b**) ATAC-seq read-density heatmaps from VCaP cells treated with DMSO (solvent control), AU-15330, or ZBC-260 (a BRD4 degrader) for indicated durations at genomic sites that are compacted (**a**) or remain unaltered (**b**) upon AU-15330 treatment. Immunoblots show loss of target proteins upon treatment of cancer cells with AU-15330 (1 μM) for increasing durations or ZBC-260 (10 nM) for 4h. Vinculin is the loading control probed on all immunoblots. This experiment was repeated independently twice. Barplot shows the changes in mRNA expression (RNA-seq) of AU-15330 (1 μM) target genes in VCaP cells treated for noted durations. (**c**) Schematic outlining the CRISPR/Cas9 and shRNA-based generation of LNCaP cells with either independent or simultaneous inactivation of SWI/SNF ATPases, SMARCA2 and SMARCA4. Immunoblots showing the decrease in target expression in the genetic models shown above. Vinculin is the loading control probed on a representative immunoblot. This experiment was repeated independently twice. (**d**) ATAC-seq read-density heatmaps from genetically engineered LNCaP cells with SMARCA2 and/or SMARCA4 functional inactivation at AU-15330-compacted genomic sites. (**e**) Binding analysis for the regulation of transcription (BART) prediction of specific transcription factors mediating the observed transcriptional changes upon AU-15330 treatment in LNCaP or VCaP cells. The top 10 significant and strong (z-score) mediators of transcriptional responses are labeled (BART, Wilcoxon rank-sum test). (**f**) Top ten de novo motifs (ranked by *p-value*) enriched within AU-15330-compacted genomic sites (HOMER, hypergeometric test) in VCaP cells. (**g**) De novo motif analysis with top 10 motifs (ranked by *p-value*) enriched within genomic sites that retain chromatin accessibility upon AU-15330 treatment in VCaP cells (HOMER hypergeometric test).

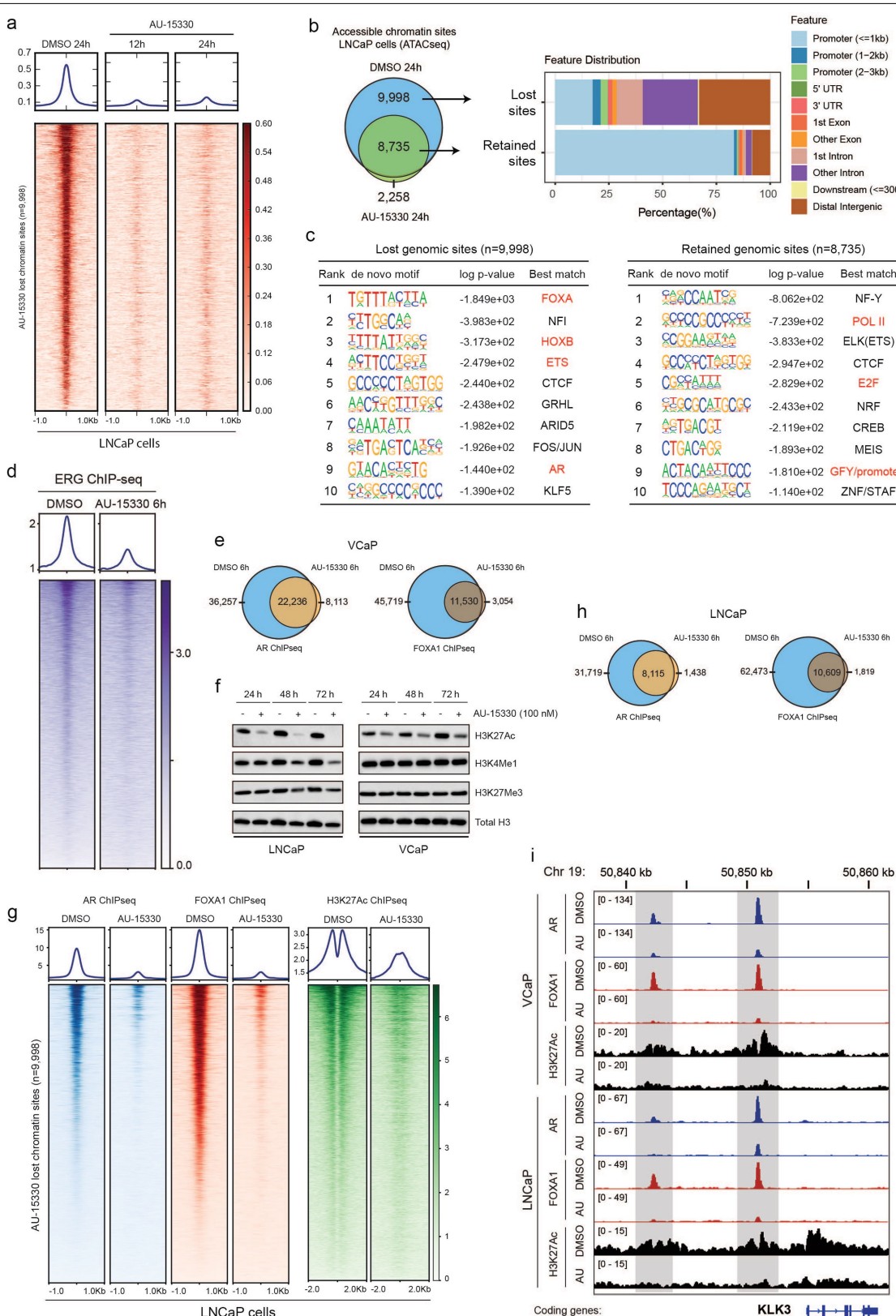

**Extended Data Fig. 4** | See next page for caption.

**Extended Data Fig. 4 | SWI/SNF inhibition condenses chromatin at enhancer sites bound by oncogenic transcription factors AR and FOXA1 in PCa cells.** (**a**) ATAC-seq read-density heatmaps from LNCaP cells treated with DMSO or AU-15330 for indicated durations at all genomic sites that lose physical accessibility upon AU-15330 treatment. (**b**) Genome-wide changes in chromatin accessibility upon AU-15330 treatment for 12h in LNCaP cells, along with genomic annotation of sites that are lost or retained in the AU-15330-treated cells. (**c**) De novo motif analysis with top 10 motifs (ranked by *p-value*) enriched within AU-15330-compacted or unaltered genomic sites in LNCaP cells (HOMER, hypergeometric test). (**d**) ChIP-seq read-density heatmaps for ERG at the AU-15330-compacted genomic sites in VCaP cells after treatment with DMSO or AU-15330 (1 μM) for indicated times and stimulation with R1881 (1 nM, 3h). (**e**) Genome-wide changes in AR and FOXA1 ChIP-seq peaks upon AU-15330 treatment (1 μM, 6h) in VCaP cells stimulated with R1881, a synthetic androgen (1 nM, 3h). (**f**) Immunoblots showing the changes in indicated chemical histone marks upon treatment with AU-15330. Vinculin is the loading control probed on a representative immunoblot. This experiment was repeated independently twice. (**g**) ChIP-seq read-density heatmaps for AR, FOXA1, and H3K27Ac at the compacted genomic sites in LNCaP cells after indicated durations of treatment with AU-15330 (1 μM). (**h**) Genome-wide changes in AR and FOXA1 ChIP-seq peaks upon AU-15330 treatment (1 μM, 6h) in LNCaP cells stimulated with R1881 (1 nM, 3h). (**i**) ChIP-seq tracks for AR, FOXA1, and H3K27Ac within the *KLK2/3* gene locus in R1881-stimulated VCaP and LNCaP cells with or without AU-15330 (AU).

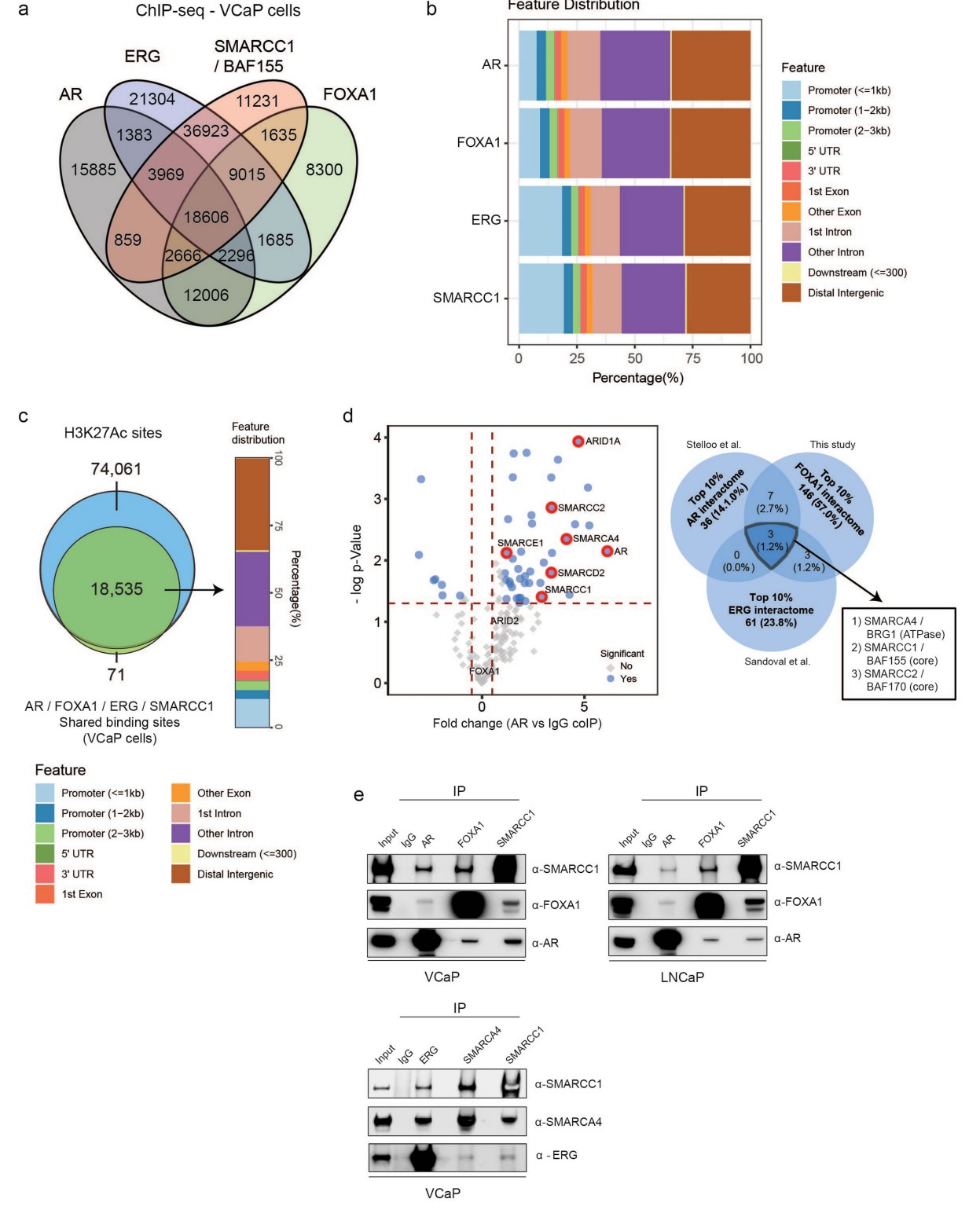

Note: ProteinA-HRP was used as the secondary antibody
for these immunoblots.

**Extended Data Fig. 5 | The SWI/SNF complex is a common chromatin cofactor of the central transcriptional machinery in PCa cells. (a)** The overlap between AR, FOXA1, ERG, and SMARCC1 ChIP-seq peaks in VCaP cells. **(b)** Genomic annotation of oncogenic transcription factor and SWI/SNF (SMARCC1) chromatin binding sites. **(c)** The overlap between transcription factor and SWI/SNF complex shared genomic sites (from a) and H3K27Ac ChIP-seq peaks along with the genomic annotations of the shared binding sites. **(d)** Left: volcano plot showing the AR interacting proteins identified from AR immunoprecipitation followed by MS. Significantly enriched SWI/SNF subunits are highlighted in red (two-sided t-test). Right: Overlap between AR, FOXA1, and ERG interacting proteins identified from in-house or publicly available datasets. **(e)** Immunoblots for indicated proteins followed by nuclear co-immunoprecipitation (IP) of AR, FOXA1, ERG, or SMARCC1 (a core SWI/SNF subunit) in VCaP and LNCaP cells after DHT (dihydrotestosterone) stimulation (10 nM, 3h). This experiment was repeated independently twice.

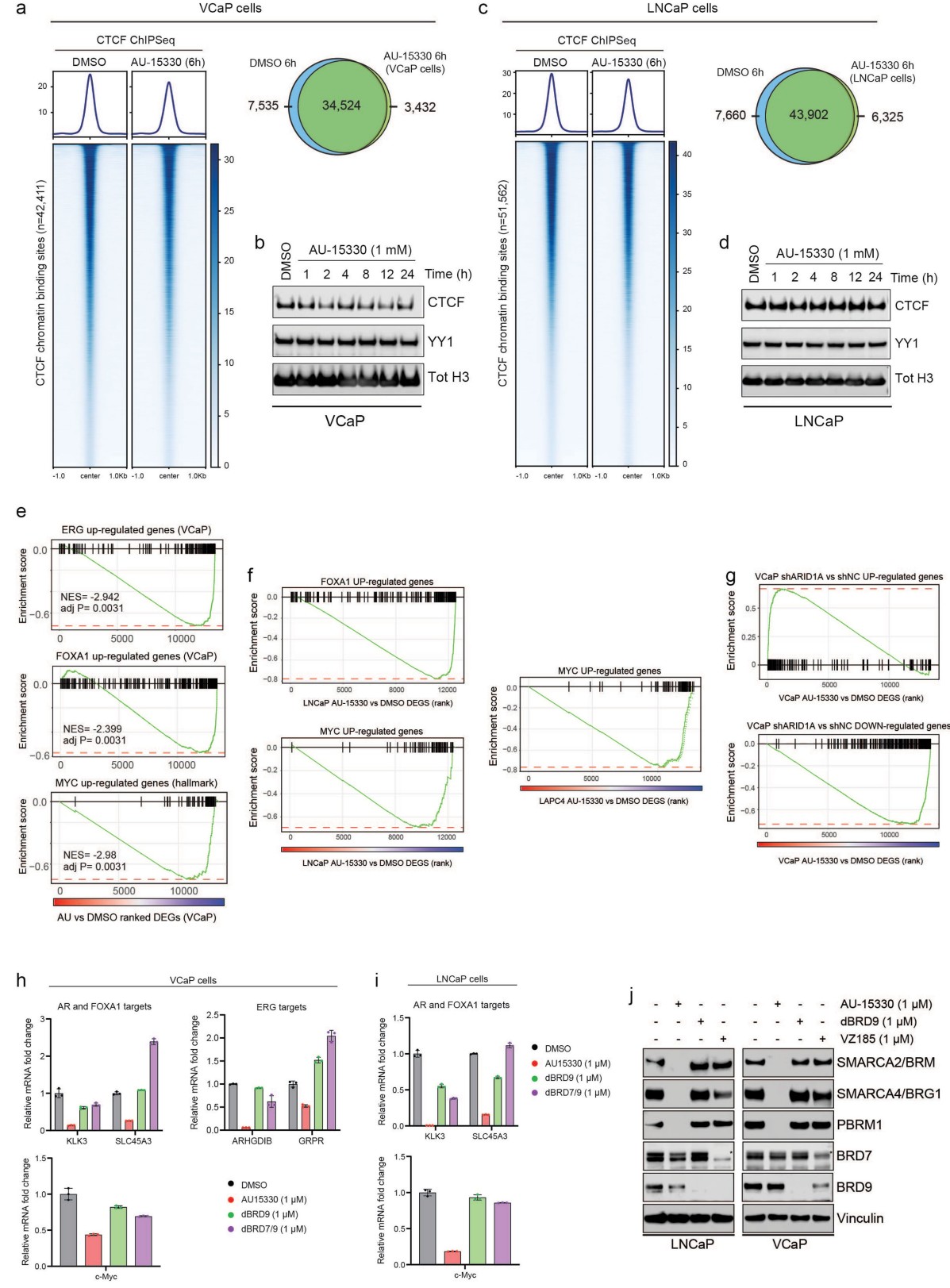

**Extended Data Fig. 6** | See next page for caption.

**Extended Data Fig. 6 | The canonical SWI/SNF complex is the primary cofactor of enhancer-binding transcription factors and is essential for enabling their oncogenic gene programs.** (**a**, **c**) Genome-wide ChIP-seq read-density heatmaps and Venn diagrams for CTCF in VCaP (**a**) or LNCaP (**c**) cells treated with either DMSO or AU-15330 (1 μM) for 6h. Vinculin is the loading control probed on a representative immunoblot. (**b**, **d**) Immunoblots of indicated proteins in VCaP (**b**) or LNCaP (**d**) cells treated with AU-15330 (1 μM) for increasing time durations. Total histone H3 is the loading control probed on all immunoblots. This experiment was repeated independently twice. (**e**) GSEA plots for ERG, FOXA1, and MYC-regulated genes using the fold change rank-ordered gene signature from AU-15330-treated (1 nM, 24h) VCaP cells. NES, net enrichment score; adj P, adjusted p-value; DEGs, differentially expressed genes.

(**f**, **g**) GSEA of FOXA1, MYC, or ARID1A-regulated genes (see **Methods** for gene sets) in the fold change rank-ordered AU-15330 gene signature in indicated PCa cells. DEGs, differentially expressed genes. (n = 2 biological replicates, GSEA enrichment test) (**h**, **i**) Expression of indicated genes (qPCR) in VCaP (**h**) or LNCaP (**i**) cells upon treatment with DMSO, AU-15330, dBRD7 (BRD7 degrader), or dBRD7/9 (dual BRD7 and BRD9 degrader) at 1 μM for 24h. Data are presented as mean +/− SD (n = 3, technical replicates) from one-of-two independent experiments. (**j**) Immunoblots for indicated proteins in LNCaP and VCaP cells treated with AU-15330, dBRD9 (BRD9 degrader), or VZ185 (BRD7/9 degrader) at indicated concentrations. Vinculin is the loading control probed on all immunoblots. This experiment was repeated independently twice.

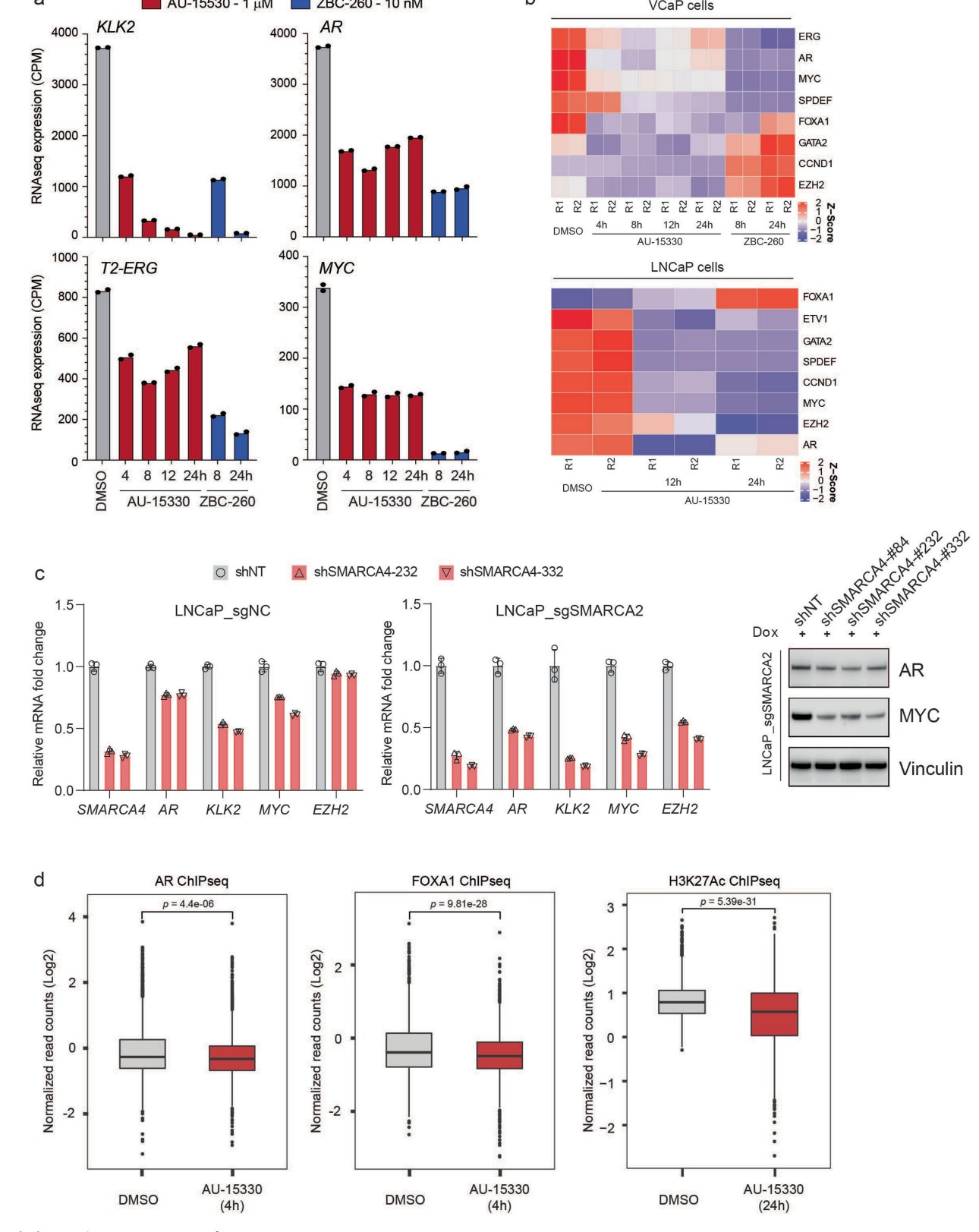

**Extended Data Fig. 7** | See next page for caption.

**Extended Data Fig. 7 | SWI/SNF inhibition down-regulates the expression of oncogenic drivers through disruption of promoter and super-enhancer interactions.** (**a**, **b**) RNA expression (RNA-seq) heatmaps from VCaP or LNCaP cells treated with DMSO, AU-15330 (1 μM), or ZBC-260 (BRD4 degrader) for the noted durations. n = 2, biological replicates. (**c**) RNA expression (qPCR) of indicated genes in stable CRISPR-engineered LNCaP-sgNC (control) or LNCaP-sgSMARCA2 (SMARCA2 inactivation) cells that were treated with a non-target control shRNA or two distinct shRNAs targeting the *SMARCA4* gene. Data are presented as mean +/− SD (n = 3, technical replicates) from one-of-two independent experiments. Right, immunoblots showing expression of the indicated protein in CRISPR/shRNA-engineered LNCaP cells. Vinculin is the loading control probed on a representative immunoblot. This experiment was repeated independently twice. (**d**) Normalized read density of AR, FOXA1 and H3K27Ac ChIP-seq signal at the super-enhancer sites (n = 1,551 sites) in VCaP cells treated with DMSO or AU-15330 (1 μM) for 4h or H3K27Ac with 24h AU-15330 (two-sided t-test). For all box plots, the center shows median, box marks quartiles 1–3, and whiskers span quartiles 1–3 ± 1.5 × IQR.

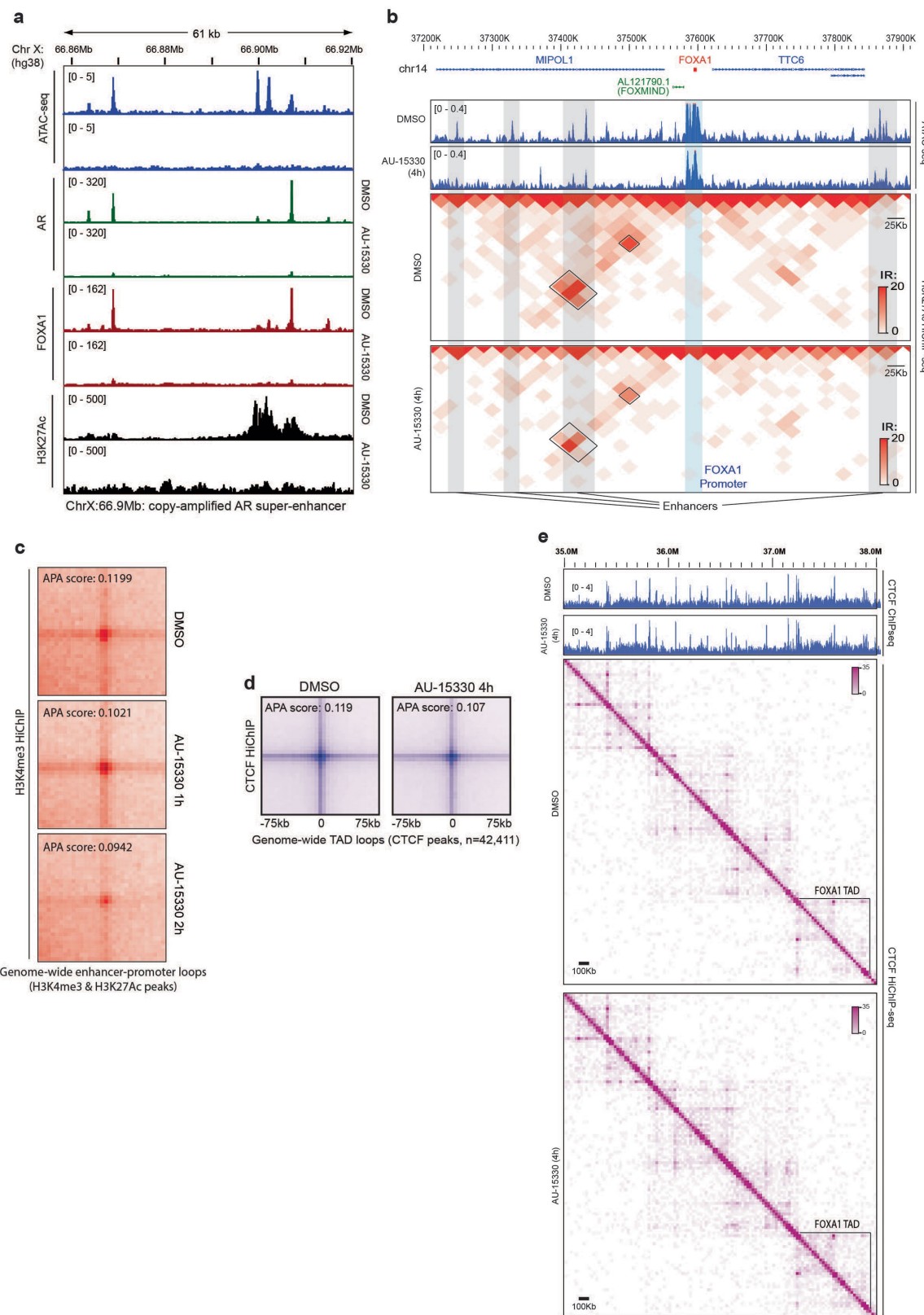

**Extended Data Fig. 8** | See next page for caption.

**Extended Data Fig. 8 | Enhancer-promoter interactions at loci of oncogenic transcription factors with AU-15330.** (**a**) ATAC-seq and ChIP-seq tracks for AR, FOXA1, and H3K27Ac within the *AR* gene locus in VCaP cells with or without AU-15330 treatment (1 μM for 6h for AR and FOXA1; 1 μM for 24h for H3K27Ac). (**b**) H3K27Ac HiChIP-seq heatmaps within the *FOXA1* gene locus in VCaP cells plus/minus treatment with AU-15330 (1 μM) for 4h (bin size = 25Kb). ATAC-seq read-density tracks from the same treatment conditions are overlaid. Grey highlights mark enhancers, while the blue highlight marks the *FOXA1* promoter. (**c**) Aggregate peak analysis (APA) plots for H3K4me3 (active promoter mark) HiChIP-seq data for all possible interactions between putative enhancers and gene promoters in VCaP cells plus/minus treatment with AU-15330 (1 μM) for noted durations. (**d**) APA plots for CTCF HiChIP-seq data for all possible interactions between CTCF-bound insulator elements in VCaP cells plus/minus treatment with AU-15330 (1 μM, 4h). TAD, topologically associating domain. (**e**) CTCF HiChIP-seq heatmaps in a gene locus at Chr14, including the *FOXA1* topologically associating domain (TAD), in VCaP cells plus/minus treatment with AU-15330 (1 μM) for 4h (bin size = 100Kb). CTCF ChIP-seq read-density tracks from VCaP cells plus/minus AU-15330 treatment (1 μM) for 6h are overlaid.

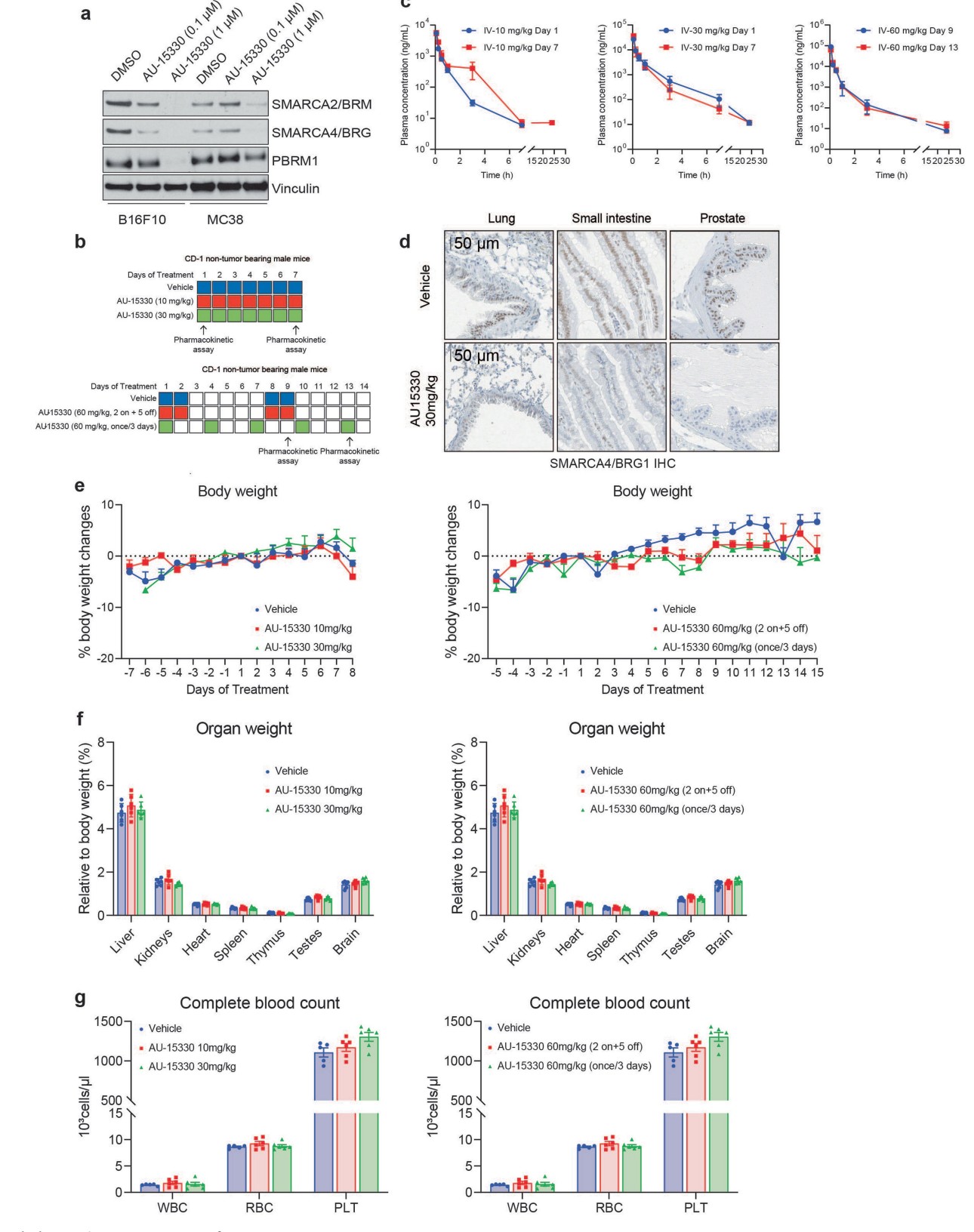

**Extended Data Fig. 9 |** See next page for caption.

**Extended Data Fig. 9 | AU-15330 is well tolerated in mice and induces on-target degradation of SMARCA2, SMARCA4, and PBRM1.** (**a**) Immunoblots of indicated proteins in B16F10 and MC38 cells treated with DMSO or AU-15330 (100 nM or 1 μM). Vinculin is the loading control probed on a representative immunoblot. This experiment was repeated independently twice. (**b**) Schematic outlining the AU-15330 *in vivo* study in non-tumor bearing CD-1 mice. Male mice were treated with vehicle (control) or AU-15330 at the indicated concentration throughout the experiment. (**c**) Pharmacokinetics profile of AU-15330 following intravenous (IV) injection in CD-1 mice. Mice received a single injection at indicated concentration of AU-15330, and plasma levels were determined by HPLC. Data are presented as mean +/− SD (n = 6, biological replicates). (**d**) Immunohistochemistry staining of SMARCA4/BRG1 was carried out using lung, small intestine, and prostate sections after necropsy to show on-target efficacy of AU-15330 *in vivo* (n = 2, biological replicates). (**e**) Body weight measurements showing AU-15330 does not affect weight of non-tumor bearing CD-1 mice. Data are presented as mean +/− SD (n = 6, biological replicates). (**f**) Major organ weight measurements (taken after necropsy) showing AU-15330 does not affect their weight in non-tumor bearing CD-1 mice. Data are presented as mean +/− SD (n = 6, biological replicates). (**g**) Complete blood count showing AU-15330 does not affect the hematologic system. Non-tumor bearing CD-1 mice were treated with vehicle or AU-15330 at the indicated concentration throughout the treatment period, and whole blood was then collected and processed. WBC, white blood cells; RBC, red blood cells; PLT, platelets. Data are presented as mean +/− SD (n = 6, biological replicates).

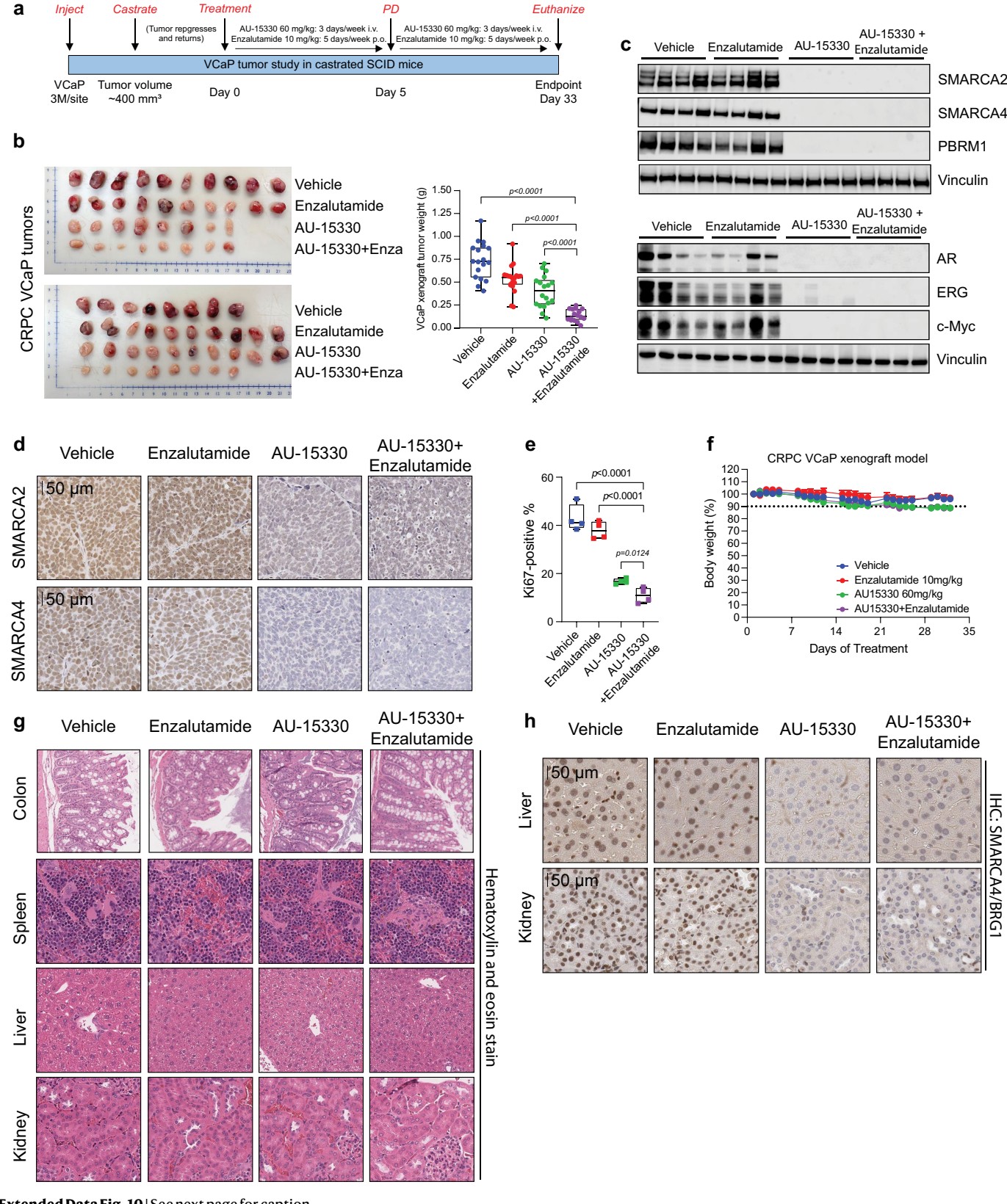

**Extended Data Fig. 10** | See next page for caption.

**Extended Data Fig. 10 | Combined *in vivo* treatment with AU-15330 and enzalutamide causes tumor regression in PCa xenografts without toxic effects on other organs.** (**a**) Schematic outlining the AU-15330 *in vivo* efficacy study using the VCaP-CRPC xenograft model. VCaP cells were subcutaneously grafted in immunocompromised mice that were castrated after 2 weeks of tumor growth to induce disease regression. This was eventually followed by tumor re-growth in the androgen-depleted conditions, generating the aggressive, castration-resistant tumors. (**b**) Individual tumors and weights from vehicle, enzalutamide, AU-15330, and AU-15330+enzalutamide groups from VCaP-CRPC study (two-sided t-test). Data are presented as mean+/−SEM (vehicle: n = 18, enzalutamide: n = 20, AU-15330: n = 18, AU-15330+enzalutamide: n = 16). For all box plots, the center shows median, box marks quartiles 1–3, and whiskers span the range. (**c**) Immunoblots of direct AU-15330 targets (upper) and oncogenic transcription factors (bottom) from VCaP-CRPC xenografts (n = 4 tumors/arm) after 5 days of *in vivo* treatment. Vinculin is the loading control probed on a representative immunoblot. (**d**) Representative immunohistochemistry images from the VCaP-CRPC xenograft study (n = 2 tumors/arm) for SMARCA2 and SMARCA4. (**e**) Box plot of the percent of cells with positive Ki-67 staining. Two-sided t-test shows significant differences between vehicle vs. enzalutamide, AU-15330, or AU-15330+enzalutamide groups. Data are presented as mean +/− SEM (n = 4, biological replicates). For all box plots, the center shows median, box marks quartiles 1–3, and whiskers span the range. (**f**) Percent body weight measurement showing the effect of vehicle, enzalutamide, AU-15330, and combination of AU-15330 and enzalutamide throughout the treatment period (two-sided t-test). Data are presented as mean +/− SEM (vehicle: n = 9, enzalutamide: n = 10, AU-15330: n = 9, AU-15330 + enzalutamide: n = 8). (**g**) H&E staining was carried out to examine the effect of AU-15330 *in vivo* using colon, spleen, liver, and kidney sections after necropsy. Representative images of H&E staining are shown. (**h**) Immunohistochemistry staining of SMARCA4/BRG1 was carried out using liver and kidney sections after necropsy to show on-target efficacy of AU-15330 *in vivo*.

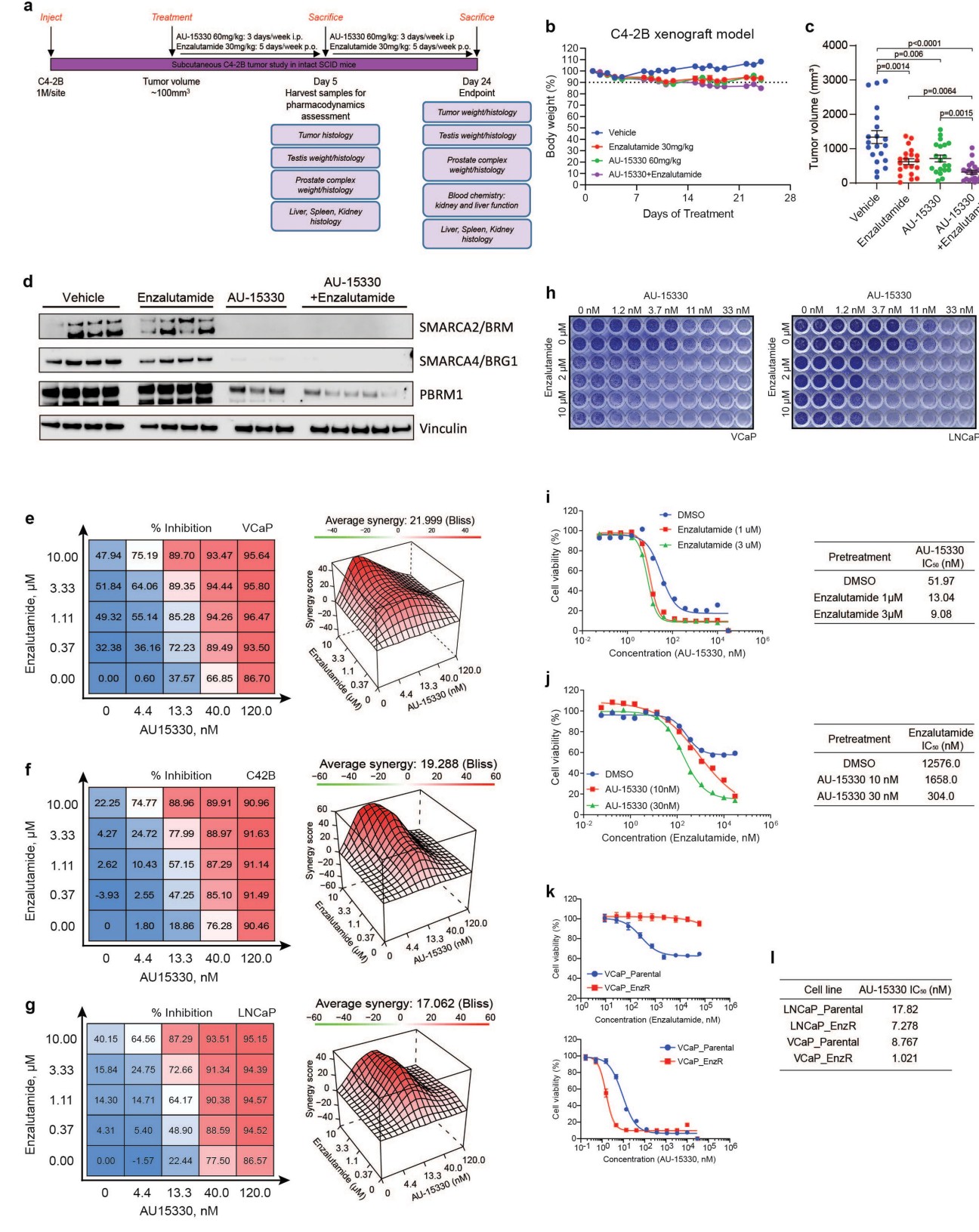

**Extended Data Fig. 11** | See next page for caption.

**Extended Data Fig. 11 | AU-15330 inhibits CRPC growth and synergizes with the AR antagonist enzalutamide.** (**a**) Schematic outlining the AU-15330 *in vivo* efficacy study using the C4-2B (CRPC) xenograft model. C4-2B-xenograft bearing male mice were castrated and, upon tumor regrowth, randomized into various treatment arms. (**b**) Body weight measurements showing the effect of the indicated treatments on animal weight. Tumor-bearing SCID mice were treated with the indicated drug throughout the treatment period, and the body weight was measured at endpoint. Data are presented as mean +/− SEM (n = 10, biological replicates). (**c**) Individual tumor volumes from different treatment groups with p-values are shown (two-sided t-test). Data are presented as mean +/− SEM (n = 20, biological replicates). (**d**) Immunoblots of direct AU-15330 targets (SMARCA2, SMARCA4, and PBRM1) in the whole cell lysate from C4-2B xenografts from all treatment arms after 5 days of *in vivo* treatment (n = 4, biological replicates). Vinculin is the loading control probed on a representative immunoblot. (**e-g**) VCaP, C4-2B, and LNCaP cells were treated with AU-15330 and/or enzalutamide at varied concentrations to determine the effect on cell growth and drug synergism, with assessments using the Bliss Independence method. Red peaks in the 3D-plots denote synergy with the average synergy scores noted above. The mean of three biological replicates is shown on top. Data are presented as mean (n = 4) from one-of-three independent experiments. (**h**) Crystal violet staining showing the synergistic effect of AU-15330 and enzalutamide on colony formation in VCaP and LNCaP. (**i,j**) Dose−response curves of VCaP cells treated with enzalutamide in combination with DMSO or AU-15330 at indicated concentrations. Data are presented as mean +/− SD (n = 4) from one-of-three independent experiments. (**k**) Dose-response curves of VCaP_Parental and VCaP_EnzR cells treated with enzalutamide or AU-15330. Data are presented as mean +/− SD (n = 6) from one-of-three independent experiments. (**l**) $IC_{50}$ for AU-15330 in enzalutamide-resistant (EnzR) LNCaP and VCaP cells after 5 days of treatment.

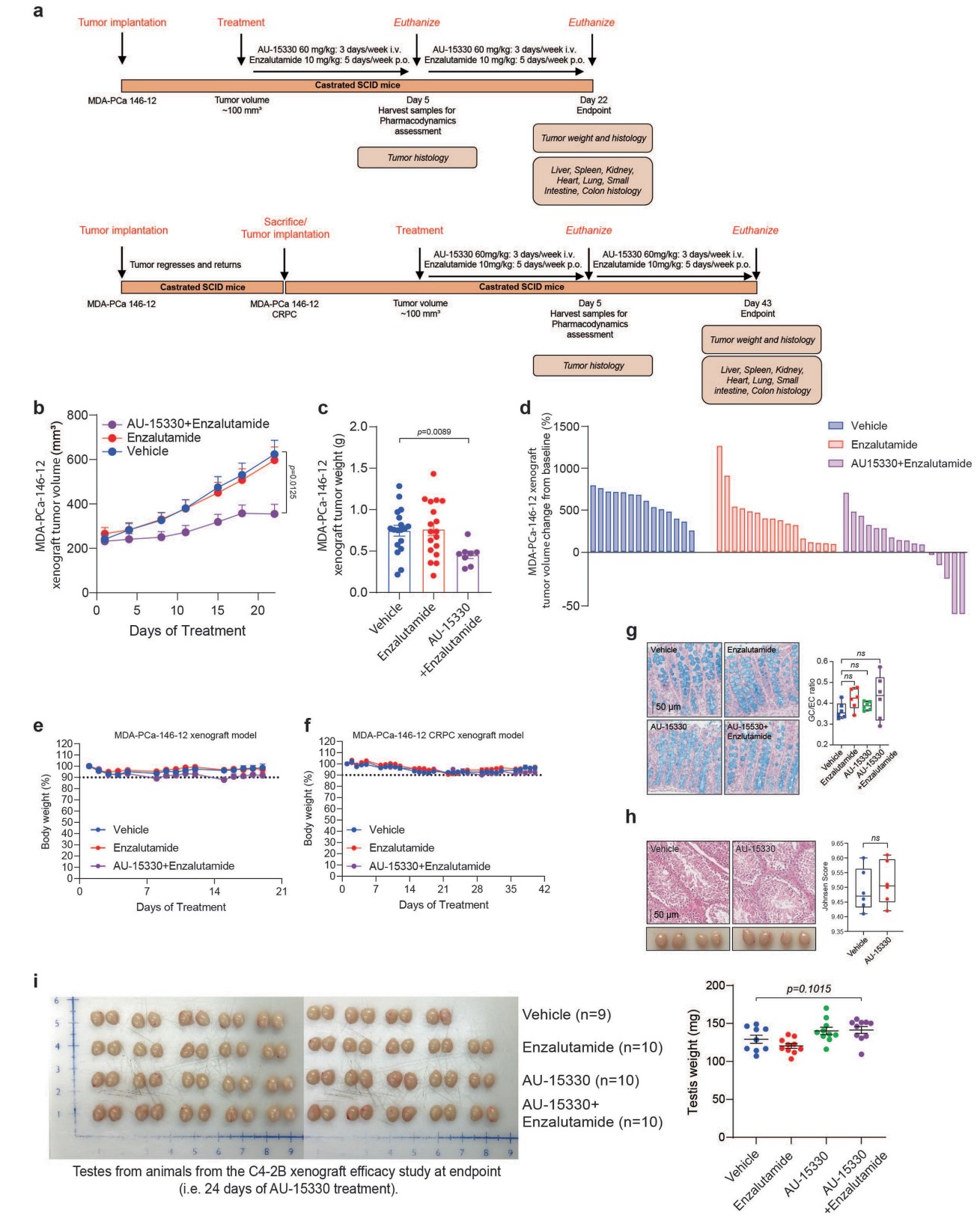

**Extended Data Fig. 12** | See next page for caption.

**Extended Data Fig. 12 | AU-15330 inhibits tumor growth of an enzalutamide-resistant patient-derived xenograft (PDX) model without evident toxicities. (a)** Schematics outlining the AU-15330 *in vivo* efficacy studies using the MDA-PCa-146-12 (top) or the MDA-PCa-146-12-CRPC (bottom) xenograft model. MDA-PCa-146-12-CRPC xenograft-bearing male mice were castrated and, upon tumor regrowth, randomized into various treatment arms that were administered vehicle, enzalutamide, or the combination of AU-15330+enzalutamide at indicated concentrations. **(b)** Tumor volume measurements (caliper twice per week) showing efficacy of enzalutamide alone or in combination with AU-15330 in the enzalutamide-resistant MDA-PCa-146-12 PDX model (n = 20/arm; two-sided t-test). Data are presented as mean +/− SEM (vehicle: n = 18, enzalutamide: n = 18, AU-15330+enzalutamide: n = 16). **(c)** Individual tumor weights from different treatment groups from the MDA-PCa-146-12 PDX study with *p*-values indicated (two-sided t-test). Data are presented as mean +/− SEM (vehicle: n = 18, enzalutamide: n = 18, AU-15330+enzalutamide: n = 8). **(d)** Waterfall plot showing percent change from baseline of individual tumors from the MDA-PCa-146-12-CRPC model with indicated treatment group after 43 days of treatment. **(e, f)** Animal body weight measurements showing the effect of vehicle, enzalutamide, and combination of AU-15330 and enzalutamide on animal weight in the **(e)** MDA-PCa-146-12 or the **(f)** MDA-PCa-146-12-CRPC PDX models. Tumor-bearing SCID mice were treated with vehicle, enzalutamide, or a combination of AU-15330 and enzalutamide at the indicated concentration throughout the treatment period. Data are presented as mean +/− SEM (for e, vehicle: n = 9, AU-15330: n = 9, AU-15330+enzalutamide: n = 8; for f, vehicle: n = 7, AU-15330: n = 8, AU-15330+enzalutamide: n = 8). **(g)** Representative Alcian blue staining images from the large intestinal tract harvested at the VCaP-CRPC efficacy study endpoint (n = 2/treatment group). *Right*, quantification of goblet:epithelial cell densities in the colon (two-sided t-test). Data are presented as mean +/− SEM (n = 6, biological replicates). **(h)** Top, Representative H&E of the testis gland harvested from DMSO or AU-15330-treated intact male mice after 21 days of *in vivo* treatment. Right, quantification of germ cell density and maturation carried out using the Johnsen scoring system (two-sided t-test). Bottom, gross images of the testis glands. Data are presented as mean +/− SEM (n = 6, biological replicates). For all box plots, the center shows median, box marks quartiles 1–3, and whiskers span the range. **(i)** Individual testes weight and images from different treatment groups of the C4-2B xenograft efficacy study at endpoint (i.e., after 24 days of treatment) with *p*-values indicated (two-sided t-test). Data are presented as mean +/− SEM (vehicle: n = 9, enzalutamide: n = 10, AU-15330: n = 10, AU-15330+enzalutamide: n = 10).

# nature research

# Reporting Summary

Nature Research wishes to improve the reproducibility of the work that we publish. This form provides structure for consistency and transparency in reporting. For further information on Nature Research policies, see our Editorial Policies and the Editorial Policy Checklist.

## Statistics

For all statistical analyses, confirm that the following items are present in the figure legend, table legend, main text, or Methods section.

| n/a | Confirmed | |
|---|---|---|
| ☐ | ☒ | The exact sample size ($n$) for each experimental group/condition, given as a discrete number and unit of measurement |
| ☐ | ☒ | A statement on whether measurements were taken from distinct samples or whether the same sample was measured repeatedly |
| ☐ | ☒ | The statistical test(s) used AND whether they are one- or two-sided<br>*Only common tests should be described solely by name; describe more complex techniques in the Methods section.* |
| ☐ | ☒ | A description of all covariates tested |
| ☐ | ☒ | A description of any assumptions or corrections, such as tests of normality and adjustment for multiple comparisons |
| ☐ | ☒ | A full description of the statistical parameters including central tendency (e.g. means) or other basic estimates (e.g. regression coefficient) AND variation (e.g. standard deviation) or associated estimates of uncertainty (e.g. confidence intervals) |
| ☐ | ☒ | For null hypothesis testing, the test statistic (e.g. $F$, $t$, $r$) with confidence intervals, effect sizes, degrees of freedom and $P$ value noted<br>*Give P values as exact values whenever suitable.* |
| ☒ | ☐ | For Bayesian analysis, information on the choice of priors and Markov chain Monte Carlo settings |
| ☒ | ☐ | For hierarchical and complex designs, identification of the appropriate level for tests and full reporting of outcomes |
| ☒ | ☐ | Estimates of effect sizes (e.g. Cohen's $d$, Pearson's $r$), indicating how they were calculated |

*Our web collection on statistics for biologists contains articles on many of the points above.*

## Software and code

Policy information about availability of computer code

| Data collection | No software was used for data collection. |
|---|---|
| Data analysis | All custom codes used for data analyses are freely available from the following public repositories:<br>https://github.com/mcieslik-mctp/papy<br>https://github.com/mcieslik-mctp/hpseq<br>https://github.com/mcieslik-mctp/bootstrap-rnascape<br>https://github.com/mcieslik-mctp/codac<br>https://github.com/mcieslik-mctp/crisp<br>https://github.com/mcieslik-mctp/<br>https://github.com/mctp/<br>https://github.com/dovetail-genomics/dovetail_tools<br><br>Computational tools used:<br>GraphPad Prism 9 and in-built statistical tools<br>SAMtools (version 1.9 or 1.13)<br>PICARD Mark Duplicates (version 2.9.0)<br>HOMER (version v.4.10)<br>MACS2 (version 2.1.1.20160309)<br>bcl2fastq conversion software (v2.20)<br>BWA (version 0.7.17-r1198-dirty)<br>Pairtools (version 0.3.0)<br>EdgeR (version 3.34.1)<br>HTSeq-count (version 0.13.5) |

deepTools (version 3.3.1)
ChipPeakAnno (version 3.0.0)
ChipSeeker (version 1.29.1)
R (version 3.6.0)
Cooler (version 0.8.11)
juicertools(version 1.22.01)
HiCExplorer (version 3.7)

For manuscripts utilizing custom algorithms or software that are central to the research but not yet described in published literature, software must be made available to editors and reviewers. We strongly encourage code deposition in a community repository (e.g. GitHub). See the Nature Research guidelines for submitting code & software for further information.

## Data

Policy information about availability of data

All manuscripts must include a data availability statement. This statement should provide the following information, where applicable:
- Accession codes, unique identifiers, or web links for publicly available datasets
- A list of figures that have associated raw data
- A description of any restrictions on data availability

All raw data for the graphs, immunoblots, and gel electrophoresis figures are included in the Source Data or Supplementary Information. All materials are available from the authors upon reasonable request. All the raw next-generation sequencing, ATAC, ChIP, RNA, and HiChIP-seq data generated in this study have been deposited in the Gene Expression Omnibus (GEO) repository at NCBI (accession code GSE171592).

# Field-specific reporting

Please select the one below that is the best fit for your research. If you are not sure, read the appropriate sections before making your selection.

☒ Life sciences ☐ Behavioural & social sciences ☐ Ecological, evolutionary & environmental sciences

For a reference copy of the document with all sections, see nature.com/documents/nr-reporting-summary-flat.pdf

# Life sciences study design

All studies must disclose on these points even when the disclosure is negative.

| | |
|---|---|
| Sample size | Sample sizes were empirically and statistically determined. For animal experiments, n=10-20 tumors were used for the pilot and efficacy studies. Using 20 tumors per treatment group, the statistical power to detect a 50% decrease in the mean tumor volume or metastatic burden in the treatment group is estimated to be 92.3% if the coefficient of variation (CV) is 40%. All in vitro experiments were performed with at least 3 technical replicates across two independent experiments. All samples sizes for various assays are listed in the Methods section or the figure legends. |
| Data exclusions | No data was excluded from the published publicly-available patient sequencing studies. For biological experiments, no data exclusions were made. |
| Replication | For all experiments, there are at least two independent biological repeats and multiple technical repeats in each. In all instances, all attempts at replicating the experiments produced similar results. |
| Randomization | For animal studies, mice were randomly assigned to treatment groups. For all other in vitro experiments, we used a common cell suspension to plate for both control and treatment groups. |
| Blinding | All histo-pathological evaluations of tissues and IHC/staining-based scoring for drug toxicity studies were carried out in a blinded manner by two independent pathologists. For all other experiments, the analyses did not require blinding as data quantification was carried out using instruments and automated workflows with no manual steps. |

# Reporting for specific materials, systems and methods

We require information from authors about some types of materials, experimental systems and methods used in many studies. Here, indicate whether each material, system or method listed is relevant to your study. If you are not sure if a list item applies to your research, read the appropriate section before selecting a response.

## Materials & experimental systems

| n/a | Involved in the study |
|---|---|
| ☐ | ☒ Antibodies |
| ☐ | ☒ Eukaryotic cell lines |
| ☒ | ☐ Palaeontology and archaeology |
| ☐ | ☒ Animals and other organisms |
| ☒ | ☐ Human research participants |
| ☒ | ☐ Clinical data |
| ☒ | ☐ Dual use research of concern |

## Methods

| n/a | Involved in the study |
|---|---|
| ☐ | ☒ ChIP-seq |
| ☒ | ☐ Flow cytometry |
| ☒ | ☐ MRI-based neuroimaging |

# Antibodies

| | |
|---|---|
| Antibodies used | Target antigen; Vendor; Catalog number; Lot number; Application; Note<br>BAF155 Cell Signaling Technology 11956S, Clone:D7F8S, Lot: 4, Western Blot, Co-IP 1:1000<br>SMARCA2/BRM Bethyl laboratories A301-016A, Lot: 1, Western Blot 1:1000<br>SMARCA4/BRG1 Cell Signaling Technoly 52251S, Lot: 1, Western Blot 1:1000<br>PBRM1 Bethyl laboratories A301-591A, Lot: 3, Western Blot 1:1000<br>BRD4 Cell Signaling Technology 13440S, Clone: E2A7X Western Blot 1:1000<br>BRD7 Proteintech 51009-2-AP Western Blot 1:1000<br>BRD9 Thermo Scientific PA5-113488, Lot: WE3273112, Western Blot 1:1000<br>Vinculin Millipore Sigma V9131 Western Blot 1:5000<br>VHL Thermo Fisher Scientific PA527322, Lot: UH2825110A, Western Blot 1:1000<br>AR Millipore Sigma 06-680, Lot: 3256650, Western Blot, Co-IP 1:1000<br>ERG Abcam ab92513  Western Blot, Co-IP 1:1000<br>FOXA1 Thermo Fisher Scientific PA5-27157  Western Blot, Co-IP 1:1000<br>c-Myc Cell Signaling Technology 5605S, Clone: D84C12, Western Blot 1:1000<br>PSA DAKO A0562, Lot: 00093790, Western Blot 1:4000<br>YY1 Diagenode C15410345 Western Blot 1:1000<br>MED1 Bethyl laboratories A300-793A Western Blot 1:1000<br>H3K27Me3 Diagenode C15410069  Western Blot 1:1000<br>H3K27Ac Cell Signaling Technology 8173, Clone: D5E4, Western Blot 1:1000<br>H3K4me3 Cell Signaling Technology 9751, Lot: 14, Clone: C42D8 Western Blot 1:1000<br>H3K4Me1 Abcam ab8895 Western Blot 1:1000<br>Cleaved PARP (Asp214) Cell Signaling Technology 9541, Clone: Asp214, Lot: 13, Western Blot 1:1000<br>SMARCA2/BRM Millipore sigma HPA029981 IHC 1:100<br>SMARCA4/BRG1 Abcam ab108318 IHC 1:100<br>AR Millipore Sigma 06-680 IHC 1:100<br>FOXA1 Thermo Fisher Scientific PA5-27157, Lot: VFS004672A, IHC 1:1000<br>ERG Cell Signaling Technology 97249S, Clone: A7L1G, Lot: 1, IHC 1:500, ChIP-seq 10 mg/7-8M cells<br>AR Millipore/Sigma 06-680 ChIP-seq 10 mg/7-8M cells<br>FOXA1 Thermo Fisher Scientific PA5-27157  ChIP-seq 10 mg/7-8M cells<br>H3K27Ac Abcam ab4729 ChIP-seq 10 mg/10M cells<br>CTCF Cell Signaling Technology 3418, Clone: D31H2, Lot: 4, HiChIP-seq 1.14 mg per IP<br>H3K4me3 Cell Signaling Technology 9751, Clone:C42D8, Lot: 14, HiChIP-seq 3.4 mg per IP<br>H3K27Ac Cell Signaling Technology 8173, Clone: D5E4, HiChIP-seq 0.4 mg per IP |
| Validation | All antibodies used in this study are from reputed commercial vendors and have been validated by the vendors (see website). QC data is directly available from all the vendor listed above and these antibodies have been commonly used in other publications. These details are included in the vendor web-links pasted below:<br>BAF155, https://www.cellsignal.com/products/primary-antibodies/smarcc1-baf155-d7f8s-rabbit-mab/11956<br>SMARCA2/BRM, https://www.bethyl.com/product/A301-016A/SMARCA2+BRM+Antibody<br>SMARCA2/BRG, https://www.sigmaaldrich.com/US/en/product/sigma/hpa029981<br>SMARCA4/BRG1, https://www.cellsignal.com/products/primary-antibodies/brg1-e9o6e-mouse-mab/52251<br>SMARCA4/BRG1, https://www.abcam.com/brg1-antibody-epr3912-ab108318.html<br>PBRM1, https://www.bethyl.com/product/A301-591A/PBRM1+Antibody<br>BRD4, https://www.cellsignal.com/products/primary-antibodies/brd4-e2a7x-rabbit-mab/13440<br>BRD7, https://www.ptgcn.com/products/BRD7-Antibody-51009-2-AP.htm<br>BRD9, https://www.thermofisher.com/antibody/product/BRD9-Antibody-Polyclonal/PA5-113488<br>Vinculin, https://www.sigmaaldrich.com/US/en/product/sigma/v9131<br>VHL, https://www.thermofisher.com/antibody/product/VHL-Antibody-Polyclonal/PA5-27322<br>AR, https://www.emdmillipore.com/US/en/product/Anti-Androgen-Receptor-Antibody,MM_NF-06-680<br>ERG, https://www.abcam.com/erg-antibody-epr3864-ab92513.html<br>ERG, https://www.cellsignal.com/products/primary-antibodies/erg-a7l1g-rabbit-mab/97249<br>FOXA1, https://www.thermofisher.com/antibody/product/FOXA1-Antibody-Polyclonal/PA5-27157<br>c-Myc, https://www.cellsignal.com/products/primary-antibodies/c-myc-d84c12-rabbit-mab/5605<br>YY1, https://www.diagenode.com/en/p/yy1-polyclonal-antibody-50-ug<br>MED1, https://www.bethyl.com/product/A300-793A/MED1+Antibody<br>H3K27Me3, https://www.diagenode.com/en/p/h3k27me3-polyclonal-antibody-classic-50-mg-34-ml<br>H3K27Ac, https://www.cellsignal.com/products/primary-antibodies/acetyl-histone-h3-lys27-d5e4-xp-rabbit-mab/8173 |

H3K27Ac, https://www.abcam.com/histone-h3-acetyl-k27-antibody-chip-grade-ab4729.html
H3K4Me1, https://www.abcam.com/Histone-H3-mono-methyl-K4-antibody-ChIP-Grade-ab8895.html?gclsrc=aw.ds|
aw.ds&gclid=CjwKCAjwzt6LBhBeEiwAbPGOgUFEy8GIMv4Wyw4MgVMXASeZXmacJ3JbieaWOcgXasSovoW1pm9ypRoCEWMQAvD_Bw
E
H3K4Me3, https://www.cellsignal.com/products/primary-antibodies/tri-methyl-histone-h3-lys4-c42d8-rabbit-mab/9751
Cleaved PARP, https://www.cellsignal.com/products/primary-antibodies/cleaved-parp-asp214-antibody-human-specific/9541
CTCF, https://www.cellsignal.com/products/primary-antibodies/ctcf-d31h2-xp-rabbit-mab/3418

# Eukaryotic cell lines

Policy information about cell lines

| | |
|---|---|
| Cell line source(s) | Most cell lines were originally obtained from ATCC, DSMZ, ECACC, or internal stock. C4-2B cells were generously provided by Evan Keller, Ph.D. at the University of Michigan (who originally purchased them from ATCC), CWR-R1 cells, and a series of enzalutamide-resistant prostate cancer cell lines (LNCaP_Parental, LNCaP_EnzR, CWR-R1_Parental, CWR-R1_EnzR, VCaP_Parental and VCaP_EnzR) were generated in the lab of and generously provided by Donald Vander Griend, Ph.D. at the University of Illinois at Chicago. HeLa cells were purchased from ATCC. All the cells were genotyped to confirm their identity at the University of Michigan Sequencing Core and tested routinely for Mycoplasma contamination. Additionally, all the cell lines were genotyped every two months to confirm their identity. LNCaP, 22RV-1, CWR-R1, PC-3, and DU145 were grown in Gibco RPMI-1640 + 10% FBS (ThermoFisher, Waltham, MA). VCaP was grown in Gibco DMEM + 10% FBS (ThermoFisher, Waltham, MA). |
| Authentication | All cell lines were biweekly tested to be free of mycoplasma contamination and genotyped every month at the University of Michigan Sequencing Core using Profiler Plus (Applied Biosystems) and compared with corresponding short tandem repeat (STR) profiles in the ATCC database to authenticate their identity in culture between passages and experiments. In particular, we ensured that the STR profile of HeLa cells were always >90% similar to the original, early passage cells. Also, HeLa cells were cultured in a separate hood to avoid any cross-contamination. |
| Mycoplasma contamination | All cells were biweekly tested for mycoplasma contamination using the MycoAlert PLUS Mycoplasma Detection Kit (Lonza) and were found to be continually negative. More details are included in the Methods section |
| Commonly misidentified lines (See ICLAC register) | None |

# Animals and other organisms

Policy information about studies involving animals; ARRIVE guidelines recommended for reporting animal research

| | |
|---|---|
| Laboratory animals | Efficacy studies: 4-6 week old male CB17 severe combined immunodeficiency (SCID) mice were procured from the University of Michigan breeding colony. Pharmacokinetics study: 9-11 week old CD-1 male mice were used. All mice were maintained under the conditions of pathogen-free, 12 hours light/12 hours dark cycle, temperatures of 18-23°C, and 40-60% humidity. |
| Wild animals | No wild animals were used in the study. |
| Field-collected samples | No field collected samples were used in the study. |
| Ethics oversight | The Institutional Animal Care & Use Committee (IACUC) ensures that the highest animal welfare standards are maintained along with the conduct of accurate, valid scientific research through the supervision, coordination, training, guidance, and review of every project proposed to include the use of vertebrate animals at the University of Michigan. |

Note that full information on the approval of the study protocol must also be provided in the manuscript.

# ChIP-seq

## Data deposition

☒ Confirm that both raw and final processed data have been deposited in a public database such as GEO.

☒ Confirm that you have deposited or provided access to graph files (e.g. BED files) for the called peaks.

| | |
|---|---|
| Data access links *May remain private before publication.* | We have deposited the raw as well as processed ATAC, RNA, ChIP and HiChIP sequencing files to the GEO superseries repository; accession #: GSE171592. |
| Files in database submission | GSE171592    Targeting SWI/SNF ATPases in enhancer-addicted prostate cancer Oct 28, 2021<br><br>---------------------------------------------------------------------------<br><br>GSE171584    Targeting SWI/SNF ATPases in enhancer-addicted prostate cancer [ATAC-seq]  Oct 28, 2021   approved  None<br>GSM5227748    VCaP_DMSO_R1_8h (ATAC-seq) Oct 28, 2021  approved  BED<br>GSM5227749    VCaP_DMSO_R2_8h (ATAC-seq) Oct 28, 2021  approved  BED<br>GSM5227750    VCaP_DMSO_R1_24h (ATAC-seq) Oct 28, 2021   approved  BED<br>GSM5227751    VCaP_DMSO_R2_24h (ATAC-seq) Oct 28, 2021   approved  BED<br>GSM5227752    VCaP_AU_R1_4h (ATAC-seq) Oct 28, 2021   approved  BED<br>GSM5227753    VCaP_AU_R2_4h (ATAC-seq) Oct 28, 2021   approved  BED<br>GSM5227754    VCaP_AU_R1_8h (ATAC-seq) Oct 28, 2021   approved  BED |

```
GSM5227755    VCaP_AU_R2_8h (ATAC-seq)  Oct 28, 2021   approved  BED
GSM5227756    VCaP_AU_R1_12h (ATAC-seq)  Oct 28, 2021   approved  BED
GSM5227757    VCaP_AU_R2_12h (ATAC-seq)  Oct 28, 2021   approved  BED
GSM5227758    VCaP_AU_R1_24h (ATAC-seq)  Oct 28, 2021   approved  BED
GSM5227759    VCaP_AU_R2_24h (ATAC-seq)  Oct 28, 2021   approved  BED
GSM5227760    VCaP_ZBC260_R1_8h (ATAC-seq)  Oct 28, 2021   approved  BED
GSM5227761    VCaP_ZBC260_R2_8h (ATAC-seq)  Oct 28, 2021   approved  BED
GSM5227762    VCaP_ZBC260_R1_24h (ATAC-seq)  Oct 28, 2021   approved  BED
GSM5227763    VCaP_ZBC260_R2_24h (ATAC-seq)  Oct 28, 2021   approved  BED
GSM5227764    LNCaP_DMSO_R1_24h (ATAC-seq)  Oct 28, 2021   approved  BED
GSM5227765    LNCaP_DMSO_R2_24h (ATAC-seq)  Oct 28, 2021   approved  BED
GSM5227766    LNCaP_AU_R1_12h (ATAC-seq)  Oct 28, 2021   approved  BED
GSM5227767    LNCaP_AU_R2_12h (ATAC-seq)  Oct 28, 2021   approved  BED
GSM5227768    LNCaP_AU_R1_24h (ATAC-seq)  Oct 28, 2021   approved  BED
GSM5227769    LNCaP_AU_R2_24h (ATAC-seq)  Oct 28, 2021   approved  BED
GSM5227770    LNCaP_sgNC+shNC_R1_72h (ATAC-seq)  Oct 28, 2021   approved  BED
GSM5227771    LNCaP_sgNC+shNC_R2_72h (ATAC-seq)  Oct 28, 2021   approved  BED
GSM5227772    LNCaP_sgSMARCA2_R1 (ATAC-seq)  Oct 28, 2021   approved  BED
GSM5227773    LNCaP_sgSMARCA2_R2 (ATAC-seq)  Oct 28, 2021   approved  BED
GSM5227774    LNCaP_sgSMARCA4_R1 (ATAC-seq)  Oct 28, 2021   approved  BED
GSM5227775    LNCaP_sgSMARCA4_R2 (ATAC-seq)  Oct 28, 2021   approved  BED
GSM5227776    LNCaP_sgSMARCA2+shSMARCA4_R1_72h (ATAC-seq)  Oct 28, 2021   approved  BED
GSM5227777    LNCaP_sgSMARCA2+shSMARCA4_R2_72h (ATAC-seq)  Oct 28, 2021   approved  BED
GSM5655507    AU-CBH-15330 @1uM for 0.5h-R1  Oct 28, 2021   approved  BED
GSM5655508    AU-CBH-15330 @1uM for 0.5h-R2  Oct 28, 2021   approved  BED
GSM5655509    AU-CBH-15330 @1uM for 1h-R1  Oct 28, 2021   approved  BED
GSM5655510    AU-CBH-15330 @1uM for 1h-R2  Oct 28, 2021   approved  BED
---------------------------------------------------------------------------
GSE171589     Targeting SWI/SNF ATPases in enhancer-addicted prostate cancer [ChIP-seq]  Oct 28, 2021   approved  None
GSM5228982    VCaP_DMSO_6h_AR (ChIP-seq)  Oct 28, 2021   approved  BED
GSM5228983    VCaP_AU_6h_AR (ChIP-seq)  Oct 28, 2021   approved  BED
GSM5228984    VCaP_DMSO_6h_FOXA1 (ChIP-seq)  Oct 28, 2021   approved  BED
GSM5228985    VCaP_AU_6h_FOXA1 (ChIP-seq)  Oct 28, 2021   approved  BED
GSM5228986    VCaP_DMSO_6h_ERG (ChIP-seq)  Oct 28, 2021   approved  BED
GSM5228987    VCaP_AU_6h_ERG (ChIP-seq)  Oct 28, 2021   approved  BED
GSM5228988    VCaP_DMSO_6h_CTCF (ChIP-seq)  Oct 28, 2021   approved  BED
GSM5228989    VCaP_AU_6h_CTCF (ChIP-seq)  Oct 28, 2021   approved  BED
GSM5228990    VCaP_DMSO_24h_H3K27Ac (ChIP-seq)  Oct 28, 2021   approved  BED
GSM5228991    VCaP_AU_24h_H3K27Ac (ChIP-seq)  Oct 28, 2021   approved  BED
GSM5228992    LNCaP_DMSO_6h_AR (ChIP-seq)  Oct 28, 2021   approved  BED
GSM5228993    LNCaP_AU_6h_AR (ChIP-seq)  Oct 28, 2021   approved  BED
GSM5228994    LNCaP_DMSO_6h_FOXA1 (ChIP-seq)  Oct 28, 2021   approved  BED
GSM5228995    LNCaP_AU_6h_FOXA1 (ChIP-seq)  Oct 28, 2021   approved  BED
GSM5228996    LNCaP_DMSO_6h_CTCF (ChIP-seq)  Oct 28, 2021   approved  BED
GSM5228997    LNCaP_AU_6h_CTCF (ChIP-seq)  Oct 28, 2021   approved  BED
GSM5228998    LNCaP_DMSO_24h_H3K27Ac (ChIP-seq)  Oct 28, 2021   approved  BED
GSM5228999    LNCaP_AU_24h_H3K27Ac (ChIP-seq)  Oct 28, 2021   approved  BED
GSM5655511    VCaP_AU1h_AR_Milli      Oct 28, 2021   approved  BED
GSM5655512    VCaP_AU1h_FOXA1-TFS      Oct 28, 2021   approved  BED
GSM5655513    VCaP_AU1h_H3K27Ac_abcam  Oct 28, 2021   approved  BED
GSM5655514    VCaP_AU2h_AR_Milli      Oct 28, 2021   approved  BED
GSM5655515    VCaP_AU2h_FOXA1-TFS      Oct 28, 2021   approved  BED
GSM5655516    VCaP_AU2h_H3K27Ac_abcam  Oct 28, 2021   approved  BED
GSM5655517    VCaP_AU4h_AR_Milli      Oct 28, 2021   approved  BED
GSM5655518    VCaP_AU4h_FOXA1-TFS      Oct 28, 2021   approved  BED
GSM5655519    VCaP_AU4h_H3K27Ac_abcam  Oct 28, 2021   approved  BED
---------------------------------------------------------------------------
GSE171523     Targeting SWI/SNF ATPases in enhancer-addicted prostate cancer [RNA-seq]  Oct 28, 2021   approved  None
GSM5226548    VCaP_DMSO_R1_24h (RNA-seq)  Oct 28, 2021   approved  TXT
GSM5226549    VCaP_DMSO_R2_24h (RNA-seq)  Oct 28, 2021   approved  TXT
GSM5226550    VCaP_AU_R1_4h (RNA-seq)  Oct 28, 2021   approved  TXT
GSM5226551    VCaP_AU_R2_4h (RNA-seq)  Oct 28, 2021   approved  TXT
GSM5226552    VCaP_AU_R1_8h (RNA-seq)  Oct 28, 2021   approved  TXT
GSM5226553    VCaP_AU_R2_8h (RNA-seq)  Oct 28, 2021   approved  TXT
GSM5226554    VCaP_AU_R1_12h (RNA-seq)  Oct 28, 2021   approved  TXT
GSM5226555    VCaP_AU_R2_12h (RNA-seq)  Oct 28, 2021   approved  TXT
GSM5226556    VCaP_AU_R1_24h (RNA-seq)  Oct 28, 2021   approved  TXT
GSM5226557    VCaP_AU_R2_24h (RNA-seq)  Oct 28, 2021   approved  TXT
GSM5226558    VCaP_ZBC260_R1_8h (RNA-seq)  Oct 28, 2021   approved  TXT
GSM5226559    VCaP_ZBC260_R2_8h (RNA-seq)  Oct 28, 2021   approved  TXT
GSM5226560    VCaP_ZBC260_R1_24h (RNA-seq)  Oct 28, 2021   approved  TXT
GSM5226561    VCaP_ZBC260_R2_24h (RNA-seq)  Oct 28, 2021   approved  TXT
GSM5226562    LNCaP_DMSO_R1_24h (RNA-seq)  Oct 28, 2021   approved  TXT
GSM5226563    LNCaP_DMSO_R2_24h (RNA-seq)  Oct 28, 2021   approved  TXT
GSM5226564    LNCaP_AU_R1_12h (RNA-seq)  Oct 28, 2021   approved  TXT
GSM5226565    LNCaP_AU_R2_12h (RNA-seq)  Oct 28, 2021   approved  TXT
```

GSM5226566 LNCaP_AU_R1_24h (RNA-seq) Oct 28, 2021 approved TXT
GSM5226567 LNCaP_AU_R2_24h (RNA-seq) Oct 28, 2021 approved TXT
GSM5226568 LAPC4_DMSO_R1_24h (RNA-seq) Oct 28, 2021 approved TXT
GSM5226569 LAPC4_DMSO_R2_24h (RNA-seq) Oct 28, 2021 approved TXT
GSM5226570 LAPC4_AU_R1_24h_0.1uM (RNA-seq) Oct 28, 2021 approved TXT
GSM5226571 LAPC4_AU_R2_24h_0.1uM (RNA-seq) Oct 28, 2021 approved TXT
GSM5226572 LAPC4_AU_R1_24h_1uM (RNA-seq) Oct 28, 2021 approved TXT
GSM5226573 LAPC4_AU_R2_24h_1uM (RNA-seq) Oct 28, 2021 approved TXT
GSM5655526 VCaP_DMSO_2h_1 Oct 28, 2021 approved TXT
GSM5655527 VCaP_DMSO_2h_2 Oct 28, 2021 approved TXT
GSM5655528 VCaP_AU-15330_1 uM_0.5h_1 Oct 28, 2021 approved TXT
GSM5655529 VCaP_AU-15330_1 uM_0.5h_2 Oct 28, 2021 approved TXT
GSM5655530 VCaP_AU-15330_1 uM_1h_1 Oct 28, 2021 approved TXT
GSM5655531 VCaP_AU-15330_1 uM_1h_2 Oct 28, 2021 approved TXT
GSM5655532 VCaP_AU-15330_1 uM_2h_1 Oct 28, 2021 approved TXT
GSM5655533 VCaP_AU-15330_1 uM_2h_2 Oct 28, 2021 approved TXT
--------------------------------------------------------

GSE171591 Targeting enhancer addiction in prostate cancer by impeding
chromatin accessibility [HiChIP-seq] Oct 28, 2021 approved None
GSM5229035 VCaP_DMSO_4h_H3K4me3 (HiChIP-seq) Oct 28, 2021 approved HIC
GSM5229036 VCaP_AU_4h_H3K4me4 (HiChIP-seq) Oct 28, 2021 approved HIC
GSM5229037 VCaP_DMSO_4h_H3K27Ac (HiChIP-seq) Oct 28, 2021 approved HIC
GSM5229038 VCaP_AU_4h_H3K27Ac (HiChIP-seq) Oct 28, 2021 approved HIC
GSM5229039 VCaP_DMSO_4h_CTCF (HiChIP-seq) Oct 28, 2021 approved HIC
GSM5229040 VCaP_AU_4h_CTCF (HiChIP-seq) Oct 28, 2021 approved HIC
------------------------------------------------------------------------

**Genome browser session**
(e.g. UCSC)

No longer applicable

## Methodology

**Replicates**

Multiple biological as well as technical replicates are included.

**Sequencing depth**

ATACseq: Sequenced to 65-70M total reads, paired-end mode, 125bp read lengths. Over 97% of uniquely mapped reads.
ChIPseq: Sequenced to 50-70M total reads, paired-end mode, 125bp read lengths. Over 97% of uniquely mapped reads.
RNAseq: Sequenced to 25-30M total reads, paired-end mode, 125bp read lengths. Over 97% of uniquely mapped reads.
HiChIPseq: Sequenced to 200-225M total reads, paired-end mode, 125bp read lengths. Over 95% of uniquely mapped reads.

**Antibodies**

See Supplementary Table 2.

**Peak calling parameters**

MACS2 (Version 2.1.1.20160309) callpeak was used for performing peak calling with the following option: 'macs2 callpeak–call-summits–verbose 3 -g hs -f BAM -n OUT–qvalue 0.05'. For H3K27ac data, the broad option was used.

**Data quality**

FastQC was used to quality check the raw sequencing data using standard metrics and default thresholds.

**Software**

Using deepTools (version 3.3.1) bamCoverage, a coverage file (bigWig format) for each sample was created. The coverage was calculated as the number of reads per bin, where bins are short consecutive counting windows. While creating the coverage file, the data was normalized with respect to each library size. ChIP peak profile plots and read-density heat maps were generated using deepTools, and cistrome overlap analyses were carried out using the ChIPpeakAnno (version 3.0.0)  or ChIPseeker (version 1.29.1) packages in R (version 3.6.0).

