## [Peer Review File · Nature]

Manuscript Title: Targeting SWI/SNF ATPases in enhancer-addicted prostate cancer

Reviewer Comments & Author Rebuttals

Reviewer Reports on the Initial Version:

Referee #1 (Remarks to the Author):

Xiao and colleagues develop a novel SMARCA2/SMARCA4/PBRM1 degrader with potent, rapid, and specific activity in vitro and in vivo. They observe growth inhibitory activity of AU-15330 against a number of cancer cell lines and particularly prostate cancer and multiple myeloma. In xenograft models, they observe inhibitory effects of AU-15330 as a single agent and in combination with Enzalutamide in VCaP-CRPC, C4-2B, and MDA-PCa-146-12 models of prostate cancer, with essentially no toxicity effects in normal or cancer-treated mice.

Mechanistically, they observe that AU-15330 treatment in cell lines reduces accessibility primarily at enhancers within 4 hours, with concomitant loss of transcription factor binding (6h), H3K27ac levels (24h), and H3K4me3 and H3K27ac HiChIP signal (4h). By 12 hours, there is considerable reduction in expression of AR, FOXA1, ERG, and MYC transcription factors, which are among super enhancer target genes most affected by AU-15330. Ultimately, gene expression of AR, FOXA1, and MYC target genes is affected.

This is a very nice piece of work and the strength of the effect of the degrader is exciting, as are the therapeutic implications for the first preclinical degrader of SWI/SNF complexes.

Major Points:

1) The rapid effects of the degrader agree quite well with recent data (Iurlaro Nat Genetics 2021, Schick Nat Genetics 2021) that SWI/SNF degradation with ACBi1 and BRM014 leads to incredibly rapid loss of chromatin accessibility within minutes. Xiao and colleagues go one step further to implicate loss of chromatin structure/promoter-enhancer looping within a time frame of hours. This is a potentially considerable advance because although several reports have implicated SWI/SNF complexes in enhancer maintenance and promoter-enhancer looping, it has been heretofore impossible to dissociate the effects of loss of TF binding/loss of histone modifications/loss of eRNA transcription from loss of looping. The authors are in a position to dissociate the primary and secondary effects in this manuscript. However, the HiChIP results are complicated by the potential effect of reductions in transcription factor expression, and/or loss of histone modifications within the 4h time frame examined, which could secondarily contribute to loss of enhancer architecture. The authors should perform AR, FOXA1, H3K4me3, and H3K27ac ChIP-seq at 4h to correspond to the HiChIP data taken at 4h to determine if accessibility is a direct effector of looping or involves these other factors. A more conclusive experiment for a direct effect of SWI/SNF remodeling on promoter-enhancer looping would be to perform TF/histone mod ChIPs and HiChIP at shorter time points, for example 1h, when there is no observable decline in transcription factor expression as shown in Figure 3b. In my opinion, this would be a significant mechanistic advance in addition to the translational data.

2) Given that several degraders are emerging, comparative data would be very useful. The authors should perform cell line lethality as in Figure 1e with ACBi1 that is commercially available and any other SMARCA4/A2/PBRM1 compounds that may be generally available (I am not aware that BRM014 is available). The comparative activity and related efficacy against prostate and multiple myeloma are important to know.

Referee #2 (Remarks to the Author):

As there is no durable cure for advanced prostate cancer, developing strategies to target aggressive prostate cancers is of significant clinical relevance. Given the known reliance of most prostate cancers on androgen receptor (AR signaling), interventions that enhance the ability of current AR-targeting agents to more effectively and/or durably suppress AR function could provide an enhanced anti-tumor effect. It has been previously well established that AR requires SWI/SNF activity for function, and that suppression of SWI/SNF activity in cell lines ablates AR activity.

The present study represents a major translational breakthrough, manifest through development of PROTACs against both central ATPases of SWI/SNF complexes (Brm and Brg1). Remarkably, these degraders show specificity for cytotoxicity in tumors driven by oncogenic transcription factors, with a particular efficacy for AR+ prostate cancer and multiple myeloma. Mechanistic studies reveal that inhibition of Brm/Brg1 disrupts chromatin accessibility at enhancers, and enhancer-promoter loops. These studies provide fresh new insight into the requirement for SWI/SNF activity in prostate cancer. Subsequent studies demonstrate putative translational relevance, wherein the degrader inhibited tumor growth *in vivo* and synergized with AR-targeted therapies. On balance, this is an elegant, well-performed and impactful study that is the first to nominate SWI/SNF as a feasible therapeutic target in prostate cancer. There are only minor considerations that the authors may wish to consider:

1. Understanding mechanisms of resistance would further enhance understanding of how the agent elicits anti-tumor effects. If resistance models have been generated and/or understood, inclusion of those data would be of relevance.
2. It would be of relevance to determine the impact of the degrader on hi-Myc prostate cancers, including NEPC.
3. Given the mechanism of action, what is the impact of sequencing AR vs. SWI/SNF targeted agents?

Referee #3 (Remarks to the Author):

The authors develop a PROTAC (AU-15330) capable of the degradation of SMARCA2, SMARCA4 and PBRM1 via functionalization of a previously described bromodomain ligand. They demonstrate that the compound is selective for the 3 bromodomain targets, before profiling the antiproliferative effect of AU-15330 against a panel of cell lines. Interestingly, AR/FOXA1+ prostate cancer cells were among the most susceptible to AU-15330. Detailed characterization of the antiproliferative effect of AU-15330 in AR/FOXA1+ prostate cancer cell lines revealed chromatin compaction at specific sites for oncogenic transcription factors, thus providing a hypothesis for the selectivity of this compound. CHIP-seq was used to confirm loss of AR and FOXA1 transcription factor binding to chromatin and RNA-seq revealed a down regulation of genes regulated by these transcription factors. AU-15330 treatment also resulted in reduction of AR and FOXA1 expression, as well as MYC and TMPRSS2-ERG, confirmed at both RNA and protein levels. The authors demonstrate AU-15330 treatment disrupts looping interactions at enhancers of AR gene expression. The authors progress their lead compound into *in vivo* experiments where they demonstrate reasonable PK properties and degradation activity with no noticeable toxicity. AU-15330 alone demonstrated moderate antitumor effect as a single agent (albeit at relatively high dose) but is synergistic with enzalutamide across 2 *in vivo* models. Additionally, AU-15330 slows tumor growth in an enzalutamide resistant model.

Whilst intriguing and suggesting a hypothesis which demands detailed characterization, this study is lacking in controls and therefore the conclusions drawn are not fully supported by the data. This manuscript also lacks the novelty required for publication in Nature since this structurally very similar PROTACs for these targets were reported and characterized by Ciulli and co-workers in 2019 and previous reports also detail the importance of these ATPases in controlling chromatin

accessibility.

The biggest issue with this manuscript, is that the authors repeatedly refer to inhibition of ATPase activity when in fact they mean degradation of ATPase protein – these are not necessarily the same thing and in this case are unlikely to phenocopy one another. If inhibition alone was sufficient to achieve this response, then why use the PROTAC rather than simply the small molecule bromodomain ligand that it is derived from? The authors need to include additional controls, such as a PROTAC with a crippled E3 ligase binder as is common in the field to differentiate between bromodomain ligand binding and protein degradation. For VHL PROTACs, this is usually a diastereomer control which does not bind VHL. Key experiments should be repeated directly comparing AU-15330 to its diastereomer control.

Additionally, there are a set of control experiments considered standard in reporting new PROTACs which are missing in this study, namely co-treatment with a proteasome inhibitor and MLN4924 to confirm on mechanism loss of target protein and quantitation of RNA levels to demonstrate that loss of protein occurs post-translationally.

The authors state that competition with excess VHL ligand is sufficient to inhibit PROTAC mediated degradation of the ATPases but this is not true in VCaP cells, as shown in extended data figure 2D, bringing into question the mechanism of degradation.

The authors attempted to phenocopy the effect of AU-15330 by dual knock out of SMARCA2 and SMARCA4 but the ATAC-seq data in Extended Data Fig. 3 shows incomplete recapitulation of the chromatin compaction phenotype, therefore suggesting that that degradation of PBRM1, inhibition of the ATPase bromodomains or an off-target effect may be somewhat responsible for the AU-15330 phenotype. The use of the diastereomer control would also help elucidate this. Throughout the manuscript various other PROTACs are employed as controls without demonstrating that they are functioning as expected. Immunoblots should be included to demonstrate loss of BRD4 and BRD7/9 as appropriate. Additionally, the use of these compounds should be discussed in terms of degradation, not inhibition.

In extended data Fig. 1 the authors refer to “dead analog” which is not mentioned anywhere else in the manuscript. This may be the diastereomer control referred too above but this must be explained and the structure shown. If it is this key control compound, then it should have been included in many, if not all, of the other assays.

The structure of AU-15330 contains a very short linker for a PROTAC which is intriguing, the manuscript should contain some details of how this compound was arrived at, as well as its structure activity relationships.

Author Rebuttals to Initial Comments:

Referee #1 (Remarks to the Author):

Xiao and colleagues develop a novel SMARCA2/SMARCA4/PBRM1 degrader with potent, rapid, and specific activity in vitro and in vivo. They observe growth inhibitory activity of AU-15330 against a number of cancer cell lines and particularly prostate cancer and multiple myeloma. In xenograft models, they observe inhibitory effects of AU-15330 as a single agent and in combination with Enzalutamide in VCaP-CRPC, C4-2B, and MDA-PCa-146-12 models of prostate cancer, with essentially no toxicity effects in normal or cancer-treated mice.

Mechanistically, they observe that AU-15330 treatment in cell lines reduces accessibility primarily at enhancers within 4 hours, with concomitant loss of transcription factor binding (6h), H3K27ac levels

(24h), and H3K4me3 and H3K27ac HiChIP signal (4h). By 12 hours, there is considerable reduction in expression of AR, FOXA1, ERG, and MYC transcription factors, which are among super enhancer target genes most affected by AU-15330. Ultimately, gene expression of AR, FOXA1, and MYC target genes is affected.

This is a very nice piece of work and the strength of the effect of the degrader is exciting, as are the therapeutic implications for the first preclinical degrader of SWI/SNF complexes.

Response: We thank the reviewer for such a positive appraisal of our work and for highlighting the immediate translational value of novel SWI/SNF therapeutics. We are particularly grateful for insightful mechanistic questions that have significantly improved the quality of this study. All major points have been addressed in a point-wise fashion below as well as related changes made to the manuscript are highlighted in yellow:

Major Points:

1) The rapid effects of the degrader agree quite well with recent data (Iurlaro Nat Genetics 2021, Schick Nat Genetics 2021) that SWI/SNF degradation with ACBi1 and BRM014 leads to incredibly rapid loss of chromatin accessibility within minutes. Xiao and colleagues go one step further to implicate loss of chromatin structure/promoter-enhancer looping within a time frame of hours. This is a potentially considerable advance because although several reports have implicated SWI/SNF complexes in enhancer maintenance and promoter-enhancer looping, it has been heretofore impossible to dissociate the effects of loss of TF binding/loss of histone modifications/loss of eRNA transcription from loss of looping. The authors are in a position to dissociate the primary and secondary effects in this manuscript. However, the HiChIP results are complicated by the potential effect of reductions in transcription factor expression, and/or loss of histone modifications within the 4h time frame examined, which could secondarily contribute to loss of enhancer architecture. The authors should perform AR, FOXA1, H3K4me3, and H3K27ac ChIP-seq at 4h to correspond to the HiChIP data taken at 4h to determine if accessibility is a direct effector of looping or involves these other factors. A more conclusive experiment for a direct effect of SWI/SNF remodeling on promoter-enhancer looping would be to perform TF/histone mod ChIPs and HiChIP at shorter time points, for example 1h, when there is no observable decline in transcription factor expression as shown in Figure 3b. In my opinion, this would be a significant mechanistic advance in addition to the translational data.

Response: We thank the reviewer for this extremely insightful comment. As requested, we have performed ATACseq, ChIPseq, and HiChIPseq assays in a time course (including early time points) to distinguish the primary and the secondary effects of complete SWI/SNF ATPase inactivation. Notably, treatment with 1 μ M of AU-15330 triggered degradation of all detectable SMARCA2/4 proteins within an hour in VCaP cells (**Extended Data Fig 3a**). At this time point, we detected an instantaneous loss in chromatin accessibility, almost as striking as the loss at 4h after AU-15330 treatment (**Fig 2a**). In ChIPseq assays, we also detected a significant decrease in the binding of both AR and FOXA1 at the AU-15330-compacted enhancer sites as early as an hour after AU-15330 treatment (**Extended Data Fig 2d**).

Figure 2

Figure 2. SWI/SNF ATPase inhibition disrupts physical chromatin accessibility at the core-enhancer circuitry to disable oncogenic transcriptional programs. (a) ATAC-seq read-density heat maps from VCaP cells treated with DMSO (solvent control), AU-15330, or ZBC-260 (a BRD4 degrader) for indicated durations (n=2 biological replicates per condition). (b) Genome-wide changes in chromatin accessibility upon AU-15330 treatment for 4 h in VCaP cells along with genomic annotation of sites that lose physical accessibility (i.e. lost) or remain unaltered (i.e. retained). (c) Top ten *de novo* motifs (ranked by p-value) enriched within AU-15330-compacted genomic sites (HOMER, hypergeometric test) in VCaP cells. (d) ChIP-seq read-density heat maps for AR and FOXA1 at the AU-15330-compacted genomic sites in VCaP cells after treatment with DMSO (solvent control) or AU-15330 (at 1 μ M) for indicated time durations and stimulation with R1881, a synthetic androgen (at 1 nM for 3 h).

Most remarkably, within one hour, H3K27Ac ChIPseq signal showed the disappearance of the characteristic “valley” centered at the enhancer sites (**Extended Data Fig 2e**; also copied below as **Fig. 2**)—formed due to transcription factor binding—without any loss in the overall abundance of this active enhancer-associated histone mark at early (<4h) time points (also shown in reviewer Figure 2 below). This implies that the eventual loss of H3K27Ac detected at the 24h time point is a secondary effect and is downstream of immediate nucleosomal compaction of enhancers that happens within minutes of SMARCA2/4 degradation. Next, H3K4me3 HiChIPseq assay revealed a significant loss in *cis*-contacts of active gene promoters starting at the 2h time point after AU-15330 treatment (i.e., within an hour of SMARCA2/4 degradation, **Extended Data Fig 8e**), while we detected modest to no change at the 1h time point. This suggests the loss in distal enhancer-promoter contacts are triggered by nucleosomal compaction and transcription factor unloading at the enhancer sites.

Figure 2: SWI/SNF ATPase inhibition disrupts physical chromatin accessibility at the core-enhancer circuitry to disable oncogenic transcriptional programs. (e) ChIP-seq read-density heat maps for H3K27Ac at the AU-15330-compact genomic sites in VCaP cells after treatment with DMSO (solvent control) or AU-15330 (at 1 μ M) for indicated time durations and stimulation with R1881, a synthetic androgen (at 1 nM for 3 h).

Thus, we propose a coalesced mechanistic model wherein concurrent degradation of both SWI/SNF ATPases triggers a cascade of events starting from instantaneous nucleosomal compaction of enhancer elements with a parallel loss in transcription factor binding (within minutes), which in turn disrupts the looping interaction of enhancers with target gene promoters (within an hour). Notably, our data also suggest that the presence of the H3K27Ac mark alone isn't sufficient to retain the three-dimensional enhancer architecture. This positions continual ATP-dependent displacement of nucleosomes by SWI/SNF complexes and transcription factor binding at enhancer sites as primary determinants of looping enhancer-promoter interactions.

2) Given that several degraders are emerging, comparative data would be very useful. The authors should perform cell line lethality as in Figure 1e with ACBI1 that is commercially available and any other SMARCA4/A2/PBRM1 compounds that may be generally available (I am not aware that BRM014 is available). The comparative activity and related efficacy against prostate and multiple myeloma are important to know.

Response: We thank the reviewer for this suggestion. As requested, we have now performed growth assays with both ACBI1 (a SMARCA2/4 degrader) and BRM014 (a SMARCA2/4 ATPase inhibitor) compounds in a select panel of normal, prostate cancer, and multiple myeloma cell lines (**Extended Data Fig 1f**). Here, consistent with our findings with AU-15330, both SMARCA2/4-targeted compounds showed preferential cytotoxicity in cancer cell lines, while having no growth inhibitory effect in normal/non-neoplastic cells (e.g., HEK293, PNT2, and BPH1). However, the half-maximal inhibitory concentrations (IC50) for both ACBI1 and BRM014 were higher compared to AU-15330 (ranging from 1.2-16 fold) in all of the tested cancer cell lines. As expected, none of the compounds had any growth

inhibitory effect in the BIN67 cell line (IC₅₀ > 10μM for all compounds)—a cellular model of small cell carcinoma of the ovary, hypercalcemic type—that endogenously harbors deleterious mutations in both the SMARCA2 and SMARCA4 genes¹.

Next, a comparison of target protein degradation and/or downstream inhibition of the gene expression revealed AU-15330 to perform slightly better than the ACB11 PROTAC, triggering both faster degradation of SMARCA2/4 and PBRM1 proteins as well as a stronger downstream inhibition of AR targets (e.g. PSA and ERG) at equimolar concentrations (**Extended Data Fig 1g**). Notably, BRM014 induced the least reduction in the expression of MYC and AR target genes (particularly ERG) at matched dosages and treatment duration. Consistently, BRM014 had the highest IC-50 values of all the three compounds in prostate cancer as well as multiple myeloma cell lines. Notably, ACB11 is yet to be tested in animal models while the seminal study on BRM014 reported dose-limiting toxicities with severe body weight loss²—none of which were noted in our *in vivo* efficacy studies with AU-15330 in mice (**Extended Data Fig 9, 10, 11c, and 12e,f**)

Referee #2 (Remarks to the Author):

As there is no durable cure for advanced prostate cancer, developing strategies to target aggressive prostate cancers is of significant clinical relevance. Given the known reliance of most prostate cancers on androgen receptor (AR signaling), interventions that enhance the ability of current AR-targeting agents to more effectively and/or durably suppress AR function could provide an enhanced anti-tumor effect. It has been previously well established that AR requires SWI/SNF activity for function, and that suppression of SWI/SNF activity in cell lines ablates AR activity.

The present study represents a major translational breakthrough, manifest through development of PROTACs against both central ATPases of SWI/SNF complexes (Brm and Brg1). Remarkably, these degraders show specificity for cytotoxicity in tumors driven by oncogenic transcription factors, with a particular efficacy for AR+ prostate cancer and multiple myeloma. Mechanistic studies reveal that inhibition of Brm/Brg1 disrupts chromatin accessibility at enhancers, and enhancer-promoter loops. These studies provide fresh new insight into the requirement for SWI/SNF activity in prostate cancer. Subsequent studies demonstrate putative translational relevance, wherein the degrader inhibited tumor growth *in vivo* and synergized with AR-targeted therapies. On balance, this is an elegant, well-performed and impactful study that is the first to nominate SWI/SNF as a feasible therapeutic target in prostate cancer. There are only minor considerations that the authors may wish to consider:

Response: We are grateful to the reviewer for highlighting the immediate translational impact of our work and recognizing this study being the first preclinical proof-of-concept for targeting the SWI/SNF complex in AR/FOXA1-dependent prostate cancer. We have addressed all of the comments in the pointwise responses below as well as related changes made to the manuscript are highlighted in yellow:

1. Understanding mechanisms of resistance would further enhance understanding of how the agent elicits anti-tumor effects. If resistance models have been generated and/or understood, inclusion of those data would be of relevance.

[Redacted]

[Redacted Text and Reviewer Figure R1]

2. It would be of relevance to determine the impact of the degrader on hi-Myc prostate cancers, including NEPC.

Response: As suggested, we have now tested the efficacy of AU-15330 in the NCI-H660 cell line that is widely considered as a model of neuroendocrine prostate cancer. Despite triggering marked degradation of both SMARCA2 and SMARCA4 ATPases, AU-15330 did not affect the growth and survival of the NCI-H660 cells (IC₅₀ = 6.8 μ M). Additional *in vitro* and *in vivo* models of neuroendocrine prostate cancer will be studied in more detail in follow-up studies.

Fig. R2: **Left**, Immunoblots for SMARCA2 (BRM), SMARCA4 (BRG1), and PBRM1 proteins from whole-cell lysates upon treatment of NCI-H660 with AU15330 at 1 μ M. **Middle**, Dose-response curve of NCI-H660 and VCaP cells treated with AU-15330. **Right**, Summary of the IC50 value of AU-15330 in NCI-H660 and VCaP cells. Data are shown as mean \pm SE (n = 6) from one of two independent experiments.

To evaluate the correlation between *MYC* expression and sensitivity to AU-15330, we extracted *MYC* gene expression levels in prostate cancer cells from the CCLE dataset. Here, of the tested cell lines, we found a strong inverse correlation between the level of *MYC* expression and IC50 concentrations of AU-15330 (**Fig. R3**). In other words, hi-*MYC* prostate cancer cell lines (namely VCaP, LNCaP, and 22RV1) seem to be acutely sensitive (IC50 < 20 nM for all) to SWI/SNF ATPase degradation. However, we anticipate this trend to hold only in cancers where supra-physiologic expression of *MYC* is wired through aberrant activation of neo/super-enhancers.

Fig. R3: Scatter plot depicting the correlation between levels of *MYC* expression and IC50 values of AU-15330 in a panel of prostate cancer cell lines.

3. Given the mechanism of action, what is the impact of sequencing AR vs. SWI/SNF targeted agents?

Response: To address this question, we pre-treated prostate cancer cells with either enzalutamide or AU-15330 at sublethal dosages for 24h, followed by an assessment of 5-day IC50 values for the other compound. Here, we found both pre-treatment with enzalutamide or AU-15330 to reciprocally and significantly reduce the IC50 value of either drug in a dose-dependent manner (**Extended Data Fig 11h, i**). Thus, the synergistic effect seen in the combination regimen may not be dependent on the sequence in which the drugs are administered. This is consistent with both compounds having a

distinct mechanism of action, with enzalutamide inhibiting AR activity via a direct antagonistic mechanism while AU-15330 compacting AR's genomic sites and, thereby, dislodging it from the chromatin. However, in a clinical setting, we envision the treatment of patients with AU-15330 to most likely follow an ongoing or prior regimen of enzalutamide.

Referee #3 (Remarks to the Author):

The authors develop a PROTAC (AU-15330) capable of the degradation of SMARCA2, SMARCA4 and PBRM1 via functionalization of a previously described bromodomain ligand. They demonstrate that the compound is selective for the 3 bromodomain targets, before profiling the antiproliferative effect of AU-15330 against a panel of cell lines. Interestingly, AR/FOXA1+ prostate cancer cells were among the most susceptible to AU-15330. Detailed characterization of the antiproliferative effect of AU-15330 in AR/FOXA1+ prostate cancer cell lines revealed chromatin compaction at specific sites for oncogenic transcription factors, thus providing a hypothesis for the selectivity of this compound. CHIP-seq was used to confirm loss of AR and FOXA1 transcription factor binding to chromatin and RNA-seq revealed a down regulation of genes regulated by these transcription factors. AU-15330 treatment also resulted in reduction of AR and FOXA1 expression, as well as MYC and TMPRSS2-ERG, confirmed at both RNA and protein levels. The authors demonstrate AU-15330 treatment disrupts looping interactions at enhancers of AR gene expression. The authors progress their lead compound into in vivo experiments where they demonstrate reasonable PK properties and degradation activity with no noticeable toxicity. AU-15330 alone demonstrated moderate antitumor effect as a single agent (albeit at relatively high dose) but is synergistic with enzalutamide across 2 in vivo models. Additionally, AU-15330 slows tumor growth in an enzalutamide resistant model.

Whilst intriguing and suggesting a hypothesis which demands detailed characterization, this study is lacking in controls and therefore the conclusions drawn are not fully supported by the data. This manuscript also lacks the novelty required for publication in Nature since this structurally very similar PROTACs for these targets were reported and characterized by Ciulli and co-workers in 2019 and previous reports also detail the importance of these ATPases in controlling chromatin accessibility.

The biggest issue with this manuscript, is that the authors repeatedly refer to inhibition of ATPase activity when in fact they mean degradation of ATPase protein – these are not necessarily the same thing and in this case are unlikely to phenocopy one another. If inhibition alone was sufficient to achieve this response, then why use the PROTAC rather than simply the small molecule bromodomain ligand that it is derived from? The authors need to include additional controls, such as a PROTAC with a crippled E3 ligase binder as is common in the field to differentiate between bromodomain ligand binding and protein degradation. For VHL PROTACs, this is usually a diastereomer control which does not bind VHL. Key experiments should be repeated directly comparing AU-15330 to its diastereomer control.

Response: We thank the reviewer for this very meaningful suggestion. As requested, we have now added key data using the parent bromodomain ligand (named AU-15139) as well as an epimer of the AU-15330 PROTAC compound (named AU-16235; earlier referred to as the “dead analog” in the paper) as important controls (**Extended Data Fig 2d**). Notably, as expected and unlike AU-15330,

treatment with AU-15139 did not induce any detectable degradation of the SWI/SNF target proteins. More impressively, treatment with the AU-16235 diastereomer control, which differs from AU-15330 at only a single chiral center harboring a key hydroxyl group in the VHL-binding moiety, also triggered no degradation of the SMARCA2/4 and PBRM1 proteins (**Extended Data Fig 2d**). Consistent with these results, neither of the control compounds induced cell death in any of the prostate cancer cell lines—even at dosages as high as 10 μ M—which are all super-sensitive to the AU-15330 compound (IC₅₀<100nM, **Extended Data Fig 2e**). This strongly suggests that the cytotoxicity of AU-15330 noted in enhancer-driven cancers is majorly attributable to its on-target degradation of the SMARCA2/4 and PBRM1 subunits of the SWI/SNF complex.

Also, we have revised the wording throughout the manuscript to eliminate any ambiguity towards AU-15330 being a potent degrader of the SMARCA2/4 ATPases, and thereby inactivating the ability of the SWI/SNF complex to displace or eject nucleosomes. All related changes made to the manuscript are highlighted in yellow in the revised document.

Additionally, there are a set of control experiments considered standard in reporting new PROTACs which are missing in this study, namely co-treatment with a proteasome inhibitor and MLN4924 to confirm on mechanism loss of target protein and quantitation of RNA levels to demonstrate that loss of protein occurs post-translationally.

Response: We have now added these control experiments. As suggested by the reviewer, we treated VCaP cells with Bortezomib (proteasome inhibitor) or MLN4924 (NEDD8-activating enzyme inhibitor⁵) for an hour prior to the treatment with AU-15330 at varied concentrations (**Extended Data Fig 2f**). Here, pre-treatment with both compounds blocked target protein degradation in a dose-dependent manner, indicating that AU-15330 depends on the proteasome and ubiquitination cascade (i.e., neddylated CUL2) for its action.

Addressing the second point, treatment with 1 μ M of AU-15330 induces almost complete loss of SMARCA2/4 and PBRM1 proteins within an hour of treatment (**Fig. R4, left**). In matched short-term treated samples, we did not detect any loss in the mRNA expression of these genes (**Fig. R4, right**). This suggests that AU-15330-mediated degradation of its target proteins occurs post-translationally without inducing any changes in the mRNA expression. All of the above control data has now been added to the revised manuscript and we thank the reviewer for this very valuable feedback.

Fig R4: **Left**, Immunoblots showing degradation of direct target proteins of AU-15330 at short treatment timepoints using 1uM of the drug in VCaP cells. **Right**, mRNA expression of AU-15330 targets in short-term treated VCaP cells.

The authors state that competition with excess VHL ligand is sufficient to inhibit PROTAC mediated degradation of the ATPases but this is not true in VCaP cells, as shown in extended data figure 2D, bringing into question the mechanism of degradation.

Response: We thank the reviewer for raising this concern. In the original rescue experiment, we had simultaneously treated VCaP cells with the VHL ligand and the AU-15330 compound. We have now repeated this experiment by pre-treating VCaP cells with the VHL ligand for an hour and then treating them AU-15330 at increasing dosages. Here, in both VCaP and LNCaP cells, we found treatment with the free VHL ligand to hinder AU-15330-mediated target protein degradation in a dose-dependent manner (**Extended Data Fig 2f**). In both cell lines, 100-1000 fold molar excess of the VHL ligand fully rescued degradation of the target proteins upon subsequent treatment with AU-15330. Notably, a similarly designed experiment involving pre-treating the cells with thalidomide (a ligand for Cereblon that functions as a substrate recognition subunit of an E3 ubiquitin ligase) had no inhibiting effect on AU-15330-triggered protein degradation at any of the tested concentrations (**Extended Data Fig 2f**)—further corroborating the VHL-dependent mechanism of protein degradation for our PROTAC compound.

The authors attempted to phenocopy the effect of AU-15330 by dual knock out of SMARCA2 and SMARCA4 but the ATAC-seq data in Extended Data Fig. 3 shows incomplete recapitulation of the chromatin compaction phenotype, therefore suggesting that that degradation of PBRM1, inhibition of the ATPase bromodomains or an off-target effect may be somewhat responsible for the AU-15330 phenotype. The use of the diastereomer control would also help elucidate this.

Response: Despite repeated attempts, we were unable to establish prostate cancer cell lines with a stable and dual knock-out of both SMARCA2 and SMARCA4 proteins. Thus, we had to employ a conditional knockdown strategy, using a doxycycline-inducible short-hairpin RNA, in combination with a single ATPase knockout using the CRISPR/Cas9 system, as shown in the schema in **Extended Data Fig 3c**. Here, we were able to only partially knock down the SMARCA4 protein that most likely explains the partial loss in chromatin accessibility (**Extended Data Fig 3d**).

Furthermore, a recently published study used a combination of CRISPR-knockout and degron systems in HAP1 cells to clearly demonstrate that loss of both SWI/SNF ATPases is indeed sufficient for an almost complete loss in chromatin accessibility⁶. Altogether, this suggests that dual SWI/SNF ATPase inactivation primarily underlies the loss in chromatin accessibility and does not require parallel degradation of PBRM1.

Throughout the manuscript various other PROTACs are employed as controls without demonstrating that they are functioning as expected. Immunoblots should be included to demonstrate loss of BRD4 and BRD7/9 as appropriate. Additionally, the use of these compounds should be discussed in terms of degradation, not inhibition.

Response: We thank the reviewer for this suggestion. We have now added immunoblots confirming the degradation of target protein(s) for each PROTAC compound at the dosages used in our assays (**Extended Data Fig 3a, 7e**). As suggested, we have also revised the wording in the manuscript to

discuss our findings in terms of complete loss of the target proteins and not just their functional inhibition.

In extended data Fig. 1 the authors refer to “dead analog” which is not mentioned anywhere else in the manuscript. This may be the diastereomer control referred too above but this must be explained and the structure shown. If it is this key control compound, then it should have been included in many, if not all, of the other assays.

Response: Yes, the “dead analog” is indeed the epimer/diastereomer control of the AU-15330 PROTAC compound. As requested, we have included the structure of this compound and discussed it appropriately in the revised manuscript. As shown, the epimer control (termed AU-16235) differs from AU-15330 at only a single chiral center harboring a key hydroxyl group in the VHL-binding moiety (**Fig. R5**). Accordingly, treatment of prostate cancer cells with AU-16235 neither triggered degradation of the SMARCA2/4 and PBRM1 proteins nor did it induce cell death in cancer cell lines (**Extended Data Fig. 3d, e**).

Fig. R5: Chemical structure of the AU-15330 PROTAC compound and its epimer control, AU-16235.

The structure of AU-15330 contains a very short linker for a PROTAC which is intriguing, the manuscript should contain some details of how this compound was arrived at, as well as its structure activity relationships.

Response: AU-15330 was designed using Aurigene’s in-house computing algorithm called ALMOND (ALgorithm for MOdeling Neosubstrate Degraders) that employs both protein-protein as well as small molecule protein docking simulations along with exhaustive conformational sampling and scoring. Intriguingly, this approach is currently limited to designing degraders with very short or no linkers. AU-15330 was initially developed more as a tool compound for preliminary evaluations, but its remarkable target specificity and degradation potency has triggered a focused effort in our group to study the use of short linkers in PROTAC designs. We hope to report our findings in future studies.

References:

1. Pan, J. *et al.* The ATPase module of mammalian SWI/SNF family complexes mediates subcomplex identity and catalytic activity-independent genomic targeting. *Nat. Genet.* **51**, 618–626 (2019).

2. Papillon, J. P. N. *et al.* Discovery of Orally Active Inhibitors of Brahma Homolog (BRM)/SMARCA2 ATPase Activity for the Treatment of Brahma Related Gene 1 (BRG1)/SMARCA4-Mutant Cancers. *J. Med. Chem.* **61**, 10155–10172 (2018).
3. Farnaby, W. *et al.* BAF complex vulnerabilities in cancer demonstrated via structure-based PROTAC design. *Nat. Chem. Biol.* **15**, 672–680 (2019).
4. Kregel, S. *et al.* Androgen receptor degraders overcome common resistance mechanisms developed during prostate cancer treatment. *Neoplasia* **22**, 111–119 (2020).
5. Soucy, T. A. *et al.* An inhibitor of NEDD8-activating enzyme as a new approach to treat cancer. *Nature* **458**, 732–736 (2009).
6. Schick, S. *et al.* Acute BAF perturbation causes immediate changes in chromatin accessibility. *Nat. Genet.* **53**, 269–278 (2021).

Reviewer Reports on the First Revision:

Referee #1 (Remarks to the Author):

The authors have addressed my concerns and provided multiple new data at early timepoints showing kinetically resolved activity of SWI/SNF in maintaining chromatin accessibility, TF binding, and chromatin looping followed by losses in H3K27ac, which is quite exciting, as well as superior activity of AU-15330 over ACBI1 and BRM014 in cancer cell lines.

Referee #2 (Remarks to the Author):

The authors have meticulously addressed the major concerns of the study. The findings significantly advance understanding of the molecular underpinnings of prostate cancer, and will be of broad interest to the transcriptional regulation, hormone action, and cancer fields.

Referee #3 (Remarks to the Author):

The authors have sufficiently addressed the reviewers comments in most cases however there are a few additional revisions required.

The resistance experiments (Fig. R1) are important and intriguing and should therefore be included in the manuscript rather than just in response to reviewers.

Control experiments in Fig. R4 should also be added to the manuscript.

Due to the incredibly short nature of the PROTAC linker, I maintain that some optimization/structure activity relationships must be included in the manuscript. Stating that a proprietary algorithm was used is not sufficient and other analogs must be included in the manuscript prior to publication.

Curve fitting throughout the manuscript is poor, leading to inaccurate IC50 values, and should be repeated with more suitable models.

Eg. Fig. R1 A – it appears that not only is the IC50 increased but the maximum response decreased.

Fig. R2.

Extended Data 1D

Extended Data 2E

Author Rebuttals to First Revision:

Referee #1 (Remarks to the Author):

The authors have addressed my concerns and provided multiple new data at early timepoints showing kinetically resolved activity of SWI/SNF in maintaining chromatin accessibility, TF binding, and chromatin looping followed by losses in H3K27ac, which is quite exciting, as well as superior activity of AU-15330 over ACBI1 and BRM014 in cancer cell lines.

Response: We thank the reviewer for all of the insightful questions and feedback that have significantly improved the quality and impact of our manuscript. We wholeheartedly share the reviewer's excitement and look forward to the publication of our findings.

Referee #2 (Remarks to the Author):

The authors have meticulously addressed the major concerns of the study. The findings significantly advance understanding of the molecular underpinnings of prostate cancer and will be of broad interest to the transcriptional regulation, hormone action, and cancer fields.

Response: We thank the reviewer for all of the meaningful suggestions and for appraising our work to be of great interest to a broad scientific community. We eagerly look forward to the timely publication of our manuscript.

Referee #3 (Remarks to the Author):

The authors have sufficiently addressed the comments in most cases; however, there are a few additional revisions required.

Response: We sincerely thank the reviewer for recommending various control experiments. These have significantly improved the quality of our study.

The resistance experiments (Fig. R1) are important and intriguing and should therefore be included in the manuscript rather than just in response to reviewers.

Response: We agree with the reviewer that the study of resistance mechanisms to AU-15330 is important and intriguing; however, we believe the data presented in reviewer Fig. R1 to be very preliminary (i.e., a single resistance clone), still requiring a series of mechanistic

experiments and validation as well as additional independent clones. Thus, we propose this preliminary data not be included in our present manuscript that primarily focuses on AU-15330's anti-tumor efficacy (and not resistance mechanisms).

Control experiments in Fig. R4 should also be added to the manuscript.

Response: We have incorporated these results in the revised manuscript (**Extended Data Figure 3a**).

Due to the incredibly short nature of the PROTAC linker, I maintain that some optimization/structure activity relationships must be included in the manuscript. Stating that a proprietary algorithm was used is not sufficient, and other analogs must be included in the manuscript prior to publication.

Response: We thank the reviewer for this suggestion. We have now included a description of our molecular modelling strategy for designing AU-15330 in the Methods section. In brief, the binding model of AU-15330 in complex with SMARCA2-BD and VHL was generated using Aurigene's proprietary computing algorithm, ALMOND (ALgorithm for MOdeling Neosubstrate Degraders) (DOI: 10.31031/MADD.2021.03.000560). The algorithm was developed using the ICM-Pro integrated modelling platform (http://www.molsoft.com/icm_pro.html) and trained to predict models of ternary complexes of bi-functional molecules with very short or no linkers. The process employs protein-protein docking simulation, exhaustive conformational sampling, small molecule-protein docking, and site-directed scoring of predicted ternary complex models. MF Score (mean force score) provides an independent score of the strength of ligand-receptor interaction, which signifies the strength of the induced protein-protein interaction in the target – E3 ligase complex, and a lower score suggests stronger binding. A cut off score of less than -250 using this algorithm was considered for prioritization. Validation of this hypothesis has been exemplified in the following table (**Table 1**), which demonstrates AU-15330 outperforms other compounds in terms of SMARCA2-BD degradation.

As the data suggests, AU-15530, which has the shortest linker, is the most potent degrader, achieving 90% degradation of SMARCA2 at 10 nM. Incrementally increasing the size of the linker in Compound 2 markedly affects the degradation efficiency (only 22% degradation at 10 nM). When the linker is extended even further as in Compound 3, no degradation of SMARCA2 is observed.

Compound Code	Structure	MF Score	% Degradation of SMARCA2 at 10 nM (H838 cells)
AU-15330		-291.1	90%
Compound 2		-278.4	22%
Compound 3		-221.6	No degradation

Table 1: SMARCA2/4 PROTAC linker-derivatives and target degradation efficiency. MF score stands for potential of mean force score, which provides an independent score of the strength of ligand-receptor interaction, with a lower score suggesting a stronger binding. All the above tested compounds differ only in their linker structure and length.

Curve fitting throughout the manuscript is poor, leading to inaccurate IC₅₀ values, and should be repeated with more suitable models.

Fig. R1 A – it appears that not only is the IC₅₀ increased but the maximum response decreased.

Fig. R2.

Extended Data 1D

Extended Data 2E

Response: We thank the reviewer for this suggestion. We have now used an improved model (4PL sigmoidal standard curve model) for curve fitting of the IC₅₀ plots and calculation of the IC₅₀ values. In **Figure R1** below, we present the side-by-side comparison of the previous IC₅₀ plots (**Figure R1a**) and the revised IC₅₀ plots (**Figure R1b, Extended Data Figure 1f**) for one example. Importantly, we found that the absolute IC₅₀ values changed only slightly with the new model and did not affect the conclusions previously made. In the revised manuscript, all IC₅₀ plots have been updated. In addition, we will be providing the raw dosage–cell viability data as part of the source data series, which can be used by researchers to fit other IC₅₀ curve fitting models as appropriate.

Figure R1. A comparison between previous (a) and current (b) IC₅₀ plots and IC₅₀ values. Panel b has been incorporated into the revised manuscript (as Extended Data Figure 1f).

Reviewer Reports on the Second Revision:

Referee #3 (Remarks to the Author):

The authors have now suitably addressed all concerns raised and this manuscript is suitable for publication.